# Eco-evolutionary dynamics modulate plant responses to global change depending on plant diversity and species identity

Peter Dietrich[1,2]*, Jens Schumacher[3], Nico Eisenhauer[2,4], Christiane Roscher[1,2]

[1]Department of Physiological Diversity, UFZ, Helmholtz Centre for Environmental Research, Leipzig, Germany; [2]German Centre of Integrative Biodiversity Research (iDiv) Halle-Jena-Leipzig, Leipzig, Germany; [3]Institute of Mathematics, Stochastics, Friedrich Schiller University Jena, Jena, Germany; [4]Institute of Biology, Experimental Interaction Ecology, Leipzig University, Leipzig, Germany

**Abstract** Global change has dramatic impacts on grassland diversity. However, little is known about how fast species can adapt to diversity loss and how this affects their responses to global change. Here, we performed a common garden experiment testing whether plant responses to global change are influenced by their selection history and the conditioning history of soil at different plant diversity levels. Using seeds of four grass species and soil samples from a 14-year-old biodiversity experiment, we grew the offspring of the plants either in their own soil or in soil of a different community, and exposed them either to drought, increased nitrogen input, or a combination of both. Under nitrogen addition, offspring of plants selected at high diversity produced more biomass than those selected at low diversity, while drought neutralized differences in biomass production. Moreover, under the influence of global change drivers, soil history, and to a lesser extent plant history, had species-specific effects on trait expression. Our results show that plant diversity modulates plant-soil interactions and growth strategies of plants, which in turn affects plant eco-evolutionary pathways. How this change affects species' response to global change and whether this can cause a feedback loop should be investigated in more detail in future studies.

## Editor's evaluation

Dietrich et al. aimed to test the hypothesis that a decline in species richness due to various global change drivers selects for traits that will make species more vulnerable to the further effects of these drivers, amplifying thus the initial diversity decline. This research is of prime importance to botanists, plant ecologists, and ecosystem ecologists wanting to model the effects of global climate change on plant diversity and productivity.

## Introduction

Human activities, such as the combustion of fossil fuels and the intensification of agriculture, are leading to global environmental changes, causing increased air temperatures, altered precipitation patterns, and rising amounts of nitrogen to ecosystems (IPCC, *Pörtner et al., 2021*). The consequences are more frequent extreme weather events such as droughts (*Dai et al., 2018*) and a growing accumulation of nitrogen in the soils (*Holland et al., 2005*). Both, drought and increased nitrogen

*For correspondence: peter.dietrich@idiv.de

**eLife digest** Over the last hundred years, human activities including burning of fossil fuels, clearing of forests, and fertilizer use have caused environmental changes that have resulted in many species of plants, animals and other forms of life becoming extinct.

Loss of plant species can change the local environment by, for example, altering the availability of nutrients and local communities of microbes in the soil. This may, in turn, cause remaining plant species to develop differently: they may take up fewer resources or become more prone to pathogens, both of which may alter their physical appearance. However, little is known about whether this happens and, if so, how rapidly such changes occur.

Since 2002, researchers in Germany have been running a long-term project known as the Jena Experiment to study how plants behave when they grow in communities with different numbers of other plant species. For the experiment, various species of grass and other plants commonly found in grasslands were grown together in different combinations. Some plots contained many species (referred to as "high diversity") and others contained only a few ("low diversity").

Here, Dietrich et al. collected seeds from four grasses grown for 12 years in Jena Experiment plots with two or six plant species. The seeds were then transferred to pots and grown in a greenhouse using soil either from the plot where the seeds originated or from another plot with a different diversity level. To simulate human-made changes in the environment, the team added nitrogen fertilizer or decreased how much they watered some of the plants.

The greenhouse experiment showed that after receiving nitrogen fertilizer, the seeds from the high diversity Jena Experiment plots grew into larger plants than the seeds from the low diversity plots. But there was no difference in size when the plants were watered less. Moreover, both fertilizer and watering treatment had different effects on the plants' physical appearance (root and leaf architecture) depending on the soil in which they were growing in.

The findings of Dietrich et al. suggest that plants may respond differently to changes in their environment based on their origins and the soil they are growing in. This study provides the first indication that species loss could accelerate a further loss of species due to changes in how the plants develop and the communities of organisms living in the soil.

input, in turn, further influence ecosystems and climatic conditions; hence, they are known as major global change drivers (*Sage, 2020*).

Some of the most tremendous negative effects of global change are changes in ecological communities (*Dornelas et al., 2014*) and the extinction of species (*Sage, 2020*), whereby plant species are particularly affected due to their low mobility, with drastic consequences for the functioning of ecosystems. Studies in grassland biodiversity experiment have shown that low- and high-diversity plant communities significantly differ in their productivity and stability (*Isbell et al., 2015*; *Marquard et al., 2009*; *Tilman et al., 2006*). Low-diversity communities were shown to lose productivity over time, while high-diversity communities are more stable, so that plant diversity-productivity relationships become more positive over time (*Cardinale et al., 2007*; *Meyer et al., 2016*; *Reich et al., 2012*). A different development of plant-soil and plant-plant interactions at low and high diversity are assumed to be the important drivers of these strengthening biodiversity effects (*Eisenhauer et al., 2019*; *Thakur et al., 2021*). At low plant diversity, an accumulation of soil-borne pathogens might be responsible for lower plant community productivity (*Mommer et al., 2018*; *Thakur et al., 2021*), while in high-diversity communities, complementarity effects among plants and an accumulation of soil-borne mutualists inhibit such negative processes, causing a higher productivity of these plant communities (*Barry et al., 2019*; *Cardinale et al., 2007*; *Reich et al., 2012*). Next to these biotic drivers, also diversity-dependent differences in abiotic conditions can influence plant community productivity (*Barry et al., 2019*). Previous studies demonstrated that a decrease of species richness alters the vegetation structure and density (*Lorentzen et al., 2008*), which in turn can have strong impacts on the availability of light, water, and nutrients for plants (*Bachmann et al., 2018*; *Fischer et al., 2018*; *Lange et al., 2019*).

Consequently, these findings raise the question, whether populations of the same plant species develop differently over time when growing at high or low diversity due to differences in

eco-evolutionary feedbacks (*Bailey et al., 2006*; *Linhart, 1988*; *Post and Palkovacs, 2009*; *terHorst et al., 2016*). There is empirical evidence that plant individuals at high diversity are selected for greater niche complementarity among species leading to a more complete use of available resources (*Zuppinger-Dingley et al., 2014*). At low plant diversity, in contrast, the accumulation of soil-borne pathogens may cause persistent species to adapt to this increase by producing more defense compounds. Thus, over time selection could favor individuals that invest more in defense and less in growth, which could be measurable as lower biomass production and altered values of traits related to growth (*Dietrich et al., 2020b*; *Lau and Lennon, 2011*; *Mraja et al., 2011*). For example, plants could decrease in height, shift along the leaf economic spectrum, and/or change their root architecture so that they can allocate more resources into defense. Indeed, several studies in the Jena Experiment demonstrated that plants not only show phenotypic plasticity in response to altered growth conditions in low- and high-diversity communities, but provided first evidence for micro-evolutionary processes. For example, only four years after sowing, the grass species *Lolium perenne* L. showed genetic differentiation from the source population, which was probably due to genetic drift as well as genotype-specific interactions with other species in plant communities of different diversity (*Nestmann et al., 2011*). Moreover, a recent study by *van Moorsel et al., 2019* in the Jena Experiment demonstrated genetic and epigenetic divergence among plants originated either from monocultures or mixtures in three out of five studied species, suggesting rapid emergence of low-diversity and high-diversity genotypes.

Taken together, low and high plant diversity may differently affect eco-evolutionary feedbacks and thus the microevolution of plants, which could have the consequence that plants selected at different diversity respond differently to global change drivers. In a previous transplant experiment in the field, it was shown that some of the studied grassland species showed differences in their phenotype depending on plant history (monoculture or mixture) and soil environment (*Lipowsky et al., 2011*). Several greenhouse or common garden studies came to similar conclusions, that is, that plants selected at low or high diversity or grown with 'own' or different soil biota vary in their productivity and trait expression (*Hahl et al., 2020*; *Rottstock et al., 2017*; *van Moorsel et al., 2018b*; *Zuppinger-Dingley et al., 2014*). Such diversity-induced differences in the phenotype could lead to different responses of plants to global change drivers. For example, it is possible that, plants selected at low diversity have a phenotype which is related to a lower resistance against drought than plants selected at high diversity. Such changes would contribute to a faster extinction of species, which makes research into these processes an essential frontier.

In summary, differences in plant-plant and plant-soil interactions at low and high diversity may lead to differences in eco-evolutionary feedbacks; however, little is known about how rapidly and pervasively these differences occur (*terHorst et al., 2016*). Moreover, it is not known whether these differences affect the response of plants to global change drivers (*Pugnaire et al., 2019*), such as drought, nitrogen input, or a combination of both, which is assumed to be a common scenario in the future (*Craven et al., 2016*; *Sage, 2020*). To address these knowledge gaps, we performed a common garden experiment using plant and soil material from a long-running biodiversity experiment (Jena Experiment). For our study, we collected seeds of four grass species and took samples of soil biota (soil samples), which both had been selected for 14 years, either at low or high diversity (communities with two or six plant species and different plant species composition). The selection of species was based on a sufficient seed production of plants, whereby the four grass species show a spectrum of different growth strategies ranging from very dominant to subordinate. Plants were grown either in soil inoculated with their home soil biota, that is, soil biota of the community, where the seeds had been collected, or in soil inoculated with away soil biota, that is, with soil biota of a different plant community (differing in plant diversity or composition). The aim of the study was to test whether plant history (origin of plants: two- or six-plant species communities), soil history (origin of soil biota: two- or six-plant species communities), and soil treatment (home or away) influence the response of the plants to global change. Therefore, plants were either non-treated (control), or exposed to drought, increased nitrogen input, or a combination of both, drought and nitrogen input, in a full factorial design. We hypothesized that.

(I) plant and soil communities in the field developed differently at low and high diversity over time, that is, plants at low diversity were selected to invest more resources into defense due to higher accumulation of soil-borne pathogens, while plants at high diversity were selected to invest into growth

via complementarity effects and higher number of soil-borne mutualists. Therefore, we expected that productivity and trait expression of control plants (without global change drivers) in our common garden experiment differ depending on plant history, soil history, and soil treatment; for example, offspring of plants selected at low diversity has a lower biomass production and altered values of traits related to growth, because more resources are invested into defense, while the opposite is true for offspring of plants selected at high diversity.

(II) global change drivers have a strong impact on biomass production and trait expression of plants. We expected that drought reduces, and nitrogen input increases, plant biomass, and that single global change drivers alter the expression of traits related to growth. The combination of both global change drivers, however, has no net effect, because drought and nitrogen input compensate each other's impacts.

(III) because of different development of plants and soil communities at low and high diversity, offspring of plants selected at different diversity and grown in different soil (home vs. away soil, soil from low- vs. high-diversity communities) respond differently to global change drivers regarding performance and trait expression.

## Results

### Hypothesis 1: Offspring of plants selected at different diversity and grown in different soil (high vs. low diversity, home vs. away) show differences in productivity and trait expression

#### Biomass production

Overall, legacy treatments had a low impact on biomass production, and explain a small portion of variance (*Table 1*). Plants grown in soil of six-species communities tended to produce more root biomass than plants in soil of two-species communities in the control (*Table 2*; *Figure 1*). At species-level, *Arrhenatherum elatius* produced more root biomass and had higher root-shoot ratio, and *Dactylis glomerata* produced more shoot and total biomass in soil of six-species than two-species communities (*Figures 1 and 2a*; *Appendix 1—Tables 1 and 3*). The other two species, *Poa trivialis* and *Alopecurus pratensis* did not differ significantly in biomass production dependent on soil or plant history (*Figures 1 and 2a*; *Appendix 1—Tables 2 and 4*). Initial shoot number showed no influence on later biomass production except for shoot biomass of *D. glomerata* and root biomass of *A. elatius*, which, however, did not change the general patterns.

#### Plant traits and pathogen infestation

Legacy treatments had no consistent effects across the four species on the expression of shoot, leaf, or root traits in the control (*Appendix 1—table 5*). At species-level, legacy treatments did not affect trait expression in *A. elatius* (*Figure 2a*; *Appendix 1—table 6*). Plants of *A. pratensis* were taller in home than in away-different soil and had thicker roots (higher root diameter) in six- than in two-species soil (*Figure 2a*; *Appendix 1—table 7*). Plants of *D. glomerata* had higher leaf greenness and stomatal conductance, when seeds originated from two-species communities (*Figure 2a*; *Appendix 1—table 8*). Plants of *P. trivialis* had lower shoot nitrogen concentration and root diameter, and higher SRL in home soil than in away soil (*Figure 2a*; *Appendix 1—table 9*).

We found a low pathogen infestation of *A. elatius* and *A. pratensis* (0.8% ± 1.9% (SD) and 0.1% ± 0.5%, respectively), mainly by the rust species *Puccinia graminis* Pers. and *Puccinia coronata* Corda. Plants of *D. glomerata* and *P. trivialis*, in contrast, were strongly infested by the mildew *Blumeria graminis* (DC.) Speer (3.1% ± 4.2% and 8.6% ± 16.5%, respectively). Regarding legacy treatments, *D. glomerata* plants had a lower infestation when grown in home soil than in away soil, while mildew infestation of *P. trivialis* plants did not differ between legacy treatments (*Figure 2a*; *Appendix 1—table 10*).

**Table 1.** Summary of mixed-effect model analyses testing the effects of species identity (N = 4), legacy treatments (plant history, soil history, soil treatment), global change treatments (drought, nitrogen input), and their interactions on plant performance (total biomass, shoot biomass, root biomass). Shown are degrees of freedom (Df), Chi², p-values (p) and explained variance (VD; calculated with variance decomposition; in %). Significant effects (p < 0.05) are given in bold, marginally significant effects (p < 0.1) in italics.

| | Total biomass | | | | Shoot biomass | | | | Root biomass | | | |
|---|---|---|---|---|---|---|---|---|---|---|---|---|
| | Df | Chi² | p | VD | Df | Chi² | p | VD | Df | Chi² | p | VD |
| Species identity (ID) | 3 | 73.25 | **< 0.001** | 19.54 | 3 | 80.17 | **< 0.001** | 17.36 | 3 | 121.30 | **< 0.001** | 63.65 |
| Plant history | 1 | 3.48 | *0.062* | 0.34 | 1 | 1.36 | 0.244 | 0.05 | 1 | 3.40 | *0.065* | 0.23 |
| Soil history | 1 | 0.01 | 0.915 | < 0.01 | 1 | 0.04 | 0.851 | < 0.01 | 1 | 0.49 | 0.484 | < 0.01 |
| Soil treatment | 2 | 2.17 | 0.338 | 0.02 | 2 | 1.20 | 0.548 | < 0.01 | 2 | 3.66 | 0.161 | 0.08 |
| Drought (D) | 1 | 83.05 | **< 0.001** | 10.30 | 1 | 110.26 | **< 0.001** | 14.11 | 1 | 2.81 | *0.094* | 0.09 |
| Nitrogen input (N) | 1 | 257.26 | **< 0.001** | 22.83 | 1 | 425.93 | **< 0.001** | 34.17 | 1 | 15.89 | **< 0.001** | 0.78 |
| Species ID x Plant history | 3 | 0.71 | 0.872 | < 0.01 | 3 | 1.77 | 0.621 | < 0.01 | 3 | 0.63 | 0.890 | < 0.01 |
| Species ID x Soil history | 3 | 1.68 | 0.642 | < 0.01 | 3 | 0.18 | 0.980 | < 0.01 | 3 | 3.64 | 0.303 | 0.02 |
| Species ID x Soil treatment | 6 | 4.29 | 0.638 | < 0.01 | 6 | 6.64 | 0.355 | 0.01 | 6 | 2.30 | 0.891 | < 0.01 |
| Species ID x D | 3 | 52.00 | **< 0.001** | 3.08 | 3 | 43.11 | **< 0.001** | 1.92 | 3 | 98.61 | **< 0.001** | 4.45 |
| Species ID x N | 3 | 30.46 | **< 0.001** | 1.55 | 3 | 33.73 | **< 0.001** | 1.35 | 3 | 18.28 | **< 0.001** | 0.62 |
| D x N | 1 | 35.27 | **< 0.001** | 1.82 | 1 | 27.47 | **< 0.001** | 1.10 | 1 | 10.90 | **0.001** | 0.40 |
| Species ID x Plant history x D | 4 | 0.92 | 0.922 | < 0.01 | 4 | 4.42 | 0.353 | 0.02 | 4 | 0.72 | 0.948 | < 0.01 |
| Species ID x Soil history x D | 4 | 1.17 | 0.883 | < 0.01 | 4 | 5.33 | 0.255 | 0.05 | 4 | 0.54 | 0.969 | < 0.01 |
| Species ID x Soil treatment x D | 8 | 2.81 | 0.946 | < 0.01 | 8 | 4.78 | 0.781 | < 0.01 | 8 | 3.30 | 0.914 | < 0.01 |
| Species ID x Plant history x N | 4 | 2.66 | 0.617 | < 0.01 | 4 | 5.75 | 0.219 | 0.05 | 4 | 1.69 | 0.792 | < 0.01 |
| Species ID x Soil history x N | 4 | 6.59 | 0.159 | 0.13 | 4 | 3.47 | 0.482 | < 0.01 | 4 | 5.26 | 0.262 | 0.04 |
| Species ID x Soil treatment x N | 8 | 9.35 | 0.314 | 0.03 | 8 | 4.62 | 0.797 | < 0.01 | 8 | 15.48 | *0.050* | 0.25 |
| Species ID x Plant history x D x N | 4 | 14.85 | **0.005** | 0.50 | 4 | 27.25 | **< 0.001** | 0.87 | 4 | 12.61 | **0.013** | 0.32 |
| Species ID x Soil history x D x N | 4 | 13.14 | **0.011** | 0.43 | 4 | 14.39 | **0.006** | 0.37 | 4 | 11.81 | **0.019** | 0.28 |
| Species ID x Soil treatment x D x N | 8 | 6.19 | 0.626 | < 0.01 | 8 | 7.91 | 0.442 | < 0.01 | 8 | 4.81 | 0.778 | < 0.01 |

**Table 2.** Summary of mixed-effect model analyses testing the effects of species identity (N = 4), legacy treatments (plant history, soil history, soil treatment), and their interactions on plant performance (total biomass, shoot biomass, root biomass and root-shoot ratio), when non-treated (control) or treated with global change drivers (drought, nitrogen input, drought and nitrogen input [D x N]).

Shown are degrees of freedom (Df), Chi² and p-values (p). Significant effects (p < 0.05) are given in bold, marginally significant effects (p < 0.1) in italics.

**Total biomass**

|  | Control | | | Drought | | | Nitrogen | | | D x N | | |
|---|---|---|---|---|---|---|---|---|---|---|---|---|
|  | Df | Chi² | p | Df | Chi² | p | Df | Chi² | p | Df | Chi² | p |
| Species ID | 3 | 57.93 | **< 0.001** | 3 | 37.43 | **< 0.001** | 3 | 60.10 | **< 0.001** | 3 | 27.83 | **< 0.001** |
| Plant history (PH) | 1 | 0.04 | 0.840 | 1 | 2.08 | 0.149 | 1 | 4.86 | **0.027** | 1 | 1.17 | 0.280 |
| Soil history (SH) | 1 | 2.60 | 0.107 | 1 | 0.44 | 0.507 | 1 | 1.15 | 0.283 | 1 | 0.10 | 0.756 |
| Soil treatment (ST) | 2 | 0.80 | 0.670 | 2 | 0.46 | 0.795 | 2 | 3.78 | 0.151 | 2 | 3.28 | 0.194 |
| Species ID x PH | 3 | 1.05 | 0.790 | 3 | 3.44 | 0.328 | 3 | 1.37 | 0.712 | 3 | 0.04 | 0.998 |
| Species ID x SH | 3 | 3.06 | 0.382 | 3 | 2.48 | 0.478 | 3 | 1.61 | 0.657 | 3 | 2.48 | 0.479 |
| Species ID x ST | 6 | 3.44 | 0.752 | 6 | 3.55 | 0.737 | 6 | 4.91 | 0.555 | 6 | 4.04 | 0.672 |

**Shoot biomass**

|  | Control | | | Drought | | | Nitrogen | | | D x N | | |
|---|---|---|---|---|---|---|---|---|---|---|---|---|
|  | Df | Chi² | p | Df | Chi² | p | Df | Chi² | p | Df | Chi² | p |
| Species ID | 3 | 45.01 | **< 0.001** | 3 | 47.33 | **< 0.001** | 3 | 64.80 | **< 0.001** | 3 | 43.66 | **< 0.001** |
| Plant history (PH) | 1 | 0.03 | 0.859 | 1 | 0.45 | 0.502 | 1 | 2.56 | 0.110 | 1 | 0.77 | 0.381 |
| Soil history (SH) | 1 | 1.57 | 0.211 | 1 | 0.11 | 0.743 | 1 | 1.97 | 0.161 | 1 | 0.06 | 0.799 |
| Soil treatment (ST) | 2 | 0.24 | 0.886 | 2 | 2.51 | 0.286 | 2 | 4.39 | 0.112 | 2 | 1.91 | 0.385 |
| Species ID x PH | 3 | 0.18 | 0.980 | 3 | 6.79 | *0.079* | 3 | 4.34 | 0.227 | 3 | 0.60 | 0.900 |
| Species ID x SH | 3 | 7.50 | *0.058* | 3 | 2.08 | 0.556 | 3 | 0.06 | 0.996 | 3 | 0.67 | 0.881 |
| Species ID x ST | 6 | 6.46 | 0.374 | 6 | 2.67 | 0.849 | 6 | 7.67 | 0.263 | 6 | 2.27 | 0.893 |

**Root biomass**

|  | Control | | | Drought | | | Nitrogen | | | D x N | | |
|---|---|---|---|---|---|---|---|---|---|---|---|---|
|  | Df | Chi² | p | Df | Chi² | p | Df | Chi² | p | Df | Chi² | p |
| Species ID | 3 | 107.40 | **< 0.001** | 3 | 93.04 | **< 0.001** | 3 | 101.11 | **< 0.001** | 3 | 81.40 | **< 0.001** |
| Plant history (PH) | 1 | < 0.01 | 0.957 | 1 | 2.79 | *0.095* | 1 | 3.34 | *0.068* | 1 | 1.20 | 0.274 |
| Soil history (SH) | 1 | 2.79 | *0.095* | 1 | 1.27 | 0.259 | 1 | 0.05 | 0.828 | 1 | 0.11 | 0.742 |
| Soil treatment (ST) | 2 | 0.60 | 0.740 | 2 | 0.74 | 0.691 | 2 | 2.08 | 0.354 | 2 | 4.40 | 0.111 |
| Species ID x PH | 3 | 2.74 | 0.434 | 3 | 1.34 | 0.720 | 3 | 0.78 | 0.855 | 3 | 0.76 | 0.860 |
| Species ID x SH | 3 | 3.68 | 0.299 | 3 | 3.00 | 0.391 | 3 | 3.11 | 0.375 | 3 | 4.02 | 0.259 |
| Species ID x ST | 6 | 7.55 | 0.273 | 6 | 5.43 | 0.490 | 6 | 3.05 | 0.803 | 6 | 9.25 | 0.160 |

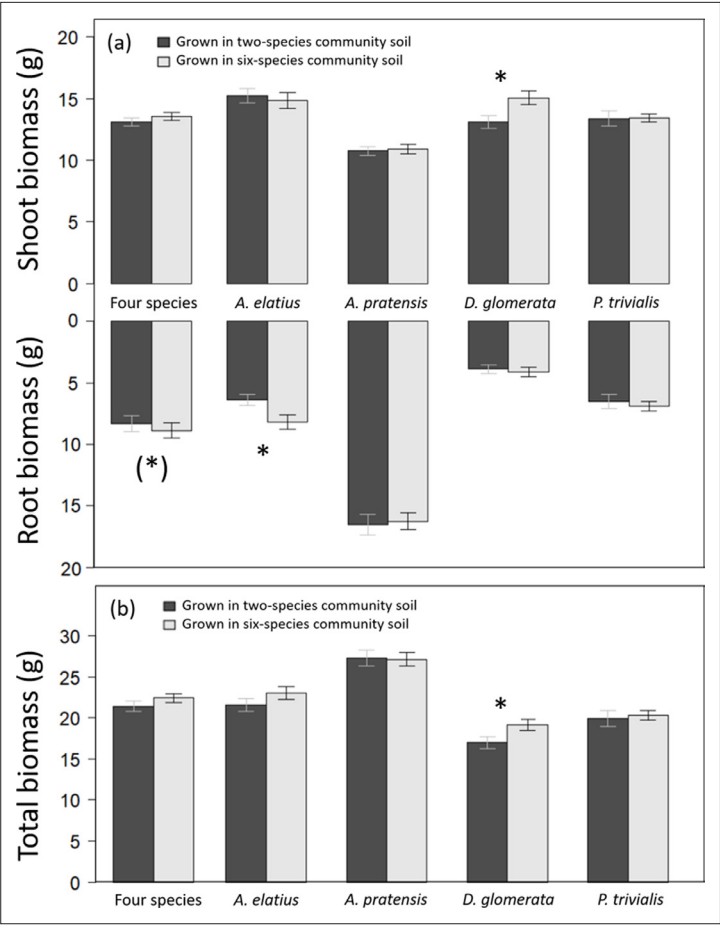

**Figure 1.** Shoot and root biomass production (**a**), and total biomass production (**b**) of plants grown either in soil originated from two-species or six-species communities across all four study species and separately for each species. Bars show mean values (± 1 SE); stars above bars indicate significant differences (p < 0.05), stars in brackets indicate marginally significant differences (p < 0.1).

## Hypothesis 2: Global change drivers have a strong impact on biomass production and trait expression

### Biomass production

Overall, global change drivers had a strong impact on almost all response variables, and explained a large portion of variance, together with species ID (*Table 1*; *Figure 3b–d*; *Appendix 2—Tables 1–9*). Compared to control plants, drought reduced shoot biomass production, which was found across all study species and at species-level (*Figure 3a and d*). In contrast, drought did not have consistent effects on root biomass (*Figure 3a and d*). Drought had positive impact on root biomass of *A. elatius* and *D. glomerata*, while root biomass of *A. pratensis* decreased under drought and did not change significantly in *P. trivialis* (*Figure 3d*). Total biomass production was decreased, when plants were exposed to drought (*Figure 3a and d*) except for *D. glomerata*, where it was not different from the control (*Figure 3d*). Root-shoot ratios increased under drought (*Figure 3a and d*), which was found for all species except for *P. trivialis* (no significant change; *Figure 3d*).

Nitrogen input increased shoot, root, and thus also total biomass across the four species (*Figure 3b*) as well as in separate analyses of *A. elatius* and *A. pratensis* (*Figure 3e*). Plants of *D. glomerata* and *P. trivialis* did not change root biomass when fertilized (*Figure 3e*). Nitrogen input caused a decrease in root-shoot ratio in all species (*Figure 3a and e*).

When plants were treated with both global change drivers in combination, the negative impact of drought on shoot biomass was cancelled out by the positive impact of nitrogen input leading to an overall slight increase of shoot biomass (compared to control plants) that was also significant at

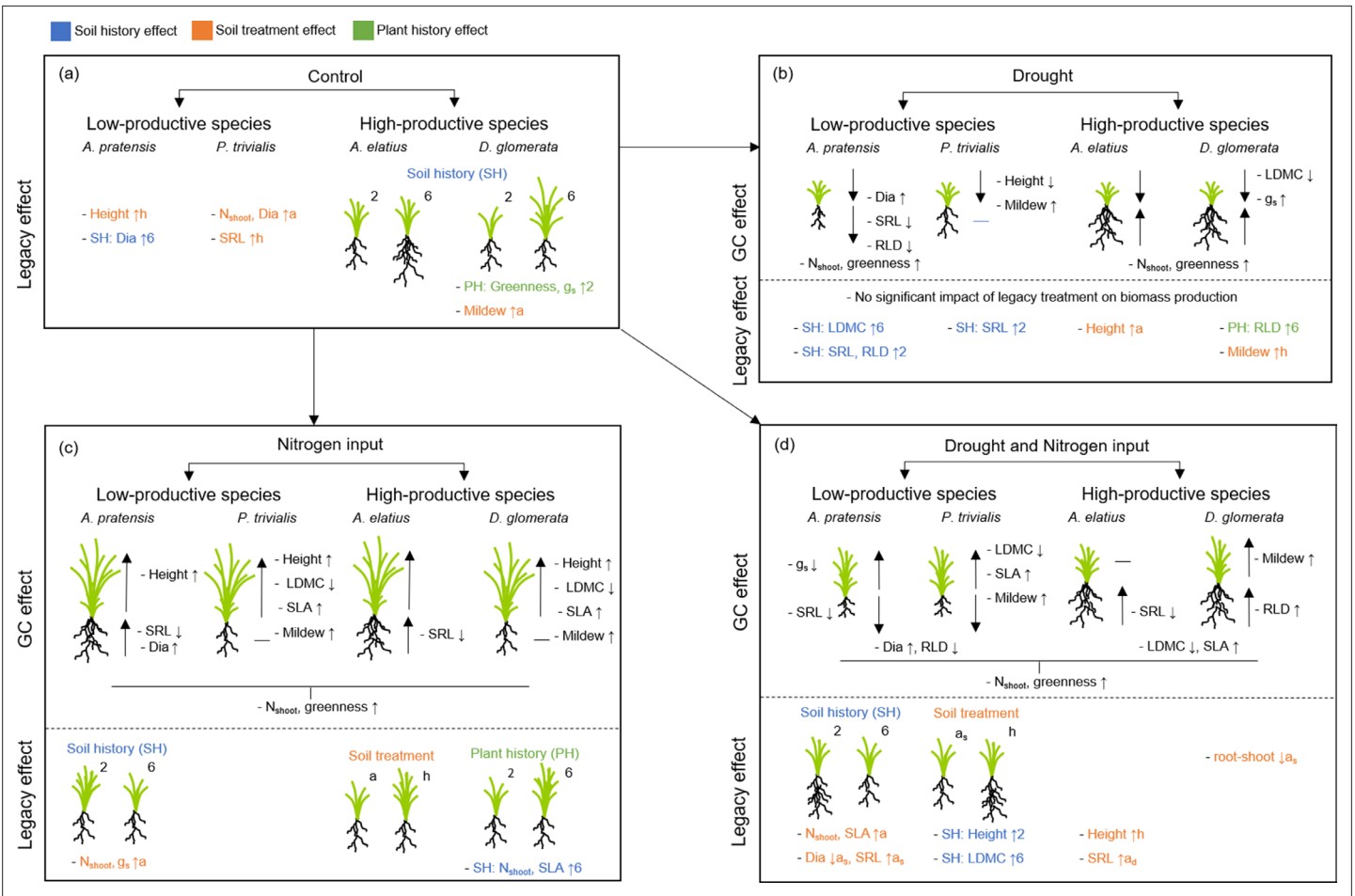

**Figure 2.** Schematic overview of the results of the common garden experiment testing how plants with a different origin (plant history) or grown in different soil (soil history, soil treatment) differ in performance and trait expression (**a**), under the influence of global change drivers drought (**b**), nitrogen input (**c**), and the combination of both (**d**). Illustrated is the impact of legacy treatments (= "legacy effect") and global change treatments ( = "global change effect") on shoot and root biomass production as well as on plant traits (growth height ("Height"), shoot nitrogen concentration ("N$_{shoot}$"), leaf greenness ("Greenness"), leaf dry matter content ("LDMC"), specific leaf area ("SLA"), stomatal conductance ("g$_s$"), mildew infestation ("Mildew"), root diameter ("Dia"), specific root length ("SRL"), root length density ("RLD")) of the four study species. For legacy effects, schematic illustrations of plants indicate differences in shoot and/or root biomass, when originated from two-species ( = 2) or six-species ( = 6) communities ( = plant history (PH); green color), when grown in two-species (=2) or six-species ( = 6) community soil ( = soil history (SH), blue color), or when grown in away ( = a) or home ( = h) soil ( = soil treatment; a$_s$ = away same soil; orange color). Arrows behind traits (for legacy effects) indicate, in which treatment group the value was significantly higher (arrow up) or lower (arrow down), e.g. "- SH: SLA ↑6" indicate that SLA in plants grown in six-species soil was higher than in two-species soil and "- LDMC ↑h" indicate that LDMC was higher in plants grown in home than in away soil. For global change effects, schematic illustrations of plants indicate whether shoot and/or root biomass of plants increased (big arrow up) or decreased (big arrow down) due to the impact of the respective global change driver (black horizontal line indicate no change). Small arrows behind traits (for global change effects) indicate and increase (arrow up) or decrease (arrow down) of the trait value due to the impact of the respective global change driver.

the species-level except for *A. elatius* (*Figure 3c and f*). Consistent with this, the positive impact of nitrogen input on root biomass was also cancelled out by drought when plants were treated with both global change drivers, i.e., control plants and plants treated with both global change drivers did not differ in root biomass production, across all study species (*Figure 3c*). At species-level, the combination of both global change drivers had an additive effect on root biomass production of *A. elatius* and *D. glomerata*, that is, plants of both species showed highest root biomass when treated with both global change drivers (*Figure 3f*). In *A. pratensis* and *P. trivialis*, both global change drivers in combination decreased root biomass production (*Figure 3f*). Taken together, the combination of both global change drivers led to a slight increase in total biomass production, across all study species and for the high-productive species *A. elatius* and *D. glomerata*, while plants of the low-productive species *A. pratensis* and *P. trivialis* had a similar total biomass production as in the control (*Figure 3c and f*).

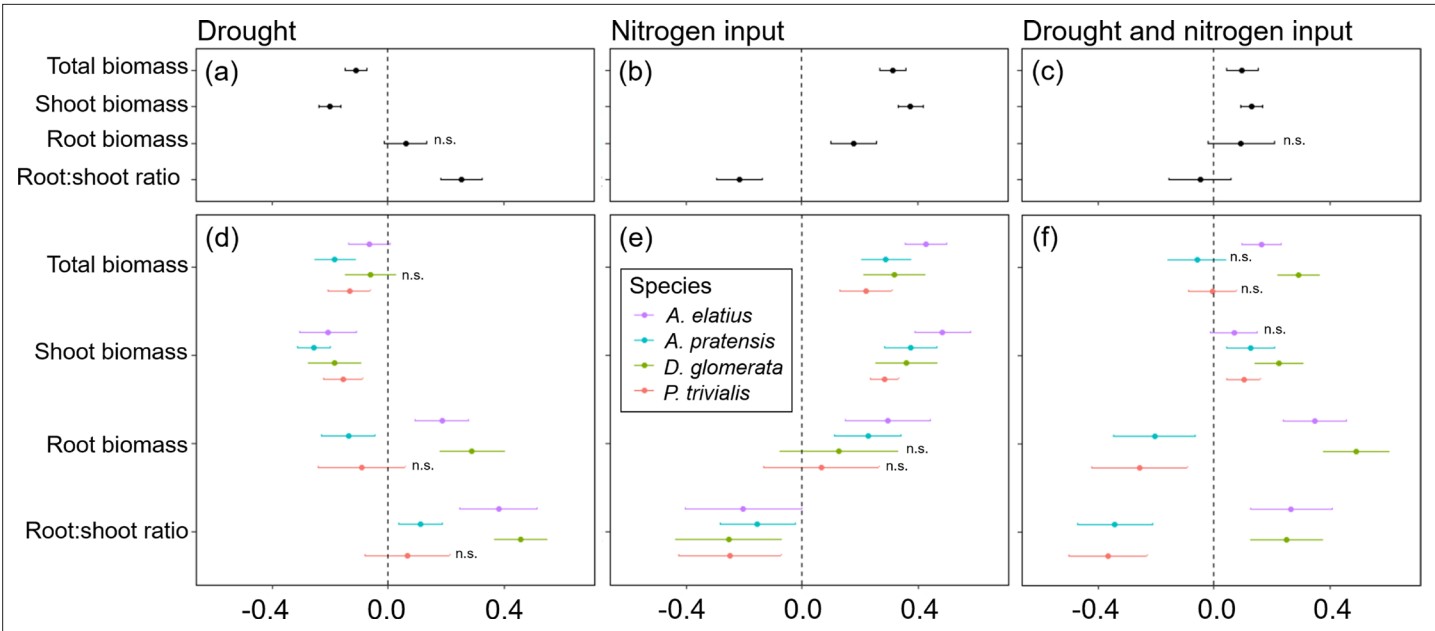

**Figure 3.** Response of plants treated with drought, nitrogen input, or a combination of both relative to non-treated plants (control) for total biomass, shoot biomass, root biomass, and root-shoot ratio across four study species (**a–c**) and separately for each species (**d–f**). Points are means and error bars are standard deviation. No symbol indicates significant differences between plants treated with global change driver and control plants, n.s. indicate no significant difference.

Root-shoot ratios were as low as in fertilized plants, across all species and in *P. trivialis* (***Figure 3c and f***). Plants of *A. elatius* and *D. glomerata* increased root-shoot ratios, similar to plants under drought (***Figure 3f***). In contrast, *A. pratensis* strongly decreased root-shoot ratios resulting in the lowest values compared to the other treatments (***Figure 3f***).

## Plant traits and pathogen infestation

Across all study species, drought did not significantly alter growth height, but nitrogen input increased height (***Figure 3b and c***). When treated with both global change drivers, drought canceled out the positive nitrogen input effect, leading to similar height of plants treated with both global change drivers and control plants (***Figure 3d***). Further, drought and nitrogen input increased shoot nitrogen concentrations and leaf greenness, with additive effects when both global change drivers were applied together (***Figure 3b–d***). Drought did not influence LDMC and SLA, while nitrogen input decreased LDMC and increased SLA (***Figure 3b and c***). When treated with both global change drivers, drought mitigated the decrease of LDMC under nitrogen input, while the increase of SLA under nitrogen input did not change with drought (***Figure 3d***). Stomatal conductance was increased, when plants were treated with drought, but did not change when fertilized irrespective of the drought treatment (***Figure 3b–d***). In terms of root traits, we found a decrease of RLD under drought (irrespective of fertilization) and an increase in root diameter under nitrogen input (irrespective of drought; ***Figure 3b–d***). Results of species-specific changes in trait expression under global change drivers can be found in ***Figure 3b–d*** and Appendix 2.

In *D. glomerata*, mildew infestation remained unchanged when treated with drought, but increased with nitrogen input. When treated with both global change drivers, mildew infestation was as high as in fertilized plants (***Figure 3b–d***). In *P. trivialis*, mildew infestation was increased under drought and when fertilized, while the combination of both global change drivers led to the highest mildew infestation (***Figure 3b–d***).

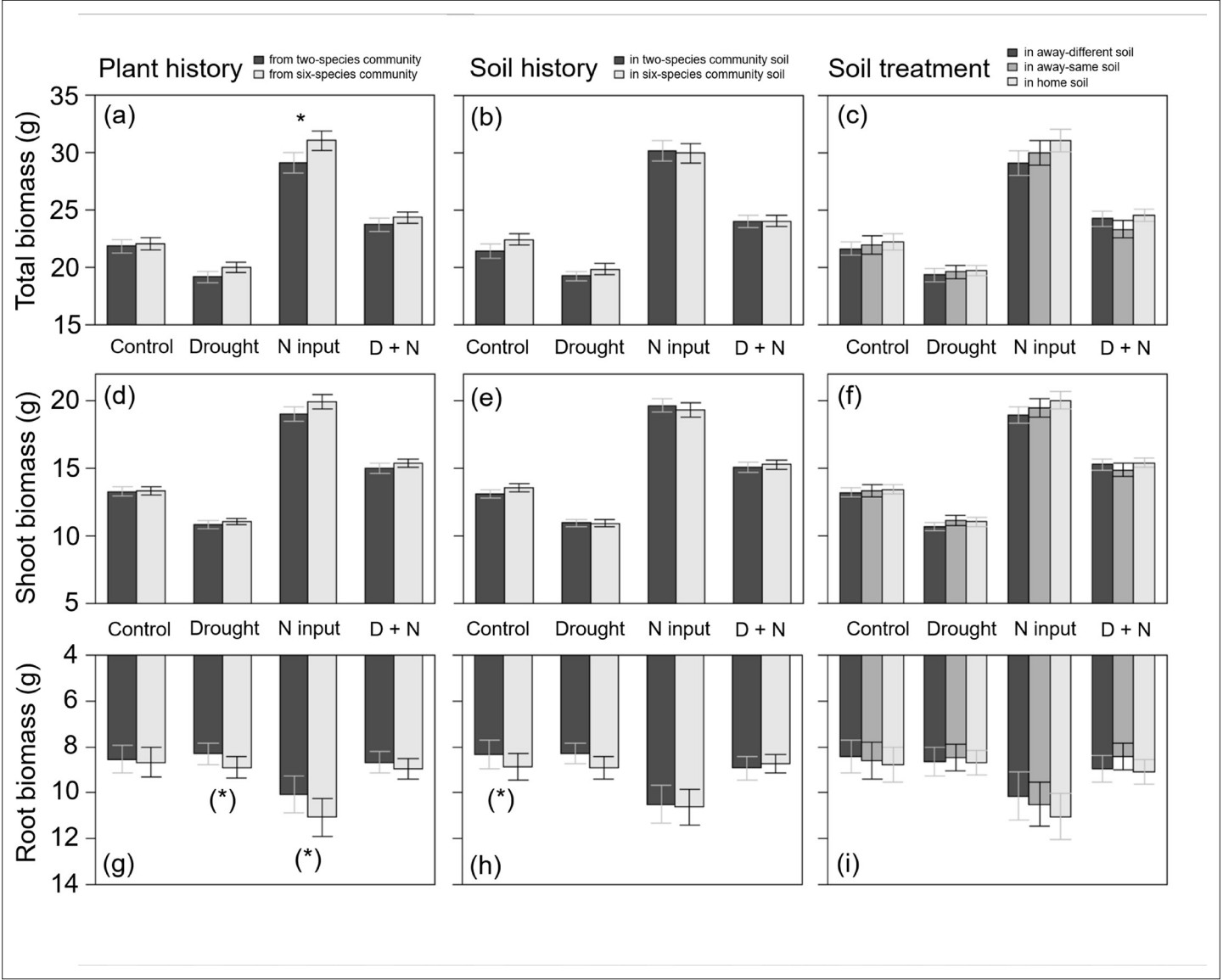

**Figure 4.** Total biomass (**a–c**), shoot biomass (**d–f**), and root biomass (**g–i**) of plants (across all four study species) originated from two- or six-species communities (plant history; **a, d, g**); grown in soil originated from two-species or six-species communities (soil history; **b, e, h**); or grown in home, away-same or away-different soil (soil treatment; **c, f, i**) and were either non-treated (control) or treated with drought, nitrogen input (N input) or a combination of both (D + N). Bars show mean values (± 1 SE); stars above bars indicate significant differences (p < 0.05), stars in brackets indicate marginally significant differences (p < 0.1).

## Hypothesis 3: Offspring of plants selected at different diversity and grown in different soil (high vs. low diversity, home vs. away) respond differently to global change drivers

### Biomass production

Plants from two- and six-species communities did not differ in shoot biomass production when treated with drought, but plants from six-species communities treated with drought tended to produce more root biomass than plants from two-species communities across all study species (*Table 2*; *Figure 4a, d and g*). At species-level, we found no significant effects of legacy treatments under drought (*Figure 2b*; *Appendix 1—Tables 1–4*).

When plants were fertilized, we found an impact of plant history across all study species: fertilized plants originated from six-species communities had a higher root and total biomass production than plants from two-species communities (*Table 2*; *Figure 4a and g*). This was also found in *D. glomerata*

plants, which tended to produce more shoot and total biomass when originated from six-species communities (*Figure 2c*; *Appendix 1—table 3*). In *A. elatius*, total biomass production of fertilized plants was significantly higher (and shoot biomass marginally significantly higher), when plants were grown in home and away-same than in away-different soil (*Figure 2c*; *Appendix 1—table 1*). In *A. pratensis*, fertilized plants grown in two-species community soil tended to produce more total biomass than in six-species community soil (*Figure 2c*; *Appendix 1—table 2*), while fertilized *P. trivialis* showed no significant differences (*Figure 2c*; *Appendix 1—table 4*).

When plants were treated with both global change drivers, the effects of nitrogen input were cancelled out or changed by drought, i.e., there was no significant impact of legacy treatments on biomass production across all study species and for *A. elatius* (*Table 2*; *Figure 2d*; *Appendix 1—table 1*). In *D. glomerata*, the significant influence of plant history disappeared, but plants in home and away-different soil showed higher root-shoot ratios than plants in away-same soil (*Figure 2d*; *Appendix 1—table 3*). Plants of *P. trivialis* treated with both global change drivers tended to have higher root biomass and root-shoot ratios when grown in home than in away-same soil (*Figure 2d*; *Appendix 1—table 4*). In contrast to the overall trend, *A. pratensis* was the only species which showed a similar response to nitrogen input and treatment with both global change drivers: the biomass production was higher in two- than in six-species community soil (for both global change drivers: significant higher root biomass and root-shoot ratios; *Figure 2d*; *Appendix 1—table 2*).

## Plant traits and pathogen infestation

Shoot nitrogen concentration was not influenced by plant or soil history when treated with drought, but fertilized plants in six-species soil had higher shoot nitrogen concentrations than in two-species soil (soil history effect). Moreover, fertilized plants had lower shoot nitrogen concentrations in home than in away-different soil (soil treatment effect). When plants were treated with both global change drivers, the nitrogen input effect on soil history was cancelled out by drought, while the impact of soil treatment did not: plants in home soil still had lower shoot nitrogen concentration than plants in away soil (*Appendix 1—table 5*). Other plant traits (growth height, leaf greenness, LDMC, SLA, stomatal conductance, root traits) did not significantly differ depending on legacy treatments when plants were treated with nitrogen or drought (*Appendix 1—table 5*). At species level, we found a large number of different responses depending on legacy treatments and type of global change driver, which can be found in *Figure 3b–d* and Appendix 1.

Mildew infestation of *D. glomerata* plants exposed to drought was higher in home than in away soil, while this drought effect was cancelled out by nitrogen input (*Appendix 1—table 10*). Mildew infestation of *P. trivialis* plants was not significantly influenced by legacy treatments, neither with nor without global change drivers (*Appendix 1—table 10*).

## Discussion

### Hypothesis 1: Offspring of plants selected at different diversity and grown in different soil (high vs. low diversity, home vs. away) show differences in productivity and trait expression

Our findings that *A. elatius* and *D. glomerata* plants in soil of high-diversity communities produce more biomass than in soil of low-diversity communities are in line with several greenhouse studies showing that soil conditioned by multiple plant species has a more positive impact on plant growth than soil conditioned by only one or two plant species (*Guerrero-Ramírez et al., 2019*; *Yang et al., 2015*). Plants probably suffered more from pathogens when grown in soil of low-diversity communities and/or benefitted more from interactions with soil mutualists in soil of high-diversity communities (*Dietrich et al., 2021*; *Guerrero-Ramírez et al., 2019*; *Schnitzer et al., 2011*). Interestingly, this soil legacy effect was only found in *A. elatius* and *D. glomerata*, which were both highly productive species in the long-term field experiment. In contrast, the low-productive species *A. pratensis* and *P. trivialis* showed no significant difference when grown in differently conditioned soils. We can only speculate about the underlying reasons. It is possible that soil of low-diversity communities containing *A. elatius* and/or *D. glomerata* had a higher number of (species-specific) pathogens than plots containing *A. pratensis* and/or *P. trivialis*, due the higher productivity of *A. elatius* and *D. glomerata* in the field and thus more resources for pathogens. This accumulation of species-specific pathogens could lead to

reduced productivity of *A. elatius* and *D. glomerata* offspring grown in low-diversity soil. However, it is also possible that *A. elatius* and *D. glomerata* benefit more, and *A. pratensis* and *P. trivialis* less, from soil mutualists, which can be more abundant in soil from high-diversity than in soil from low-diversity plant communities.

In contrast to biomass production, we did not find any significant influence of soil history on plant trait expression of *A. elatius* and D. *glomerata*. Nevertheless, we detected some other legacy treatment effects on plant trait expression, which was also found in related studies (*van Moorsel et al., 2018a*; *van Moorsel et al., 2018b*). The impact of soil history on root diameter of *A. pratensis*, and the impact of soil treatment (home/away) on the growth height of *A. pratensis*, on shoot nitrogen concentrations and root traits of *P. trivialis*, and on mildew infestation of *D. glomerata* indicate that plant-soil interactions influence growth, defense, and resource use strategies of plants (*Xi et al., 2021*), while this impact is species-specific. Moreover, *D. glomerata* plants had higher leaf greenness and stomatal conductance, when originated from low-diversity than from high-diversity plant communities. This could be an adaptation to higher light availability and lower soil moisture in low-diversity communities due to lower shading (*Bachmann et al., 2018*; *Fischer et al., 2018*; *Lorentzen et al., 2008*).

## Hypothesis 2: Global change drivers have a strong impact on the productivity and trait expression of plants

In accordance with our second hypothesis, we found that drought reduced total biomass production. This was mainly caused by a loss of shoot biomass, while drought differently affected root biomass production of the studied grass species. Individuals of *A. elatius* and *D. glomerata* increased in root biomass at the expense of shoot biomass, leading to higher root-shoot ratio under drought. This is a commonly observed strategy to avoid dehydration, which enables plants to tap water from deeper soil layers (in the field) and at the same time minimizes the water loss caused by transpiration (*Eziz et al., 2017*). In contrast, the low-productive species either decreased instead of increased root biomass (*A. pratensis*) or did not change root biomass production (*P. trivialis*) under drought. Interestingly, the low-productive species had a three times higher loss of total biomass under drought (*A. pratensis*: –17.1%; *P. trivialis*: –15.3%) than the highly-productive species (*A. elatius*: –6.4%; *D. glomerata*: –5.7%, no significant loss of total biomass in *D. glomerata*). Presumably, the drought resistance strategy of *A. elatius* and *D. glomerata* is more effective, which is possibly a competitive advantage under the field conditions of the Jena Experiment, explaining the dominance of these species.

The influence of drought on the expression of plant traits was plant species-specific, except for shoot nitrogen concentrations and leaf greenness, which increased under drought in three species (except *P. trivialis*). Similar results were found in previous studies (*Kocoń and Staniak, 2014*; *Rolando et al., 2015*) and indicate a general strategy against drought stress: plants decrease the cell density of shoot tissues, in line with the reduction of shoot biomass to minimize the water loss, leading to an increase in the concentration of nitrogen compounds and chlorophyll (strong correlation between leaf greenness and chlorophyll concentration were found in *Bachmann et al., 2018*). At species-level, the low-productive species showed trait expression changes similar to biomass loss under drought, while the highly-productive species *D. glomerata* decreased in LDMC and increased in stomatal conductance, which is contrary to recent studies showing the opposite strategy to resist drought (high LDMC, low stomatal conductance; *Bristiel et al., 2018*; *Jaballah et al., 2008*; *Lozano et al., 2020*). The results may differ because *D. glomerata* in our study was infested by the mildew *Blumeria graminis*, which may have changed the leaf structure, and thus also trait expression changes under drought.

Furthermore, our second hypothesis was confirmed by showing that nitrogen input increased biomass production. At species-level, shoot biomass was increased in all four species, while root biomass was enhanced only in *A. elatius* and *A. pratensis*. In *D. glomerata* and *P. trivialis*, there was also a slight, but non-significant increase in root biomass. Both species showed a strong increase in mildew infestation when fertilized. This confirms the nitrogen-disease hypothesis indicating that nitrogen supply increases infection severity by altering leaf properties and resources for pathogens (*Dordas, 2008*). In *D. glomerata* and *P. trivialis*, severe infestation by powdery mildew *Blumeria graminis* may have led to a decrease in rates of net photosynthesis (*Hibberd et al., 1996*; *Mandal et al., 2009*), so that the reduced amount of energy was mainly invested in shoot biomass, e.g., for a higher leaf turnover, and less in root biomass.

We found consistent changes in plant trait expression over all four species in response to nitrogen input: growth height (except *A. elatius*), shoot nitrogen concentrations, and leaf greenness increased in all four species when fertilized, confirming an earlier study by *Siebenkäs et al., 2015*. Further nitrogen-induced changes in trait expression were likely affected by mildew infestation: the highly-infested species (*D. glomerata, P. trivialis*) showed lower LDMC and higher SLA, while LDMC and SLA of non-infested species did not change. Probably, *D. glomerata* and *P. trivialis* plants responded to the increase in mildew infestation with a change in the leaf architecture (*Cappelli et al., 2020*), which could enable plants to turn over their leaves more quickly and thus produce constantly new and unaffected leaves. With regard to root traits, the non-infested species decreased in specific root length (and *A. pratensis* also in root diameter), while root traits remained unchanged in the highly-infested species. The decrease in SRL and increase in diameter (i.e. thicker and shorter roots) in combination with the increase in root biomass of the fertilized *A. elatius* and *A. pratensis* plants indicate that these plants changed the root architecture building fewer fine roots when nutrient availability is enhanced, which is in line with similar research (*Siebenkäs et al., 2015*).

Finally, we hypothesized that global change drivers cancel out each other's impact when applied together. This was true for the low-productive species *A. pratensis* and *P. trivialis*, which did not change in total biomass production compared to control plants as also found in other research (*Carlsson et al., 2017*). However, the strong decrease in root-shoot ratios indicates that *A. pratensis* and *P. trivialis* plants changed their growth strategies. Interestingly, the high-productive species *A. elatius* and *D. glomerata* slightly increased in total biomass, which is mainly explainable by the additive positive impact of drought and nitrogen input on root biomass, resulting in increased root-shoot ratios. Obviously, dominant (or highly-productive) species in our study benefitted more strongly from the combined application of the global change drivers in comparison to subordinate (or low-productive) species. Assuming that dry periods are becoming more frequent (*Ruosteenoja et al., 2017*) and nitrogen deposition may steadily rise (*Reay et al., 2008*), our results suggest that competitive interactions change under the impact of multiple global change drivers, and subordinate species may become more severely threatened by extinction (*Pugnaire et al., 2019*).

Moreover, our results show that the combined effects of the two global change drivers on plant trait expression may differ from the effect of drought or nitrogen input alone, with strong negative effects for some plant species (e.g. highest mildew infestation of *P. trivialis* under combined impact of global change drivers). This suggests that plants change in physiology and morphology and thus in their response to global change, when a combined impact becomes more frequent, with an unknown influence on community composition and ecosystem functioning in the long term. This finding underlines the need for studies investigating multiple, interacting global change drivers (*Rillig et al., 2019*; *Thakur et al., 2018*).

## Hypothesis 3: Offspring of plants selected at different diversity and grown in different soil (high vs. low diversity, home vs. away) respond differently to global change drivers

The soil history effect, that is, the beneficial effect of soil biota from high-diversity plant communities on biomass production of control plants, disappeared in treatments with global change drivers, which may be explainable by a change in soil community structure under drought (*Kaisermann et al., 2017*; *Pugnaire et al., 2019*) and/or nitrogen input (*Wei et al., 2018*). In line with our result, similar studies have shown that drought (*Fry et al., 2018*; *Wilschut and van Kleunen, 2021*) and nitrogen input (*in 't Zandt, 2019*) can interrupt or change plant-soil interactions.

Next to soil history, we also found altered plant responses to global change drivers when plants originated from low- or high-diversity communities (plant history). When treated with drought, there was no significant difference, but nitrogen input had a more positive impact on plants originated from high-diversity than from low-diversity communities. Possibly plants at high diversity were selected for greater niche complementarity (*Zuppinger-Dingley et al., 2014*), while plants at low diversity were selected for increased defense against species-specific pathogens (*Eisenhauer et al., 2019*), that accumulate in low-diversity environments (*Dietrich et al., 2021*; *van Ruijven et al., 2020*). Consequently, the offspring of individuals originated from high-diversity communities may be more efficient in allocating additional resources in increased growth, explaining our results. Interestingly, we did not find any significant plant history effect in plants treated with both global change drivers, indicating

that drought had a strong impact on the growth strategy of the plants and can counteract positive diversity effects.

Finally, we found that plants in home and away soil may respond differently to global change drivers; however, this was only true for the high-productive species *A. elatius*: plants benefitted more from fertilization in home and away-same than in away-different soil. The home advantage supports the idea that a decrease of plant diversity can lead to changes in plant-soil interactions and thus to differences in eco-evolutionary feedbacks at low and high diversity (*terHorst et al., 2016*). With our data in hand, we cannot determine the exact reason why we found the home advantage under fertilization but not under control conditions; however, our results show that plants may respond differently to global change drivers depending on the soil community with which they interact, which is consistent with previous findings (*Puy et al., 2021*).

Similar to the biomass production results, almost all differences in trait expression found in control plants disappeared when treated with global change drivers. Instead, many other changes in trait expression occurred depending on the type of global change driver treatment and plant species identity. Taken together, these results indicate that mainly soil biota (soil history and soil treatment) and only to a lesser extent plant history play an important role in the expression of traits under the influence of global change drivers, which is in line with previous findings (*Puy et al., 2021*). This suggests that the soil biota composition is strongly associated with the physiology and morphology of the plants. Therefore, shifts in soil biota composition due to plant species loss and/or global change driver impact can have strong effects on the response of plants to global change, which could further accelerate plant community change and species loss (*Pugnaire et al., 2019*; *Yang et al., 2021*).

## Conclusion

In the present study, we showed for the first time that offspring of plants selected at low and high plant diversity differently respond to global change and that plant-soil interactions play a significant role in this process. These differences were mainly related to changes in trait expression, while changes in biomass production were minor, and they were strongly dependent on plant species identity and their competitiveness in the field, as well as the type of global change driver (drought, nitrogen input, or both). Although we did not find clear evidence that plants selected at low diversity generally suffer more under global change than plants selected at high diversity, it is possible that the species-specific responses alter species interactions and accelerate global change effects in the long run. To better assess the risk of such a potential feedback loop, future research is urgently necessary, especially, studies that test the long-term influence of global change drivers on plants and soil biota selected at different diversity under more realistic conditions, for example as plants growing in communities under field conditions.

## Materials and methods

### The Jena Experiment

Seed and soil material for our common garden experiment was collected from a long-term biodiversity experiment, the Jena Experiment, which is located in the floodplain of the Saale river near Jena (Thuringia, Germany, 50° 55′N, 11° 35′E, 130 m a.s.l.) (*Roscher et al., 2004*; *Weisser et al., 2017*). Before the establishment of the Jena Experiment in 2002, the site was a highly fertilized arable field, which had been used for growing wheat and vegetables from the early 1960s until 2000. Mean annual air temperature recorded from 2007 to 2016 at the experimental site (Weather Station Jena-Saaleaue, Max-Planck-Institute for Biogeochemistry Jena, https://www.bgc-jena.mpg.de/wetter/) was 9.7 °C, and mean annual precipitation was 587 mm. The soil of the study site is a Eutric Fluvisol, whereas soil texture changes from sandy loam to silty clay with increasing distance from the river Saale. Thus, four blocks were arranged parallel to the riverside (*Roscher et al., 2004*).

Material for our study was collected in a sub-experiment of the Jena Experiment, the so-called Dominance Experiment. The species pool of this experiment included nine species, which often reach dominance in Central European mesophilic grasslands of the Arrhenatherion type (*Ellenberg, 1988*): five grasses, two legumes, and two herbs. Sown plant species richness levels were 1, 2, 3, 4, 6, and 9 species. Each species occurred eight times in the different compositions of each species-richness level. Moreover, each possible two-species combination was present with equal frequency

at each species-richness level of the mixtures (i.e., 2–9 species; more information about the design can be found in *Roscher et al., 2004*). In May 2002, seeds were sown with a density of 1000 viable seeds per m². Seeds from all species were purchased from a commercial supplier (Rieger-Hofmann GmBH, Blaufelden-Raboldshause, Germany). From 2002 to 2009, plants were grown in plots of 3.5 × 3.5 m; from 2010 onwards, plot size was reduced to 1 × 1 m. Plots were mown every year in June and September and mown plant material was removed. All plots were regularly weeded and never fertilized.

## Seed collection, selection of study species, and experimental plots

In summer 2016, we collected seed material from the nine species in all Dominance Experiment plots (as bulk sample per species and plot) and stored them in a freezer (at –20 °C) until further use. We chose four grass species (*Alopecurus pratensis* L., *Arrhenatherum elatius* (L.) P. Beauv. ex J. Presl et C. Presl, *Dactylis glomerata* L., *Poa trivialis* L.) as study species. Furthermore, we selected 12 plots per species, six two-species and six six-species plots, that is, 48 plots in total, where sufficient seed material was available. It should be noted that from here on we refer to plots with two plant species as 'low-diversity communities' and plots with six species as 'high-diversity communities', although the species richness of high-diversity communities can be much greater in nature depending on the considered scale. We chose six-species plots as 'high-diversity communities', because they were available with different community compositions and could better represent the effects of species richness as a measure of diversity than the replicates of the 9-species plots with the same species composition. Moreover, previous studies have shown that plant productivity and soil biota communities already differ between 2- and 6-species communities (*Dietrich et al., 2021*; *Roscher et al., 2007*). The selected plots were evenly distributed in the four blocks of the experiment (*Roscher et al., 2004*). The study species differed strongly in their biomass production in the Dominance Experiment plots. In the two-species plots, all four species showed a high biomass production; however, in the six-species plots, only *A. elatius* and *D. glomerata* were highly-productive, while *A. pratensis* and *P. trivialis* showed intermediate levels and decreased in biomass production over the years (*Clark et al., 2019*; *Roscher et al., 2007*). For simplification, from here onwards, *A. elatius* and *D. glomerata* are referred to as 'highly-productive' species, while *A. pratensis* and *P. trivialis* are referred to as 'low-productive' species.

## Preparation of background substrate and study plants

For the pot substrate, we used a sterilized sand-soil mix ( = background substrate), which was then inoculated with fresh living soil (5% of the total substrate by weight) from the selected plots. This inoculation method is a common procedure to investigate plant-soil interactions and has the advantage that only low amounts of living soil are needed and that potential abiotic feedbacks are eliminated (*Pernilla Brinkman et al., 2010*). To produce sterile background substrate, we collected 1.6 m³ soil substrate from the Jena Experiment in May 2017. This soil substrate was a mix of excavated soil material from different experimental plots, which was stored for several years at the experimental area. The soil substrate was sieved to 10 mm, homogenized, and mixed with 0.4 m³ quartz sand (WF 33, Quarzwerke GmbH, Walbeck, Germany). Afterwards, the soil-sand mix was steam-sterilized twice for 150 minutes at ~80 °C. More information about the steam-sterilization method and changes of abiotic and biotic soil properties can be found in *Dietrich et al., 2020a*.

For the preparation of study plants, QuickPot trays of 20 cm³ volume (Hermann Meyer KG, Rellingen, Germany) were sterilized with a potassium hypochlorite solution (Eau de Javel: 2.6 g KClO to 100 ml water; 1:1) and filled with an autoclaved mixture of sand and soil from the Jena Experiment (1:1; sterilized twice for 40 min at 121 °C) in June 2017. Each species and origin (i.e., plot) was sown with two or three seeds per pot plate cell. QuickPot trays were placed in an open greenhouse (Research Station Bad Lauchstädt, UFZ) to promote germination by natural daily temperature fluctuations. Trays were regularly watered (with demineralized water). On 29 June 2017, *A. pratensis* seeds were reseeded because of low germination rate. For the other three species, one seedling per pot plate cell was removed if more than two seeds were germinated. It should be noted that we cannot exclude possible maternal effects in our common garden experiment, because we used seed material collected in different plots of the field experiment; however, differences in maternal effects can also

be important drivers of eco-evolutionary feedbacks and can significantly influence plant responses to global change (**Puy et al., 2021**; **Rottstock et al., 2017**).

## Common garden experiment

In July 2017, 12 soil cores (5 cm diameter, 10 cm depth) were taken in a grid of 20 × 20 cm in each Dominance Experiment plot selected for the study and stored in a cooling chamber (4 °C). Soil cores were pooled per plot and sieved through a sieve with 5 mm mesh size to remove stones and coarse roots. Then, 2800 cm$^3$ steam-sterilized background substrate was thoroughly mixed with 150 cm$^3$ fresh-sieved living soil and filled in a heat-cleaned pot (3 L, diameter 14.9 cm, height 18 cm) with 12 replicates per plot. Seedlings per pot plate cell were separated, and two seedlings per species with same plot origin were transplanted into one pot (**Figure 5**). In four pots per plot, we transplanted plants, which had the same plot origin as the inoculated soil (home soil treatment); in the other eight pots, plant and soil origin were different (away soil treatment). In four of these away pots, species richness of plant and soil origin was the same, but plant species composition was different (away-same soil treatment), and in the other four away pots, species richness of plant and soil origin was different (away-different soil treatment; **Figure 5**). Seedlings of *D. glomerata* were transplanted on 18 July 2017, followed by *A. elatius* (20 July 2017), *P. trivialis* (20 and 24 July 2017), and *A. pratensis* (26–28 July 2017). Seedlings were immediately watered with 200 ml demineralized water after transplantation, and the initial number of shoots was counted. In total, the experiment consisted of 576 pots, each with two plants. The pots were placed in an open greenhouse with a roof, which automatically closes at rain, and ambient temperatures (Research Station Bad Lauchstädt, UFZ). Pots were distributed in six blocks placing the 12 pots filled with soil from one plot in one block, that is, in each block, there were 12 pots with soil of one two-species and one six-species plot per species. The position of the pots within the blocks was randomly chosen and changed once a month to avoid potential side effects by neighboring pots and edge effects of the tables.

During the first week after planting, plants were watered every day with 200 ml demineralized water. From week two to four, all pots were watered every other day with 380 ml demineralized water without further treatments to allow the establishment of plants and soil biota in the pots (380 ml were used to achieve a water saturation of the soil of 60%; calculation can be found in Appendix 3). On 23 August 2017, treatments with the global change drivers were started. For every treatment (control, drought, nitrogen input, combination of drought and nitrogen input), we used three of the 12 pots per plot (one home, one away-same, and one away-different pot, respectively; **Figure 5**).

 I. For control, pots were watered as before (380 ml; every other day) and were not fertilized.
 II. Drought was simulated by reduced water saturation ( = 30% water saturation = 225 ml; calculation can be found in Appendix 3). Pots were still watered every other day but with 225 ml instead of 380 ml demineralized water.
 III. Nitrogen input was applied once a week with 95 mg $NH_4NO_3$ (33.125 mg nitrogen) resulting in a total nitrogen amount of 265 mg after eight fertilization events, which is equivalent to a nitrogen input of 150 kg ha$^{-1}$ year$^{-1}$ nitrogen (medium value for managed grasslands in Germany; **Häußermann et al., 2019**). Fertilized plants were watered as before (380 ml; every other day).
 IV. For the combination of drought and nitrogen input, pots were watered with a reduced amount (225 ml) and were fertilized once a week (in the same way as for the nitrogen input treatment alone).

Once a month, all pots were weighted before watering. The measured weight per pot was subtracted from dry soil weight plus the assigned amount of water (380 or 225 ml). The difference revealed the amount of water which was then used to water the pot to keep the anticipated levels of water saturation for the drought and control treatment.

## Data collection

After 11 weeks of growth with global change driver treatments, plants were harvested block-wise (between 16 October and 8 November 2018). Before harvest, aboveground traits and leaf fungal pathogen infestation were measured (**Table 3**). For growth height (in cm), we measured the stretched shoot length of the longest vegetative shoot per plant. Only 15% of the plants had flowered, which was neglected due to the small case number. For leaf greenness (unitless estimate of foliar chlorophyll content), three fully expanded leaves from vegetative shoots of each plant were measured with a

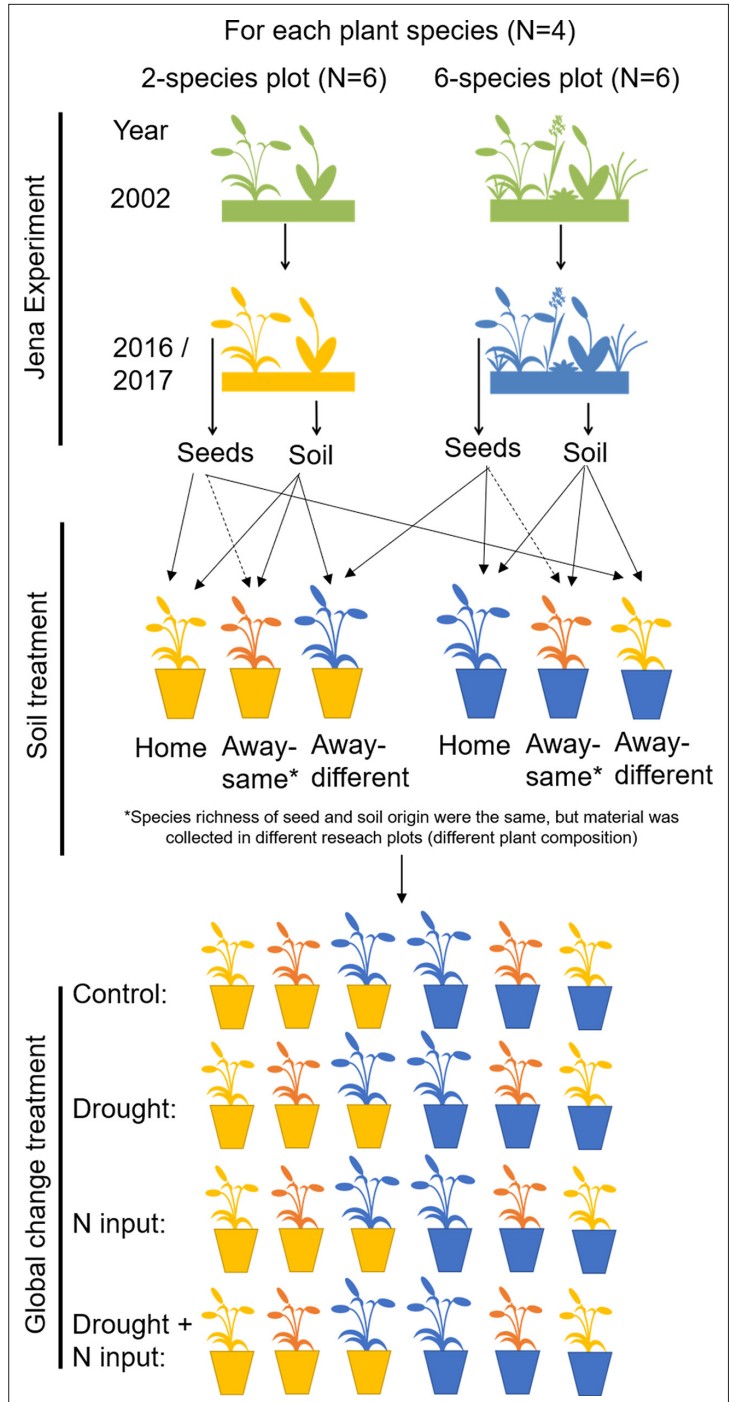

**Figure 5.** Overview of experimental design. In 2016, ripe seeds of four grass species were collected in two- and six-species plots of the Dominance Experiment (Jena Experiment), stored in a freezer and allowed to germinate in spring 2017. After germination, soil samples were collected from the plots and mixed with sterilized background soil (5% + 95%), filled in pots and planted with two seedlings (12 pot replicates per plot). In four pots per plot, plant and soil had the same plot origin (home soil); in four pots, species richness of plant and soil origin were the same, but plant species composition was different (away-same soil) and in four pots, species richness of plant and soil origin were different ( = different origin of plant and soil; away-different soil; total $Nr_{pots}$ = 576). Plants were exposed to global change drivers: drought, nitrogen input, or the combination of drought and nitrogen input, or were not treated (control).

**Table 3.** Summary list of response variables and experimental factors of the common garden experiment.

| Variable | Abbreviation | Unit | Description |
|---|---|---|---|
| **Response variables** | | | |
| Biomass production | | | |
| Total biomass | Total Bm | $g_{total}$ | Shoot and root biomass per pot |
| Shoot mass | Shoot Bm | $g_{shoot}$ | Shoot biomass per pot |
| Root mass | Root Bm | $g_{root}$ | Root biomass per pot |
| Root-shoot ratio | - | $g_{root}\ g_{shoot}^{-1}$ | Root biomass divided by shoot biomass per pot |
| Aboveground traits | | | |
| Growth height | - | cm | Stretched shoot length of longest vegetative shoot[*] |
| Shoot nitrogen concentration | $N_{Shoot}$ | $mg\ N\ g_{shoot}^{-1}$ | Nitrogen mass per dry shoot mass |
| Leaf greenness | - | - | Unitless estimate of leaf chlorophyll concentration[*] |
| Specific leaf area | SLA | $mm_{leaf}^2\ mg_{leaf}^{-1}$ | Leaf area per dry leaf mass[*] |
| Leaf dry matter content | LDMC | $mg_{leaf}\ g_{leaf}^{-1}$ | Dry leaf mass per water-saturated fresh leaf mass[*] |
| Stomatal conductance | $g_s$ | $mmol\ m^{-2}\ s^{-1}$ | Stomatal conductance per leaf area[*] |
| Belowground traits | | | |
| Root diameter | Dia | mm | Average root diameter of the root subsample |
| Specific root length | SRL | $m_{root}\ g_{root}^{-1}$ | Root length per dry root biomass (subsample) |
| Root length density | RLD | $cm_{root}\ cm_{soil}^{-3}$ | Root length (extrapolated) per soil volume (pot) |
| Pathogen infestation | - | % | Percentage of infested leaf area (estimated)[*] |
| **Experimental factors** | | | |
| Species identity | Species ID | - | Study species |
| Legacy treatments | | | |
| Plant history | PH | - | Species richness of the plant community, where the seeds were collected – two or six plant species |
| Soil history | SH | - | Species richness of plant community, where the soil for inoculation was taken – two or six plant species |
| Soil treatment | ST | - | Origin of seed and soil in one pot: <br>• same plot origin = home soil treatment <br>• different plot origin, but same species richness = away-same soil treatment <br>• different plot origin, different species richness = away-different soil treatment |
| Global change driver treatments | global change driver / global change / GC | | |
| Drought treatment | Drought / D | - | 30% instead of 60% water saturation |
| Nitrogen input treatment | Nitrogen input / N | - | Fertilization with $NH_4NO_3$ (150 kg $ha^{-1}$ $year^{-1}$ nitrogen) |

[*]averaged per pot.

SPAD 502 Plus Chlorophyll Meter (Spectrum Technologies, Inc) and values were averaged per plant. Stomatal conductance ($g_s$; $mmol\ m^{-2}\ s^{-1}$) was measured at one fully expanded leaf per plant (i.e., two leaves per pot) with a SC-1 Leaf Porometer (Decagon Devices Inc). This was done block-wise and always one day after watering, between 10 a.m. and 3 p.m. Shortly before harvest, the percentage of total leaf area, which was infested by fungal pathogens was estimated for each plant. A subsample of leaves per species was taken to identify pathogens morphologically at the species level under a light microscope. Moreover, three fully expanded leaves per individual were cut, packed in wet paper towels to achieve water saturation, and stored overnight in a cooling chamber at 4 °C. On the next day, leaves were weighed as bulk sample per pot (i.e. six leaves) after removing water droplets with

tissue paper. Total leaf area was measured with a leaf area meter (LI-3000C Area Meter equipped with LI3050C transparent conveyer belt accessory, LICOR, USA) Afterwards, leaf samples were dried at 70 °C for 48 hr and weighed. LDMC was calculated as the ratio of dry weight to fresh weight ($mg_{leaf}$ $g_{leaf}^{-1}$) and SLA as the ratio of leaf area to dry weight ($mm_{leaf}^2$ $mg_{leaf}^{-1}$).

For biomass harvest, plants were cut at ground level, and roots were cleaned by rinsing off all soil over a 0.5 mm sieve. The fresh root biomass was weighed and a subsample of around 1–2 g fresh weight was stored at –20 °C. At a later point, roots were thawed and scanned on a flatbed scanner at 800 dpi (Epson Expression 10000 XL scanner, Regent Instruments, Quebec, Canada), and root diameter and root length of the subsample were measured with an image analysis software (WinRHIZO; Regent Instruments, Quebec City, Canada). Specific root length (SRL) was calculated as the ratio of root length to root dry biomass (of the subsample; $m_{root}$ $g_{root}^{-1}$) and root length density (RLD) as the ratio of root length to soil volume in the pot (root length was extrapolated from the ratio of dry root biomass of the measured subsample to total dry root mass per pot; $cm_{root}$ $cm_{soil}^{-3}$).

All biomass samples were dried at 70 °C for 48 hr and then weighed. To calculate total shoot biomass per pot (each with two individuals), dry shoot biomass and dry leaf mass of the sample used for leaf area measurements were added. To calculate total root biomass, dry biomass of the scanned subsample was extrapolated from the ratio of fresh root biomass to dry root biomass per pot and added to the weighed dry root biomass per pot.

For chemical analysis, shoot biomass of each pot was chopped, and a subsample was ground with a ball mill. Then, 10 mg milled material was used to determine shoot nitrogen concentration with near-infrared spectroscopy (MPA Multi Purpose FT-NIR Analyzer, Bruker GmbH, Ettlingen, Germany). The calibration models used to predict shoot nitrogen concentrations were derived from laboratory data generated from previous samples of grass species. The accuracy of the predictions was verified by a repeated nitrogen concentration analysis of 45 randomly selected samples with an elemental analyzer (Vario EL Element Analyzer, Elementar, Hanau, Germany). Significant positive correlation ($p < 0.001$, $r = 0.97$, N = 45) between concentrations resulted from near-infrared spectroscopy and analysis with the elemental analyzer demonstrate high accuracy of our predictions.

## Data analysis

To test whether the plants performed differently depending on legacy treatments (plant history, soil history, soil treatment [home/away]), or type of global change treatment, linear mixed-effects models were fitted for all measured response variables per pot as summarized in *Table 3*. Furthermore, some variables were transformed to meet the assumptions of normality and variance homogeneity: if necessary, root biomass and RLD were square root-transformed and root-shoot ratio, SLA, stomatal conductance, SRL, and pathogen infestation were log-transformed. Furthermore, outlier values of LDMC of three *P. trivialis* pots (extremely low values), and LDMC and SLA of one *A. elatius* pot (extremely low LDMC, high SLA) were excluded from the analysis.

For mixed-effect model analysis, we started with a null model with the random effects only (fitted with maximum likelihood). We used seed plot identity (plot, where the seeds had been collected) and soil plot identity (plot, from which the inoculation soil had been taken) as random effects. Then, we successively added the fixed effects with species identity first, followed by the legacy treatments: plant history (species richness of the plant community, where the seeds had been collected: two or six), soil history (species richness of the plant community, where the soil for inoculation had been taken: two or six), and soil treatment (home, away-same, away-different), followed by the global change driver treatments: drought (control or drought) and nitrogen input (control or nitrogen), and finally all interactions between species identity and the other fixed effects to check whether species differ in their responses. For analysis of stomatal conductance, we used daytime and air temperature as covariates, which were entered before adding the experimental factors to account for possible effects of the measurement time. Moreover, to decompose the variability attributable to model terms, mixed-effect models (for biomass production) were fitted, but with the restricted maximum likelihood method. Then, the share of explained variability was estimated as the difference between total variability attributed to random effects in models not including, and models including, the respective fixed effect ( = variance decomposition; *Siebenkäs et al., 2015*). Because of multiple significant interactions between species identity and other fixed effects (*Table 1*), we further analyzed the response variables separately per species. Therefore,

we used the same fixed effect structure as explained above, but without species identity and additionally with the interactions between legacy treatments and global change driver treatments (which was not done in the first model, because otherwise, it would have become too complex). For pathogen infestation, we only analyzed data of *D. glomerata* and *P. trivialis*, because of very low infestation rates of *A. elatius* and *A. pratensis* plants. To test whether initial size influenced the performance of the phytometers later in the experiment, we added initial shoot number as a fixed effect before the other fixed effects in separate models for analysis of shoot and root biomass production.

Because of multiple significant interactions between legacy treatments and global change driver treatments (*Appendix 2—Tables 1–10*), we further analyzed the data for each global change driver treatment separately. We used plant history, soil history, and soil treatment as fixed effects for species-specific analysis, and for analyses across all four species, we extended the models by fitting species identity first and all possible interactions between species identity and legacy treatments in the end.

All models were fitted with maximum likelihood (ML), and likelihood ratio tests were used to decide on the significance of the fixed effects. Tukey's HSD test was used to test differences among soil treatment groups. All calculations and statistical analyses were done in R (version 3.6.1, R Development Core Team, http://www.R-project.org) including the package *lme4* (glmer and lmer; *Bates et al., 2015*) and *multcomp* (Tukey HSD; *Hothorn et al., 2008*) for mixed-effects model analysis.

## Acknowledgements

We thank the technicians of the experimental research station Bad Lauchstädt, who helped with the establishment, maintenance and harvest of the experiment. Furthermore, we are grateful to Laura Bergmann for help during the experiment, and Peter Otto for identifying the fungal pathogens. The Jena Experiment is funded by the German Research Foundation (FOR 1451, FOR 5000), supported by the University of Jena, the Max Planck Institute for Biogeochemistry and the German Centre for Integrative Biodiversity Research (iDiv) Halle-Jena-Leipzig, funded by the German Research Foundation (FZT 118). Further support was provided by the Heinrich Böll Foundation (Ph.D. scholarship to PD).

## Additional information

### Funding

| Funder | Grant reference number | Author |
| --- | --- | --- |
| Heinrich Böll Stiftung | Ph.D. scholarship | Peter Dietrich |
| Deutsche Forschungsgemeinschaft | FOR 5000 | Nico Eisenhauer Christiane Roscher |
| Deutsche Forschungsgemeinschaft | FZT 118 | Nico Eisenhauer Christiane Roscher |
| Deutsche Forschungsgemeinschaft | FOR 1451 | Nico Eisenhauer Christiane Roscher |

The funders had no role in study design, data collection and interpretation, or the decision to submit the work for publication.

### Author contributions

Peter Dietrich, Conceptualization, Data curation, Formal analysis, Funding acquisition, Investigation, Methodology, Visualization, Writing - original draft; Jens Schumacher, Conceptualization, Formal analysis, Writing - review and editing; Nico Eisenhauer, Funding acquisition, Supervision, Writing - review and editing; Christiane Roscher, Conceptualization, Data curation, Funding acquisition, Methodology, Project administration, Supervision, Writing - review and editing

### Author ORCIDs

Peter Dietrich http://orcid.org/0000-0002-7742-6064
Nico Eisenhauer http://orcid.org/0000-0002-0371-6720

Decision letter and Author response
Decision letter https://doi.org/10.7554/eLife.74054.sa1
Author response https://doi.org/10.7554/eLife.74054.sa2

## Additional files

### Supplementary files
• Transparent reporting form

### Data availability
The data reported in this paper have been deposited in Dryad, which can be publicly accessed at https://doi.org/10.5061/dryad.gmsbcc2p7.

The following dataset was generated:

| Author(s) | Year | Dataset title | Dataset URL | Database and Identifier |
|---|---|---|---|---|
| Dietrich P, Schumacher J, Eisenhauer N, Roscher C | 2022 | Eco-evolutionary feedbacks modulate plant responses to global change depending on plant diversity and species identity | https://doi.org/10.5061/dryad.gmsbcc2p7 | Dryad Digital Repository, 10.5061/dryad.gmsbcc2p7 |

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

## Appendix 1

### Hypothesis 3: offspring of plants selected at different diversity and grown in different soil (high vs. low diversity, home vs. away) respond differently to global change drivers

Plant traits and pathogen infestation (across species and for each species)

Growth height did not differ depending on soil or plant history when plants were treated with global change drivers across all study species and for *D. glomerata* (Table S1, S4). Plants of *A. elatius* in home soil were smaller than plants in away-same soil (Table S2). Nitrogen input had no influence, while plants were tallest in home soil and smallest in away-different soil when treated with both global change drivers (Table S2). Plants of *A. pratensis* exposed to drought were taller when grown in home than in away-different soil; however, this positive home effect was also only found in control plants (marginal significant; Table S3). When fertilized, the positive home effect on growth height disappeared (Table S3). Plants of *P. trivialis* were taller in two- than in six-species community soil when treated with both global change drivers, but they were not different when treated separately with drought or nitrogen input (Table S5).

Leaf greenness and shoot nitrogen concentrations were not influenced by legacy treatments when exposed to drought. When fertilized, plants still did not differ in leaf greenness but had higher shoot nitrogen concentrations in six-species than in two-species soil, found across all study species and for *D. glomerata* (Table S1, S4). Moreover, fertilized plants had a lower shoot nitrogen concentration in home than in away-different soil, found across all species and for *A. pratensis* (Table S1, S3). When plants were treated with both global change drivers, the nitrogen input effect on soil history was cancelled out by drought (across all species and for *D. glomerata*), while the impact of soil treatment did not: plants in home soil still had lower shoot nitrogen concentration than plants in away soil (across all species and for *A. pratensis*).

Plants treated with global change drivers did not differ significantly in LDMC or SLA dependent on legacy treatments, across all study species and in *A. elatius* (Table S1, S2). Drought resulted in higher LDMC of *A. pratensis* plants grown in six-species soil, and the combined application of drought and nitrogen input resulted in lower SLA in home than in away soil (Table S3). Fertilized *D. glomerata* plants had higher SLA in six- than in two-species community soil (Table S4). Plants of *P. trivialis* treated with both global change drivers had lower LDMC in two- than in six-species community soil (Table S5).

Stomatal conductance ($g_s$) did not differ significantly depending on legacy treatments when plants were treated with global change drivers across all study species and for *A. elatius* and *P. trivialis* (Table S1, S2, S5). In *A. pratensis*, fertilized plants showed a lower $g_s$ when grown in home than in away soil. This effect was cancelled out by drought, when treated with both global change drivers (Table S3). In *D. glomerata*, plants had higher $g_s$ when originated from six-species communities and treated with both global change drivers; however, this was also found in control plants (Table S4).

Across all study species, root diameter, SRL and RLD were not influenced by legacy treatments when treated with global change drivers (Table S1). In *A. elatius*, root traits also did not differ, when treated with single global change drivers, but under the combined influence of both global change drivers, plants grown in away-different soil showed the highest SRL, and plants in away-same soil had the lowest SRL (Table S2). In *A. pratensis*, plants exposed to drought had higher SRL and RLD in two- than in six-species soil. When fertilized, we did not find an effect of legacy treatment, but the combination of both global change drivers led to higher SRL and lower root diameter when plants were grown in away-same than in away-different or home soil (Table S3). In *D. glomerata*, RLD of plants exposed to drought was higher when originated from six-species than from two-species communities. This positive diversity impact disappeared when fertilized (Table S4). In *P. trivialis*, SRL were lower in plants grown in six-species community soil, when exposed to drought. When fertilized, this difference disappeared (Table S5).

Mildew infestation of *D. glomerata* plants exposed to drought was higher in home than in away soil, while this drought effect was cancelled out by nitrogen input (Table S6). Mildew infestation of *P. trivialis* plants was not significantly influenced by plant or soil history, neither with nor without global change drivers (Table S6).

**Appendix 1—table 1.** Summary of mixed-effect model analyses testing the effects of legacy treatments (plant history, soil history, soil treatment) on plant performance (total biomass, shoot biomass, root biomass and root-shoot ratio) of *A. elatius*, when non-treated (control) or treated with GC drivers (drought, nitrogen input, drought and nitrogen input (D x N)).

Shown are degrees of freedom (Df), Chi² and p-values (p). Significant effects (p < 0.05) are given in bold, marginally significant effects (p < 0.1) in italics.

| | Total biomass | | | | | | | | | | | |
| --- | --- | --- | --- | --- | --- | --- | --- | --- | --- | --- | --- | --- |
| | Control | | | Drought | | | Nitrogen | | | D x N | | |
| *A. elatius* | Df | Chi² | p | Df | Chi² | p | Df | Chi² | p | Df | Chi² | p |
| Plant history | 1 | 0.35 | 0.557 | 1 | 0.82 | 0.364 | 1 | 0.71 | 0.401 | 1 | 0.26 | 0.613 |
| Soil history | 1 | 1.08 | 0.298 | 1 | 0.76 | 0.383 | 1 | 0.06 | 0.811 | 1 | 0.47 | 0.494 |
| Soil treatment | 2 | 0.10 | 0.949 | 2 | 2.91 | 0.233 | 2 | 6.44 | **0.040** | 2 | 0.99 | 0.610 |

| | Shoot biomass | | | | | | | | | | | |
| --- | --- | --- | --- | --- | --- | --- | --- | --- | --- | --- | --- | --- |
| | Control | | | Drought | | | Nitrogen | | | D x N | | |
| | Df | Chi² | p | Df | Chi² | p | Df | Chi² | p | Df | Chi² | p |
| Plant history | 1 | 0.12 | 0.726 | 1 | 3.36 | *0.067* | 1 | 1.27 | 0.260 | 1 | 0.01 | 0.904 |
| Soil history | 1 | 0.35 | 0.556 | 1 | 0.24 | 0.621 | 1 | 0.55 | 0.460 | 1 | 0.63 | 0.428 |
| Soil treatment | 2 | 2.08 | 0.354 | 2 | 2.89 | 0.236 | 2 | 5.24 | *0.073* | 2 | 0.98 | 0.613 |

| | Root biomass | | | | | | | | | | | |
| --- | --- | --- | --- | --- | --- | --- | --- | --- | --- | --- | --- | --- |
| | Control | | | Drought | | | Nitrogen | | | D x N | | |
| | Df | Chi² | p | Df | Chi² | p | Df | Chi² | p | Df | Chi² | p |
| Plant history | 1 | 0.03 | 0.860 | 1 | 0.15 | 0.701 | 1 | 0.01 | 0.916 | 1 | 0.36 | 0.551 |
| Soil history | 1 | 3.81 | *0.051* | 1 | 0.62 | 0.433 | 1 | 0.17 | 0.676 | 1 | 0.22 | 0.636 |
| Soil treatment | 2 | 2.05 | 0.359 | 2 | 2.38 | 0.304 | 2 | 2.25 | 0.325 | 2 | 1.68 | 0.432 |

| | Root-shoot ratio | | | | | | | | | | | |
| --- | --- | --- | --- | --- | --- | --- | --- | --- | --- | --- | --- | --- |
| | Control | | | Drought | | | Nitrogen | | | D x N | | |
| | Df | Chi² | p | Df | Chi² | p | Df | Chi² | p | Df | Chi² | p |
| Plant history | 1 | 0.07 | 0.797 | 1 | 1.62 | 0.203 | 1 | 0.16 | 0.691 | 1 | 0.31 | 0.576 |
| Soil history | 1 | 4.86 | **0.027** | 1 | 0.24 | 0.626 | 1 | 0.50 | 0.479 | 1 | 0.07 | 0.787 |
| Soil treatment | 2 | 3.11 | 0.211 | 2 | 2.39 | 0.302 | 2 | 0.18 | 0.915 | 2 | 1.88 | 0.391 |

**Appendix 1—table 2.** Summary of mixed-effect model analyses testing the effects of legacy treatments (plant history, soil history, soil treatment) on plant performance (total biomass, shoot biomass, root biomass and root-shoot ratio) of *A. pratensis*, when non-treated (control) or treated with GC drivers (drought, nitrogen input, drought and nitrogen input (D x N)).

Shown are degrees of freedom (Df), Chi² and p-values (p). Significant effects (p < 0.05) are given in bold, marginally significant effects (p < 0.1) in italics.

| | Total biomass | | | | | | | | | | | |
| --- | --- | --- | --- | --- | --- | --- | --- | --- | --- | --- | --- | --- |
| | Control | | | Drought | | | Nitrogen | | | D x N | | |
| *A. pratensis* | Df | Chi² | p | Df | Chi² | p | Df | Chi² | p | Df | Chi² | p |
| Plant history | 1 | 0.05 | 0.820 | 1 | 0.27 | 0.603 | 1 | 1.63 | 0.202 | 1 | 1.44 | 0.230 |
| Soil history | 1 | 0.02 | 0.879 | 1 | 1.05 | 0.306 | 1 | 2.97 | *0.085* | 1 | 2.07 | 0.151 |
| Soil treatment | 2 | 3.43 | 0.180 | 2 | 0.17 | 0.917 | 2 | 1.29 | 0.525 | 2 | 2.80 | 0.247 |

*Continued on next page*

| **Shoot biomass** | | | | | | | | | | | |
| | **Control** | | | **Drought** | | | **Nitrogen** | | | **D x N** | | |
| | Df | Chi² | p | Df | Chi² | p | Df | Chi² | p | Df | Chi² | p |
|---|---|---|---|---|---|---|---|---|---|---|---|---|
| Plant history | 1 | 0.11 | 0.741 | 1 | 0.29 | 0.590 | 1 | 0.65 | 0.421 | 1 | 2.23 | 0.135 |
| Soil history | 1 | 0.14 | 0.710 | 1 | 0.33 | 0.564 | 1 | 0.86 | 0.354 | 1 | <0.01 | 0.971 |
| Soil treatment | 2 | 0.15 | 0.927 | 2 | 1.84 | 0.398 | 2 | 1.03 | 0.596 | 2 | 1.35 | 0.509 |

| **Root biomass** | | | | | | | | | | | |
| | **Control** | | | **Drought** | | | **Nitrogen** | | | **D x N** | | |
| | Df | Chi² | p | Df | Chi² | p | Df | Chi² | p | Df | Chi² | p |
|---|---|---|---|---|---|---|---|---|---|---|---|---|
| Plant history | 1 | 0.13 | 0.719 | 1 | 0.23 | 0.629 | 1 | 0.97 | 0.324 | 1 | 0.47 | 0.495 |
| Soil history | 1 | 0.15 | 0.703 | 1 | 1.16 | 0.281 | 1 | 1.83 | 0.176 | 1 | 3.98 | **0.046** |
| Soil treatment | 2 | 2.78 | 0.250 | 2 | 1.38 | 0.501 | 2 | 0.47 | 0.789 | 2 | 3.16 | 0.206 |

| **Root-shoot ratio** | | | | | | | | | | | |
| | **Control** | | | **Drought** | | | **Nitrogen** | | | **D x N** | | |
| | Df | Chi² | p | Df | Chi² | p | Df | Chi² | p | Df | Chi² | p |
|---|---|---|---|---|---|---|---|---|---|---|---|---|
| Plant history | 1 | 0.13 | 0.719 | 1 | 0.01 | 0.920 | 1 | 0.30 | 0.584 | 1 | 0.90 | 0.342 |
| Soil history | 1 | 0.20 | 0.654 | 1 | 0.31 | 0.579 | 1 | 0.42 | 0.517 | 1 | 4.57 | **0.033** |
| Soil treatment | 2 | 1.33 | 0.514 | 2 | 4.94 | *0.084* | 2 | 0.04 | 0.982 | 2 | 0.37 | 0.832 |

**Appendix 1—table 3.** Summary of mixed-effect model analyses testing the effects of legacy treatments (plant history, soil history, soil treatment) on plant performance (total biomass, shoot biomass, root biomass and root-shoot ratio) of *D. glomerata*, when non-treated (control) or treated with GC drivers (drought, nitrogen input, drought and nitrogen input (D x N)).

Shown are degrees of freedom (Df), Chi² and p-values (p). Significant effects (p < 0.05) are given in bold, marginally significant effects (p < 0.1) in italics.

| **Total biomass** | | | | | | | | | | | |
| | **Control** | | | **Drought** | | | **Nitrogen** | | | **D x N** | | |
| *D. glomerata* | Df | Chi² | p | Df | Chi² | p | Df | Chi² | p | Df | Chi² | p |
|---|---|---|---|---|---|---|---|---|---|---|---|---|
| Plant history | 1 | 0.56 | 0.456 | 1 | 2.22 | 0.136 | 1 | 3.09 | *0.079* | 1 | 0.13 | 0.715 |
| Soil history | 1 | 6.28 | **0.012** | 1 | 0.76 | 0.384 | 1 | 0.73 | 0.394 | 1 | <0.01 | 0.978 |
| Soil treatment | 2 | 1.52 | 0.467 | 2 | 0.94 | 0.626 | 2 | 1.26 | 0.533 | 2 | 0.73 | 0.693 |

| **Shoot biomass** | | | | | | | | | | | |
| | **Control** | | | **Drought** | | | **Nitrogen** | | | **D x N** | | |
| | Df | Chi² | p | Df | Chi² | p | Df | Chi² | p | Df | Chi² | p |
|---|---|---|---|---|---|---|---|---|---|---|---|---|
| Plant history | 1 | 0.02 | 0.885 | 1 | 1.28 | 0.259 | 1 | 3.18 | *0.075* | 1 | 0.22 | 0.640 |
| Soil history | 1 | 8.27 | **0.004** | 1 | 0.81 | 0.369 | 1 | 0.33 | 0.567 | 1 | 0.15 | 0.700 |
| Soil treatment | 2 | 3.06 | 0.216 | 2 | 0.44 | 0.801 | 2 | 3.34 | 0.188 | 2 | 0.14 | 0.932 |

| **Root biomass** | | | | | | | | | | | |
| | **Control** | | | **Drought** | | | **Nitrogen** | | | **D x N** | | |
| | Df | Chi² | p | Df | Chi² | p | Df | Chi² | p | Df | Chi² | p |
|---|---|---|---|---|---|---|---|---|---|---|---|---|
| Plant history | 1 | 1.40 | 0.236 | 1 | 2.55 | 0.111 | 1 | 1.98 | 0.160 | 1 | 0.04 | 0.848 |
| Soil history | 1 | 0.90 | 0.343 | 1 | 0.45 | 0.501 | 1 | 0.99 | 0.319 | 1 | 0.21 | 0.644 |
| Soil treatment | 2 | 2.49 | 0.288 | 2 | 2.06 | 0.358 | 2 | 0.02 | 0.992 | 2 | 3.16 | 0.206 |

| | Root-shoot ratio | | | | | | | | | | | |
| --- | --- | --- | --- | --- | --- | --- | --- | --- | --- | --- | --- | --- |
| | **Control** | | | **Drought** | | | **Nitrogen** | | | **D x N** | | |
| | Df | Chi² | p | Df | Chi² | p | Df | Chi² | p | Df | Chi² | p |
| Plant history | 1 | 1.65 | 0.199 | 1 | 1.71 | 0.191 | 1 | 0.93 | 0.335 | 1 | 0.01 | 0.936 |
| Soil history | 1 | <0.01 | 0.983 | 1 | 0.44 | 0.505 | 1 | 0.43 | 0.514 | 1 | 0.75 | 0.387 |
| Soil treatment | 2 | 3.14 | 0.208 | 2 | 2.84 | 0.242 | 2 | 0.20 | 0.906 | 2 | 7.72 | **0.021** |

**Appendix 1—table 4.** Summary of mixed-effect model analyses testing the effects of legacy treatments (plant history, soil history, soil treatment) on plant performance (total biomass, shoot biomass, root biomass and root-shoot ratio) of *P. trivialis*, when non-treated (control) or treated with GC drivers (drought, nitrogen input, drought and nitrogen input (D x N)).

Shown are degrees of freedom (Df), Chi² and p-values (p). Significant effects (p < 0.05) are given in bold, marginally significant effects (p < 0.1) in italics.

| | Total biomass | | | | | | | | | | | |
| --- | --- | --- | --- | --- | --- | --- | --- | --- | --- | --- | --- | --- |
| | **Control** | | | **Drought** | | | **Nitrogen** | | | **D x N** | | |
| *P. trivialis* | Df | Chi² | p | Df | Chi² | p | Df | Chi² | p | Df | Chi² | p |
| Plant history | 1 | 0.12 | 0.732 | 1 | 1.25 | 0.264 | 1 | 0.28 | 0.599 | 1 | 0.43 | 0.513 |
| Soil history | 1 | 0.12 | 0.731 | 1 | 0.14 | 0.704 | 1 | 0.07 | 0.796 | 1 | 0.05 | 0.826 |
| Soil treatment | 2 | 0.01 | 0.995 | 2 | 1.82 | 0.404 | 2 | 1.69 | 0.430 | 2 | 4.06 | 0.131 |

| | Shoot biomass | | | | | | | | | | | |
| --- | --- | --- | --- | --- | --- | --- | --- | --- | --- | --- | --- | --- |
| | **Control** | | | **Drought** | | | **Nitrogen** | | | **D x N** | | |
| | Df | Chi² | p | Df | Chi² | p | Df | Chi² | p | Df | Chi² | p |
| Plant history | 1 | 0.01 | 0.920 | 1 | 1.91 | 0.167 | 1 | 0.39 | 0.532 | 1 | 0.01 | 0.943 |
| Soil history | 1 | <0.01 | 0.973 | 1 | 0.47 | 0.492 | 1 | 0.46 | 0.499 | 1 | 0.19 | 0.663 |
| Soil treatment | 2 | 1.34 | 0.511 | 2 | 0.81 | 0.667 | 2 | 1.22 | 0.545 | 2 | 2.96 | 0.227 |

| | Root biomass | | | | | | | | | | | |
| --- | --- | --- | --- | --- | --- | --- | --- | --- | --- | --- | --- | --- |
| | **Control** | | | **Drought** | | | **Nitrogen** | | | **D x N** | | |
| | Df | Chi² | p | Df | Chi² | p | Df | Chi² | p | Df | Chi² | p |
| Plant history | 1 | 0.21 | 0.647 | 1 | 0.66 | 0.417 | 1 | 1.48 | 0.224 | 1 | 1.45 | 0.229 |
| Soil history | 1 | 0.33 | 0.566 | 1 | 1.24 | 0.266 | 1 | 0.74 | 0.389 | 1 | 0.03 | 0.870 |
| Soil treatment | 2 | 1.36 | 0.506 | 2 | 1.10 | 0.577 | 2 | 1.99 | 0.370 | 2 | 5.03 | *0.081* |

| | Root-shoot ratio | | | | | | | | | | | |
| --- | --- | --- | --- | --- | --- | --- | --- | --- | --- | --- | --- | --- |
| | **Control** | | | **Drought** | | | **Nitrogen** | | | **D x N** | | |
| | Df | Chi² | p | Df | Chi² | p | Df | Chi² | p | Df | Chi² | p |
| Plant history | 1 | 0.23 | 0.630 | 1 | 0.14 | 0.708 | 1 | 2.00 | 0.158 | 1 | 2.25 | 0.134 |
| Soil history | 1 | 0.23 | 0.630 | 1 | 3.19 | *0.074* | 1 | 1.57 | 0.211 | 1 | 0.15 | 0.697 |
| Soil treatment | 2 | 3.61 | 0.164 | 2 | 0.68 | 0.711 | 2 | 2.16 | 0.340 | 2 | 5.12 | *0.077* |

**Appendix 1—table 5.** Summary of mixed-effect model analyses testing the effects of species identity, legacy treatments (plant history, soil history, soil treatment) and their interactions on plant trait expressions, when non-treated (control) or treated with GC drivers (drought, nitrogen input, drought and nitrogen input (D x N)).

Shown are degrees of freedom (Df), Chi² and p-values (p). Significant effects (p < 0.05) are given in bold, marginally significant effects (p < 0.1) in italics.

**Growth height**

| | Control | | | Drought | | | Nitrogen | | | D x N | | |
|---|---|---|---|---|---|---|---|---|---|---|---|---|
| | Df | Chi² | p | Df | Chi² | p | Df | Chi² | p | Df | Chi² | p |
| Species ID | 3 | 36.51 | <0.001 | 3 | 46.47 | <0.001 | 3 | 26.45 | <0.001 | 3 | 53.85 | <0.001 |
| Plant history (PH) | 1 | 1.76 | 0.185 | 1 | 1.08 | 0.299 | 1 | 0.06 | 0.812 | 1 | 0.75 | 0.387 |
| Soil history (SH) | 1 | 0.48 | 0.488 | 1 | 0.86 | 0.354 | 1 | 1.52 | 0.217 | 1 | 1.40 | 0.237 |
| Soil treatment (ST) | 2 | 3.99 | 0.136 | 2 | 5.49 | *0.064* | 2 | 2.68 | 0.262 | 2 | 4.37 | 0.113 |
| Species ID x PH | 3 | 4.12 | 0.249 | 3 | 4.53 | 0.210 | 3 | 2.62 | 0.455 | 3 | 0.17 | 0.982 |
| Species ID x SH | 3 | 3.65 | 0.301 | 3 | 1.16 | 0.762 | 3 | 1.14 | 0.766 | 3 | 6.66 | *0.084* |
| Species ID x ST | 6 | 8.19 | 0.224 | 6 | 13.52 | **0.035** | 6 | 6.01 | 0.423 | 6 | 7.18 | 0.305 |

**Shoot nitrogen concentration**

| | Control | | | Drought | | | Nitrogen | | | D x N | | |
|---|---|---|---|---|---|---|---|---|---|---|---|---|
| | Df | Chi² | p | Df | Chi² | p | Df | Chi² | p | Df | Chi² | p |
| Species ID | 3 | 49.63 | <0.001 | 3 | 23.08 | <0.001 | 3 | 73.52 | <0.001 | 3 | 30.02 | <0.001 |
| Plant history (PH) | 1 | 0.94 | 0.333 | 1 | 0.08 | 0.775 | 1 | 0.50 | 0.480 | 1 | 0.03 | 0.871 |
| Soil history (SH) | 1 | <0.01 | 0.963 | 1 | 1.50 | 0.221 | 1 | 4.67 | **0.031** | 1 | <0.01 | 0.953 |
| Soil treatment (ST) | 2 | 2.94 | 0.230 | 2 | 1.32 | 0.517 | 2 | 7.52 | **0.023** | 2 | 8.53 | **0.014** |
| Species ID x PH | 3 | 2.80 | 0.424 | 3 | 5.03 | 0.170 | 3 | 4.00 | 0.262 | 3 | 2.20 | 0.533 |
| Species ID x SH | 3 | 1.14 | 0.767 | 3 | 2.99 | 0.392 | 3 | 7.02 | *0.071* | 3 | 0.31 | 0.958 |
| Species ID x ST | 6 | 12.36 | *0.054* | 6 | 6.88 | 0.332 | 6 | 6.13 | 0.409 | 6 | 4.73 | 0.579 |

**Leaf greenness**

| | Control | | | Drought | | | Nitrogen | | | D x N | | |
|---|---|---|---|---|---|---|---|---|---|---|---|---|
| | Df | Chi² | p | Df | Chi² | p | Df | Chi² | p | Df | Chi² | p |
| Species ID | 3 | 45.88 | <0.001 | 3 | 44.96 | <0.001 | 3 | 54.85 | <0.001 | 3 | 71.04 | <0.001 |
| Plant history (PH) | 1 | 1.61 | 0.204 | 1 | 0.11 | 0.740 | 1 | 0.43 | 0.514 | 1 | 0.02 | 0.876 |
| Soil history (SH) | 1 | 0.18 | 0.675 | 1 | 1.84 | 0.175 | 1 | 1.04 | 0.308 | 1 | 0.11 | 0.738 |
| Soil treatment (ST) | 2 | 2.10 | 0.350 | 2 | 1.62 | 0.444 | 2 | 0.41 | 0.813 | 2 | 1.62 | 0.445 |
| Species ID x PH | 3 | 4.39 | 0.222 | 3 | 3.98 | 0.264 | 3 | 1.88 | 0.600 | 3 | 2.78 | 0.427 |
| Species ID x SH | 3 | 4.45 | 0.216 | 3 | 3.44 | 0.329 | 3 | 0.89 | 0.829 | 3 | 0.35 | 0.950 |
| Species ID x ST | 6 | 3.54 | 0.739 | 6 | 3.92 | 0.688 | 6 | 8.79 | 0.186 | 6 | 3.38 | 0.759 |

**LDMC**

| | Control | | | Drought | | | Nitrogen | | | D x N | | |
|---|---|---|---|---|---|---|---|---|---|---|---|---|
| | Df | Chi² | p | Df | Chi² | p | Df | Chi² | p | Df | Chi² | p |
| Species ID | 3 | 32.76 | <0.001 | 3 | 22.47 | <0.001 | 3 | 78.30 | <0.001 | 3 | 43.04 | <0.001 |
| Plant history (PH) | 1 | 0.33 | 0.565 | 1 | 2.01 | 0.156 | 1 | 0.03 | 0.861 | 1 | 0.03 | 0.870 |
| Soil history (SH) | 1 | 0.02 | 0.887 | 1 | 0.56 | 0.456 | 1 | 0.06 | 0.808 | 1 | 0.17 | 0.680 |
| Soil treatment (ST) | 2 | 2.83 | 0.243 | 2 | 1.27 | 0.529 | 2 | 1.34 | 0.511 | 2 | 0.80 | 0.670 |
| Species ID x PH | 3 | 1.71 | 0.635 | 3 | 0.26 | 0.967 | 3 | 1.00 | 0.802 | 3 | 4.79 | 0.188 |
| Species ID x SH | 3 | 1.69 | 0.638 | 3 | 4.04 | 0.257 | 3 | 5.48 | 0.140 | 3 | 2.91 | 0.405 |
| Species ID x ST | 6 | 3.52 | 0.742 | 6 | 1.10 | 0.981 | 6 | 5.73 | 0.454 | 6 | 11.22 | *0.082* |

**SLA**

| | Control | | | Drought | | | Nitrogen | | | D x N | | |
|---|---|---|---|---|---|---|---|---|---|---|---|---|
| | Df | Chi² | p | Df | Chi² | p | Df | Chi² | p | Df | Chi² | p |
| Species ID | 3 | 86.36 | <0.001 | 3 | 57.20 | <0.001 | 3 | 101.71 | <0.001 | 3 | 73.53 | <0.001 |
| Plant history (PH) | 1 | 0.19 | 0.661 | 1 | 0.39 | 0.530 | 1 | 1.55 | 0.214 | 1 | 0.33 | 0.567 |

*Continued on next page*

*Continued*

**SLA**

| | Control | | | Drought | | | Nitrogen | | | D x N | | |
|---|---|---|---|---|---|---|---|---|---|---|---|---|
| | Df | Chi² | p | Df | Chi² | p | Df | Chi² | p | Df | Chi² | p |
| Soil history (SH) | 1 | 0.64 | 0.425 | 1 | 0.01 | 0.926 | 1 | 3.35 | *0.067* | 1 | 0.26 | 0.607 |
| Soil treatment (ST) | 2 | 4.38 | 0.112 | 2 | 1.43 | 0.488 | 2 | 2.32 | 0.313 | 2 | 1.50 | 0.472 |
| Species ID x PH | 3 | 1.58 | 0.663 | 3 | 1.26 | 0.738 | 3 | 0.96 | 0.810 | 3 | 4.38 | 0.223 |
| Species ID x SH | 3 | 2.26 | 0.521 | 3 | 1.47 | 0.690 | 3 | 3.69 | 0.297 | 3 | 1.90 | 0.592 |
| Species ID x ST | 6 | 2.38 | 0.882 | 6 | 2.88 | 0.824 | 6 | 4.08 | 0.666 | 6 | 14.22 | **0.027** |

**Stomatal conductance**

| | Control | | | Drought | | | Nitrogen | | | D x N | | |
|---|---|---|---|---|---|---|---|---|---|---|---|---|
| | Df | Chi² | p | Df | Chi² | p | Df | Chi² | p | Df | Chi² | p |
| Temperature | 1 | 0.75 | 0.388 | 1 | 1.40 | 0.237 | 1 | 3.18 | *0.074* | 1 | 0.18 | 0.670 |
| Daytime | 1 | 18.95 | <0.001 | 1 | 13.20 | <0.001 | 1 | 5.72 | <0.001 | 1 | 16.06 | <0.001 |
| Species ID | 3 | 45.36 | <0.001 | 3 | 24.61 | <0.001 | 3 | 42.88 | <0.001 | 3 | 21.71 | <0.001 |
| Plant history (PH) | 1 | 0.60 | 0.438 | 1 | 0.01 | 0.910 | 1 | 0.48 | 0.490 | 1 | 2.95 | *0.086* |
| Soil history (SH) | 1 | 0.10 | 0.757 | 1 | 0.05 | 0.818 | 1 | 1.15 | 0.283 | 1 | 0.07 | 0.797 |
| Soil treatment (ST) | 2 | 0.08 | 0.963 | 2 | 2.67 | 0.263 | 2 | 4.85 | *0.088* | 2 | 0.20 | 0.905 |
| Species ID x PH | 3 | 4.59 | 0.204 | 3 | 3.18 | 0.365 | 3 | 4.89 | 0.180 | 3 | 4.89 | 0.180 |
| Species ID x SH | 3 | 2.60 | 0.457 | 3 | 3.53 | 0.317 | 3 | 3.23 | 0.358 | 3 | 3.36 | 0.340 |
| Species ID x ST | 6 | 8.36 | 0.213 | 6 | 4.47 | 0.614 | 6 | 3.82 | 0.701 | 6 | 4.76 | 0.575 |

**Root diameter**

| | Control | | | Drought | | | Nitrogen | | | D x N | | |
|---|---|---|---|---|---|---|---|---|---|---|---|---|
| | Df | Chi² | p | Df | Chi² | p | Df | Chi² | p | Df | Chi² | p |
| Species ID | 3 | 97.02 | <0.001 | 3 | 103.81 | <0.001 | 3 | 93.37 | <0.001 | 3 | 106.66 | <0.001 |
| Plant history (PH) | 1 | 0.87 | 0.352 | 1 | 0.02 | 0.883 | 1 | <0.01 | 0.951 | 1 | 0.08 | 0.775 |
| Soil history (SH) | 1 | 0.17 | 0.680 | 1 | 0.22 | 0.643 | 1 | 1.41 | 0.235 | 1 | 0.03 | 0.873 |
| Soil treatment (ST) | 2 | 2.42 | 0.298 | 2 | 0.93 | 0.629 | 2 | 1.28 | 0.528 | 2 | 0.46 | 0.793 |
| Species ID x PH | 3 | 0.79 | 0.852 | 3 | 0.19 | 0.979 | 3 | 4.53 | 0.291 | 3 | 3.28 | 0.350 |
| Species ID x SH | 3 | 6.10 | 0.107 | 3 | 3.40 | 0.334 | 3 | 5.40 | 0.145 | 3 | 0.31 | 0.959 |
| Species ID x ST | 6 | 9.36 | 0.155 | 6 | 2.06 | 0.914 | 6 | 1.41 | 0.965 | 6 | 13.49 | **0.036** |

**SRL**

| | Control | | | Drought | | | Nitrogen | | | D x N | | |
|---|---|---|---|---|---|---|---|---|---|---|---|---|
| | Df | Chi² | p | Df | Chi² | p | Df | Chi² | p | Df | Chi² | p |
| Species ID | 3 | 125.58 | <0.001 | 3 | 123.96 | <0.001 | 3 | 117.21 | <0.001 | 3 | 144.90 | <0.001 |
| Plant history (PH) | 1 | 0.31 | 0.579 | 1 | 0.04 | 0.833 | 1 | 2.81 | *0.094* | 1 | 0.05 | 0.830 |
| Soil history (SH) | 1 | <0.01 | 0.986 | 1 | 1.17 | 0.279 | 1 | 1.37 | 0.242 | 1 | 1.48 | 0.224 |
| Soil treatment (ST) | 2 | 1.46 | 0.483 | 2 | 0.67 | 0.717 | 2 | 4.01 | 0.135 | 2 | 0.28 | 0.869 |
| Species ID x PH | 3 | 5.15 | 0.161 | 3 | 2.11 | 0.550 | 3 | 2.96 | 0.397 | 3 | 2.31 | 0.510 |
| Species ID x SH | 3 | 3.89 | 0.274 | 3 | 6.14 | 0.105 | 3 | 3.40 | 0.334 | 3 | 1.93 | 0.586 |
| Species ID x ST | 6 | 13.23 | **0.040** | 6 | 2.92 | 0.819 | 6 | 2.90 | 0.821 | 6 | 14.70 | **0.023** |

**RLD**

| | Control | | | Drought | | | Nitrogen | | | D x N | | |
|---|---|---|---|---|---|---|---|---|---|---|---|---|
| | Df | Chi² | p | Df | Chi² | p | Df | Chi² | p | Df | Chi² | p |
| Species ID | 3 | 99.14 | <0.001 | 3 | 101.33 | <0.001 | 3 | 91.27 | <0.001 | 3 | 75.25 | <0.001 |

*Continued on next page*

*Continued*

|  | RLD | | | | | | | | | | | |
|---|---|---|---|---|---|---|---|---|---|---|---|---|
|  | **Control** | | | **Drought** | | | **Nitrogen** | | | **D x N** | | |
|  | Df | Chi² | p | Df | Chi² | p | Df | Chi² | p | Df | Chi² | p |
| Plant history (PH) | 1 | 0.00 | 0.956 | 1 | 3.36 | *0.067* | 1 | 0.11 | 0.742 | 1 | 0.98 | 0.323 |
| Soil history (SH) | 1 | 2.93 | *0.087* | 1 | 0.14 | 0.710 | 1 | 0.67 | 0.413 | 1 | 0.55 | 0.460 |
| Soil treatment (ST) | 2 | 2.50 | 0.286 | 2 | 2.56 | 0.279 | 2 | 0.03 | 0.983 | 2 | 4.98 | *0.083* |
| Species ID x PH | 3 | 1.35 | 0.716 | 3 | 5.11 | 0.164 | 3 | 2.59 | 0.459 | 3 | 0.59 | 0.900 |
| Species ID x SH | 3 | 5.42 | 0.144 | 3 | 2.89 | 0.409 | 3 | 0.45 | 0.929 | 3 | 0.49 | 0.921 |
| Species ID x ST | 6 | 2.77 | 0.838 | 6 | 4.44 | 0.617 | 6 | 0.91 | 0.989 | 6 | 6.27 | 0.393 |

**Appendix 1—table 6.** Summary of mixed-effect model analyses testing the effects of legacy treatments (plant history, soil history, soil treatment) on plant trait expressions of *A. elatius*, when non-treated (control) or treated with GC drivers (drought, nitrogen input, drought and nitrogen input (D x N)).

Shown are degrees of freedom (Df), Chi² and p-values (p). Significant effects (p < 0.05) are given in bold, marginally significant effects (p < 0.1) in italics.

| *A. elatius* | Growth height | | | | | | | | | | | |
|---|---|---|---|---|---|---|---|---|---|---|---|---|
|  | **Control** | | | **Drought** | | | **Nitrogen** | | | **D x N** | | |
|  | Df | Chi² | p | Df | Chi² | p | Df | Chi² | p | Df | Chi² | p |
| Plant history | 1 | 0.32 | 0.569 | 1 | 2.94 | *0.087* | 1 | 1.01 | 0.314 | 1 | 0.13 | 0.719 |
| Soil history | 1 | 1.50 | 0.221 | 1 | 0.07 | 0.787 | 1 | 0.14 | 0.706 | 1 | 0.29 | 0.593 |
| Soil treatment | 2 | 2.67 | 0.263 | 2 | 10.64 | **0.005** | 2 | 1.55 | 0.461 | 2 | 7.58 | **0.023** |

|  | Shoot nitrogen concentration | | | | | | | | | | | |
|---|---|---|---|---|---|---|---|---|---|---|---|---|
|  | **Control** | | | **Drought** | | | **Nitrogen** | | | **D x N** | | |
|  | Df | Chi² | p | Df | Chi² | p | Df | Chi² | p | Df | Chi² | p |
| Plant history | 1 | 0.52 | 0.472 | 1 | 3.46 | *0.063* | 1 | <0.01 | 0.974 | 1 | 0.06 | 0.802 |
| Soil history | 1 | 0.89 | 0.347 | 1 | 0.04 | 0.843 | 1 | 1.64 | 0.200 | 1 | 0.06 | 0.803 |
| Soil treatment | 2 | 1.40 | 0.497 | 2 | 1.54 | 0.462 | 2 | 1.99 | 0.369 | 2 | 2.07 | 0.354 |

|  | Leaf greenness | | | | | | | | | | | |
|---|---|---|---|---|---|---|---|---|---|---|---|---|
|  | **Control** | | | **Drought** | | | **Nitrogen** | | | **D x N** | | |
|  | Df | Chi² | p | Df | Chi² | p | Df | Chi² | p | Df | Chi² | p |
| Plant history | 1 | 1.19 | 0.275 | 1 | 0.60 | 0.438 | 1 | 1.13 | 0.288 | 1 | 0.22 | 0.636 |
| Soil history | 1 | 1.50 | 0.221 | 1 | 0.99 | 0.321 | 1 | 0.03 | 0.862 | 1 | 0.15 | 0.699 |
| Soil treatment | 2 | 5.20 | *0.074* | 2 | 0.44 | 0.801 | 2 | 3.64 | 0.162 | 2 | 0.84 | 0.656 |

|  | LDMC | | | | | | | | | | | |
|---|---|---|---|---|---|---|---|---|---|---|---|---|
|  | **Control** | | | **Drought** | | | **Nitrogen** | | | **D x N** | | |
|  | Df | Chi² | p | Df | Chi² | p | Df | Chi² | p | Df | Chi² | p |
| Plant history | 1 | 0.01 | 0.942 | 1 | 1.15 | 0.284 | 1 | <0.01 | 0.987 | 1 | 1.02 | 0.313 |
| Soil history | 1 | 0.07 | 0.798 | 1 | 0.13 | 0.718 | 1 | 0.04 | 0.837 | 1 | 0.31 | 0.580 |
| Soil treatment | 2 | 0.03 | 0.985 | 2 | 0.34 | 0.844 | 2 | 2.00 | 0.369 | 2 | 2.44 | 0.295 |

|  | SLA | | | | | | | | | | | |
|---|---|---|---|---|---|---|---|---|---|---|---|---|
|  | **Control** | | | **Drought** | | | **Nitrogen** | | | **D x N** | | |
|  | Df | Chi² | p | Df | Chi² | p | Df | Chi² | p | Df | Chi² | p |
| Plant history | 1 | 0.44 | 0.507 | 1 | 0.61 | 0.435 | 1 | 0.48 | 0.488 | 1 | 1.63 | 0.202 |
| Soil history | 1 | 0.04 | 0.836 | 1 | 0.22 | 0.638 | 1 | 0.88 | 0.348 | 1 | 1.08 | 0.300 |
| Soil treatment | 2 | 0.59 | 0.744 | 2 | 0.13 | 0.936 | 2 | 2.74 | 0.254 | 2 | 3.10 | 0.212 |

| Stomatal conductance | | | | | | | | | | | |
|---|---|---|---|---|---|---|---|---|---|---|---|
| | Control | | | Drought | | | Nitrogen | | | D x N | | |
| | Df | Chi² | p | Df | Chi² | p | Df | Chi² | p | Df | Chi² | p |
| Temperature | 1 | 0.05 | 0.827 | 1 | 0.53 | 0.465 | 1 | 0.91 | 0.340 | 1 | 0.09 | 0.763 |
| Daytime | 1 | 6.15 | **0.013** | 1 | 3.92 | **0.048** | 1 | 0.68 | 0.408 | 1 | 0.37 | 0.544 |
| Plant history | 1 | 0.49 | 0.484 | 1 | 0.05 | 0.824 | 1 | 1.23 | 0.267 | 1 | 0.18 | 0.670 |
| Soil history | 1 | 0.83 | 0.361 | 1 | 0.13 | 0.718 | 1 | 0.92 | 0.336 | 1 | <0.01 | 0.998 |
| Soil treatment | 2 | 0.96 | 0.618 | 2 | 1.69 | 0.429 | 2 | 2.99 | 0.224 | 2 | 0.33 | 0.846 |

| Root diameter | | | | | | | | | | | |
|---|---|---|---|---|---|---|---|---|---|---|---|
| | Control | | | Drought | | | Nitrogen | | | D x N | | |
| | Df | Chi² | p | Df | Chi² | p | Df | Chi² | p | Df | Chi² | p |
| Plant history | 1 | 0.24 | 0.627 | 1 | <0.01 | 0.972 | 1 | 0.46 | 0.497 | 1 | 0.45 | 0.503 |
| Soil history | 1 | 1.37 | 0.242 | 1 | 0.53 | 0.467 | 1 | 2.59 | 0.108 | 1 | 0.10 | 0.754 |
| Soil treatment | 2 | 4.85 | *0.089* | 2 | 0.52 | 0.770 | 2 | 1.00 | 0.605 | 2 | 3.86 | 0.145 |

| SRL | | | | | | | | | | | |
|---|---|---|---|---|---|---|---|---|---|---|---|
| | Control | | | Drought | | | Nitrogen | | | D x N | | |
| | Df | Chi² | p | Df | Chi² | p | Df | Chi² | p | Df | Chi² | p |
| Plant history | 1 | 0.80 | 0.371 | 1 | 0.16 | 0.686 | 1 | 2.32 | 0.128 | 1 | 0.54 | 0.462 |
| Soil history | 1 | 0.06 | 0.807 | 1 | 0.02 | 0.884 | 1 | 2.66 | 0.103 | 1 | 0.21 | 0.649 |
| Soil treatment | 2 | 2.94 | 0.230 | 2 | 1.81 | 0.404 | 2 | 4.63 | *0.099* | 2 | 9.49 | **0.009** |

| RLD | | | | | | | | | | | |
|---|---|---|---|---|---|---|---|---|---|---|---|
| | Control | | | Drought | | | Nitrogen | | | D x N | | |
| | Df | Chi² | p | Df | Chi² | p | Df | Chi² | p | Df | Chi² | p |
| Plant history | 1 | 1.02 | 0.313 | 1 | 0.03 | 0.859 | 1 | 2.42 | 0.120 | 1 | 1.44 | 0.230 |
| Soil history | 1 | 2.51 | 0.113 | 1 | 1.14 | 0.286 | 1 | 1.03 | 0.310 | 1 | 0.46 | 0.500 |
| Soil treatment | 2 | 4.52 | 0.104 | 2 | 1.24 | 0.539 | 2 | 0.26 | 0.878 | 2 | 1.40 | 0.497 |

**Appendix 1—table 7.** Summary of mixed-effect model analyses testing the effects of legacy treatments (plant history, soil history, soil treatment) on plant trait expressions of *A. pratensis*, when non-treated (control) or treated with GC drivers (drought, nitrogen input, drought and nitrogen input (D x N)).

Shown are degrees of freedom (Df), Chi² and p-values (p). Significant effects (p < 0.05) are given in bold, marginally significant effects (p < 0.1) in italics.

| *A. pratensis* | Growth height | | | | | | | | | | |
|---|---|---|---|---|---|---|---|---|---|---|---|
| | Control | | | Drought | | | Nitrogen | | | D x N | | |
| | Df | Chi² | p | Df | Chi² | p | Df | Chi² | p | Df | Chi² | p |
| Plant history | 1 | 1.50 | 0.221 | 1 | 0.56 | 0.454 | 1 | 0.03 | 0.868 | 1 | 0.94 | 0.332 |
| Soil history | 1 | 0.44 | 0.508 | 1 | 0.15 | 0.700 | 1 | 0.03 | 0.874 | 1 | 0.82 | 0.365 |
| Soil treatment | 2 | 5.77 | *0.056* | 2 | 6.56 | **0.038** | 2 | 3.00 | 0.223 | 2 | 0.26 | 0.879 |

| Shoot nitrogen concentration | | | | | | | | | | | |
|---|---|---|---|---|---|---|---|---|---|---|---|
| | Control | | | Drought | | | Nitrogen | | | D x N | | |
| | Df | Chi² | p | Df | Chi² | p | Df | Chi² | p | Df | Chi² | p |
| Plant history | 1 | 1.75 | 0.186 | 1 | 0.17 | 0.680 | 1 | 0.10 | 0.755 | 1 | 0.84 | 0.358 |
| Soil history | 1 | 0.37 | 0.544 | 1 | 0.96 | 0.328 | 1 | 0.01 | 0.939 | 1 | 0.00 | 0.966 |
| Soil treatment | 2 | 4.61 | *0.100* | 2 | 1.74 | 0.419 | 2 | 9.05 | **0.011** | 2 | 6.83 | **0.033** |

**Leaf greenness**

| | Control | | | Drought | | | Nitrogen | | | D x N | | |
|---|---|---|---|---|---|---|---|---|---|---|---|---|
| | Df | Chi² | p | Df | Chi² | p | Df | Chi² | p | Df | Chi² | p |
| Plant history | 1 | 0.07 | 0.786 | 1 | 1.03 | 0.311 | 1 | 0.58 | 0.445 | 1 | 0.18 | 0.673 |
| Soil history | 1 | 0.03 | 0.869 | 1 | 1.85 | 0.174 | 1 | 0.90 | 0.343 | 1 | 0.19 | 0.661 |
| Soil treatment | 2 | 1.16 | 0.560 | 2 | 0.60 | 0.743 | 2 | 0.61 | 0.737 | 2 | 2.21 | 0.332 |

**LDMC**

| | Control | | | Drought | | | Nitrogen | | | D x N | | |
|---|---|---|---|---|---|---|---|---|---|---|---|---|
| | Df | Chi² | p | Df | Chi² | p | Df | Chi² | p | Df | Chi² | p |
| Plant history | 1 | 0.34 | 0.561 | 1 | 0.38 | 0.538 | 1 | 0.40 | 0.527 | 1 | 2.17 | 0.140 |
| Soil history | 1 | 0.11 | 0.736 | 1 | 3.62 | 0.057 | 1 | 2.32 | 0.128 | 1 | 0.05 | 0.821 |
| Soil treatment | 2 | 0.36 | 0.835 | 2 | 1.42 | 0.492 | 2 | 1.18 | 0.555 | 2 | 3.91 | 0.141 |

**SLA**

| | Control | | | Drought | | | Nitrogen | | | D x N | | |
|---|---|---|---|---|---|---|---|---|---|---|---|---|
| | Df | Chi² | p | Df | Chi² | p | Df | Chi² | p | Df | Chi² | p |
| Plant history | 1 | 0.07 | 0.786 | 1 | 0.32 | 0.572 | 1 | 0.00 | 0.984 | 1 | 1.28 | 0.259 |
| Soil history | 1 | 0.20 | 0.654 | 1 | 2.81 | 0.094 | 1 | 0.23 | 0.632 | 1 | 0.05 | 0.828 |
| Soil treatment | 2 | 2.21 | 0.331 | 2 | 0.70 | 0.704 | 2 | 1.18 | 0.555 | 2 | 8.59 | **0.014** |

**Stomatal conductance**

| | Control | | | Drought | | | Nitrogen | | | D x N | | |
|---|---|---|---|---|---|---|---|---|---|---|---|---|
| | Df | Chi² | p | Df | Chi² | p | Df | Chi² | p | Df | Chi² | p |
| Temperature | 1 | 1.17 | 0.279 | 1 | 0.22 | 0.642 | 1 | 0.44 | 0.507 | 1 | 0.17 | 0.678 |
| Daytime | 1 | 0.77 | 0.379 | 1 | 0.07 | 0.786 | 1 | 1.13 | 0.289 | 1 | 8.38 | **0.004** |
| Plant history | 1 | 0.05 | 0.824 | 1 | 0.16 | 0.690 | 1 | 0.66 | 0.415 | 1 | 0.61 | 0.436 |
| Soil history | 1 | 1.30 | 0.255 | 1 | 0.14 | 0.706 | 1 | 0.79 | 0.373 | 1 | 0.53 | 0.466 |
| Soil treatment | 2 | 2.35 | 0.308 | 2 | 4.41 | 0.110 | 2 | 2.55 | **0.002** | 2 | 1.59 | 0.452 |

**Root diameter**

| | Control | | | Drought | | | Nitrogen | | | D x N | | |
|---|---|---|---|---|---|---|---|---|---|---|---|---|
| | Df | Chi² | p | Df | Chi² | p | Df | Chi² | p | Df | Chi² | p |
| Plant history | 1 | 0.28 | 0.595 | 1 | 0.18 | 0.673 | 1 | 0.20 | 0.653 | 1 | 0.09 | 0.770 |
| Soil history | 1 | 5.61 | **0.018** | 1 | 0.95 | 0.331 | 1 | 1.34 | 0.246 | 1 | 0.01 | 0.942 |
| Soil treatment | 2 | 1.02 | 0.602 | 2 | 0.29 | 0.865 | 2 | 1.25 | 0.535 | 2 | 6.06 | **0.048** |

**SRL**

| | Control | | | Drought | | | Nitrogen | | | D x N | | |
|---|---|---|---|---|---|---|---|---|---|---|---|---|
| | Df | Chi² | p | Df | Chi² | p | Df | Chi² | p | Df | Chi² | p |
| Plant history | 1 | 0.42 | 0.515 | 1 | 0.24 | 0.623 | 1 | 0.61 | 0.435 | 1 | 0.01 | 0.916 |
| Soil history | 1 | 0.33 | 0.567 | 1 | 7.10 | **0.008** | 1 | 0.17 | 0.677 | 1 | 2.73 | 0.098 |
| Soil treatment | 2 | 5.24 | 0.073 | 2 | 0.88 | 0.644 | 2 | 0.11 | 0.945 | 2 | 6.03 | **0.049** |

**RLD**

| | Control | | | Drought | | | Nitrogen | | | D x N | | |
|---|---|---|---|---|---|---|---|---|---|---|---|---|
| | Df | Chi² | p | Df | Chi² | p | Df | Chi² | p | Df | Chi² | p |
| Plant history | 1 | 0.28 | 0.595 | 1 | 0.12 | 0.729 | 1 | 0.08 | 0.781 | 1 | 0.09 | 0.763 |
| Soil history | 1 | 0.75 | 0.387 | 1 | 4.79 | **0.029** | 1 | 0.13 | 0.716 | 1 | 0.03 | 0.861 |
| Soil treatment | 2 | 0.28 | 0.869 | 2 | 2.39 | 0.303 | 2 | 0.19 | 0.909 | 2 | 3.02 | 0.221 |

**Appendix 1—table 8.** Summary of mixed-effect model analyses testing the effects of legacy treatments (plant history, soil history, soil treatment) on plant trait expressions of *D. glomerata*, when non-treated (control) or treated with GC drivers (drought, nitrogen input, drought and nitrogen input (D x N)).

Shown are degrees of freedom (Df), Chi² and p-values (p). Significant effects (P < 0.05) are given in bold, marginally significant effects (P < 0.1) in italics.

| *D. glomerata* | **Growth height** | | | | | | | | | | | |
|---|---|---|---|---|---|---|---|---|---|---|---|---|
| | **Control** | | | **Drought** | | | **Nitrogen** | | | **D x N** | | |
| | **Df** | **Chi²** | **p** | **Df** | **Chi²** | **p** | **Df** | **Chi²** | **p** | **Df** | **Chi²** | **p** |
| Plant history | 1 | 0.73 | 0.394 | 1 | 0.11 | 0.741 | 1 | 0.06 | 0.802 | 1 | 0.01 | 0.912 |
| Soil history | 1 | 0.69 | 0.405 | 1 | 0.91 | 0.340 | 1 | 1.25 | 0.263 | 1 | 0.18 | 0.675 |
| Soil treatment | 2 | 1.66 | 0.436 | 2 | 1.06 | 0.589 | 2 | 2.37 | 0.306 | 2 | 1.09 | 0.581 |

| | **Shoot nitrogen concentration** | | | | | | | | | | | |
|---|---|---|---|---|---|---|---|---|---|---|---|---|
| | **Control** | | | **Drought** | | | **Nitrogen** | | | **D x N** | | |
| | **Df** | **Chi²** | **p** | **Df** | **Chi²** | **p** | **Df** | **Chi²** | **p** | **Df** | **Chi²** | **p** |
| Plant history | 1 | 1.13 | 0.289 | 1 | 0.56 | 0.455 | 1 | 2.38 | 0.123 | 1 | 0.56 | 0.453 |
| Soil history | 1 | <0.01 | 0.952 | 1 | 2.18 | 0.140 | 1 | 8.44 | **0.004** | 1 | 0.05 | 0.818 |
| Soil treatment | 2 | 2.72 | 0.257 | 2 | 2.46 | 0.293 | 2 | 3.07 | 0.215 | 2 | 0.71 | 0.701 |

| | **Leaf greenness** | | | | | | | | | | | |
|---|---|---|---|---|---|---|---|---|---|---|---|---|
| | **Control** | | | **Drought** | | | **Nitrogen** | | | **D x N** | | |
| | **Df** | **Chi²** | **p** | **Df** | **Chi²** | **p** | **Df** | **Chi²** | **p** | **Df** | **Chi²** | **p** |
| Plant history | 1 | 4.93 | **0.026** | 1 | 0.02 | 0.886 | 1 | 0.17 | 0.680 | 1 | 0.13 | 0.723 |
| Soil history | 1 | 1.23 | 0.267 | 1 | 1.17 | 0.279 | 1 | 0.15 | 0.703 | 1 | 0.01 | 0.908 |
| Soil treatment | 2 | 2.33 | 0.313 | 2 | 3.58 | 0.167 | 2 | 1.16 | 0.560 | 2 | 0.68 | 0.713 |

| | **LDMC** | | | | | | | | | | | |
|---|---|---|---|---|---|---|---|---|---|---|---|---|
| | **Control** | | | **Drought** | | | **Nitrogen** | | | **D x N** | | |
| | **Df** | **Chi²** | **p** | **Df** | **Chi²** | **p** | **Df** | **Chi²** | **p** | **Df** | **Chi²** | **p** |
| Plant history | 1 | 0.86 | 0.353 | 1 | 1.18 | 0.278 | 1 | 0.37 | 0.540 | 1 | 0.64 | 0.423 |
| Soil history | 1 | 2.03 | 0.154 | 1 | 0.12 | 0.727 | 1 | 2.21 | 0.137 | 1 | 0.28 | 0.594 |
| Soil treatment | 2 | 2.36 | 0.307 | 2 | 0.20 | 0.905 | 2 | 1.74 | 0.418 | 2 | 3.05 | 0.218 |

| | **SLA** | | | | | | | | | | | |
|---|---|---|---|---|---|---|---|---|---|---|---|---|
| | **Control** | | | **Drought** | | | **Nitrogen** | | | **D x N** | | |
| | **Df** | **Chi²** | **p** | **Df** | **Chi²** | **p** | **Df** | **Chi²** | **p** | **Df** | **Chi²** | **p** |
| Plant history | 1 | 1.41 | 0.235 | 1 | 0.01 | 0.904 | 1 | 1.50 | 0.220 | 1 | 0.14 | 0.706 |
| Soil history | 1 | 2.29 | 0.130 | 1 | 0.28 | 0.595 | 1 | 3.86 | **0.050** | 1 | 0.02 | 0.888 |
| Soil treatment | 2 | 2.60 | 0.272 | 2 | 1.88 | 0.392 | 2 | 0.09 | 0.956 | 2 | 0.89 | 0.641 |

| | **Stomatal conductance** | | | | | | | | | | | |
|---|---|---|---|---|---|---|---|---|---|---|---|---|
| | **Control** | | | **Drought** | | | **Nitrogen** | | | **D x N** | | |
| | **Df** | **Chi²** | **p** | **Df** | **Chi²** | **p** | **Df** | **Chi²** | **p** | **Df** | **Chi²** | **p** |
| Temperature | 1 | 1.12 | 0.289 | 1 | <0.01 | 0.951 | 1 | 0.04 | 0.843 | 1 | 0.08 | 0.782 |
| Daytime | 1 | 24.06 | **<0.001** | 1 | 12.16 | **<0.001** | 1 | 4.04 | **0.044** | 1 | 4.37 | **0.037** |
| Plant history | 1 | 3.77 | *0.052* | 1 | 1.05 | 0.304 | 1 | 1.79 | 0.181 | 1 | 4.89 | **0.027** |
| Soil history | 1 | 1.44 | 0.231 | 1 | 1.55 | 0.214 | 1 | 0.47 | 0.493 | 1 | 2.34 | 0.126 |
| Soil treatment | 2 | 0.43 | 0.805 | 2 | 1.62 | 0.445 | 2 | 0.27 | 0.872 | 2 | 1.04 | 0.595 |

**Root diameter**

| | Control | | | Drought | | | Nitrogen | | | D x N | | |
|---|---|---|---|---|---|---|---|---|---|---|---|---|
| | Df | Chi² | p | Df | Chi² | p | Df | Chi² | p | Df | Chi² | p |
| Plant history | 1 | 0.64 | 0.422 | 1 | 0.02 | 0.876 | 1 | 1.83 | 0.176 | 1 | 2.43 | 0.119 |
| Soil history | 1 | 0.33 | 0.567 | 1 | 2.50 | 0.114 | 1 | 0.34 | 0.559 | 1 | 0.16 | 0.691 |
| Soil treatment | 2 | 0.60 | 0.741 | 2 | 3.21 | 0.201 | 2 | 0.16 | 0.924 | 2 | 2.03 | 0.363 |

**SRL**

| | Control | | | Drought | | | Nitrogen | | | D x N | | |
|---|---|---|---|---|---|---|---|---|---|---|---|---|
| | Df | Chi² | p | Df | Chi² | p | Df | Chi² | p | Df | Chi² | p |
| Plant history | 1 | 2.55 | 0.111 | 1 | 0.54 | 0.462 | 1 | 0.08 | 0.777 | 1 | 0.36 | 0.548 |
| Soil history | 1 | 1.73 | 0.188 | 1 | 1.42 | 0.233 | 1 | 0.32 | 0.570 | 1 | 0.22 | 0.643 |
| Soil treatment | 2 | 2.23 | 0.329 | 2 | 0.24 | 0.888 | 2 | 2.28 | 0.320 | 2 | 2.38 | 0.304 |

**RLD**

| | Control | | | Drought | | | Nitrogen | | | D x N | | |
|---|---|---|---|---|---|---|---|---|---|---|---|---|
| | Df | Chi² | p | Df | Chi² | p | Df | Chi² | p | Df | Chi² | p |
| Plant history | 1 | 0.01 | 0.923 | 1 | 7.58 | **0.006** | 1 | 0.77 | 0.380 | 1 | 0.03 | 0.862 |
| Soil history | 1 | 0.27 | 0.602 | 1 | 0.02 | 0.901 | 1 | 0.54 | 0.464 | 1 | 0.18 | 0.673 |
| Soil treatment | 2 | 0.36 | 0.835 | 2 | 4.51 | 0.105 | 2 | 0.96 | 0.619 | 2 | 5.25 | *0.073* |

**Appendix 1—table 9.** Summary of mixed-effect model analyses testing the effects of legacy treatments (plant history, soil history, soil treatment) on plant trait expressions of *P. trivialis*, when non-treated (control) or treated with GC drivers (drought, nitrogen input, drought and nitrogen input (D x N)).

Shown are degrees of freedom (Df), Chi² and p-values (p). Significant effects (p < 0.05) are given in bold, marginally significant effects (p < 0.1) in italics.

*P. trivialis* — **Growth height**

| | Control | | | Drought | | | Nitrogen | | | D x N | | |
|---|---|---|---|---|---|---|---|---|---|---|---|---|
| | Df | Chi² | p | Df | Chi² | p | Df | Chi² | p | Df | Chi² | p |
| Plant history | 1 | 2.81 | *0.094* | 1 | 0.32 | 0.571 | 1 | 0.98 | 0.323 | 1 | 0.16 | 0.688 |
| Soil history | 1 | 0.62 | 0.429 | 1 | 0.92 | 0.338 | 1 | 1.12 | 0.289 | 1 | 5.02 | **0.025** |
| Soil treatment | 2 | 4.77 | *0.092* | 2 | 1.59 | 0.452 | 2 | 2.99 | 0.224 | 2 | 1.14 | 0.566 |

**Shoot nitrogen concentration**

| | Control | | | Drought | | | Nitrogen | | | D x N | | |
|---|---|---|---|---|---|---|---|---|---|---|---|---|
| | Df | Chi² | p | Df | Chi² | p | Df | Chi² | p | Df | Chi² | p |
| Plant history | 1 | <0.01 | 0.986 | 1 | 0.01 | 0.934 | 1 | 0.07 | 0.785 | 1 | 0.01 | 0.915 |
| Soil history | 1 | 0.15 | 0.695 | 1 | 0.57 | 0.452 | 1 | 0.06 | 0.802 | 1 | 0.45 | 0.503 |
| Soil treatment | 2 | 9.66 | **0.008** | 2 | 2.33 | 0.313 | 2 | 1.18 | 0.554 | 2 | 3.86 | 0.145 |

**Leaf greenness**

| | Control | | | Drought | | | Nitrogen | | | D x N | | |
|---|---|---|---|---|---|---|---|---|---|---|---|---|
| | Df | Chi² | P | Df | Chi² | P | Df | Chi² | P | Df | Chi² | P |
| Plant history | 1 | 0.14 | 0.708 | 1 | 2.41 | 0.120 | 1 | 0.04 | 0.845 | 1 | 2.35 | 0.126 |
| Soil history | 1 | 1.41 | 0.236 | 1 | 1.10 | 0.295 | 1 | 0.38 | 0.537 | 1 | 0.09 | 0.769 |
| Soil treatment | 2 | 0.13 | 0.936 | 2 | 0.37 | 0.833 | 2 | 5.22 | *0.074* | 2 | 0.97 | 0.616 |

**LDMC**

| | Control | | | Drought | | | Nitrogen | | | D x N | | |
|---|---|---|---|---|---|---|---|---|---|---|---|---|
| | Df | Chi² | P | Df | Chi² | P | Df | Chi² | P | Df | Chi² | P |
| Plant history | 1 | 0.81 | 0.369 | 1 | 0.24 | 0.627 | 1 | 0.05 | 0.826 | 1 | 1.34 | 0.247 |
| Soil history | 1 | 0.08 | 0.776 | 1 | 0.01 | 0.927 | 1 | 0.47 | 0.492 | 1 | 4.25 | **0.039** |
| Soil treatment | 2 | 3.34 | 0.188 | 2 | 0.72 | 0.696 | 2 | 3.01 | 0.222 | 2 | 2.64 | 0.268 |

**SLA**

| | Control | | | Drought | | | Nitrogen | | | D x N | | |
|---|---|---|---|---|---|---|---|---|---|---|---|---|
| | Df | Chi² | p | Df | Chi² | p | Df | Chi² | p | Df | Chi² | p |
| Plant history | 1 | 0.26 | 0.611 | 1 | 0.21 | 0.643 | 1 | 1.44 | 0.231 | 1 | 0.80 | 0.372 |
| Soil history | 1 | 0.41 | 0.522 | 1 | 0.40 | 0.528 | 1 | 1.47 | 0.226 | 1 | 0.33 | 0.565 |
| Soil treatment | 2 | 2.29 | 0.319 | 2 | 0.53 | 0.769 | 2 | 3.35 | 0.187 | 2 | 4.08 | 0.130 |

**Stomatal conductance**

| | Control | | | Drought | | | Nitrogen | | | D x N | | |
|---|---|---|---|---|---|---|---|---|---|---|---|---|
| | Df | Chi² | p | Df | Chi² | p | Df | Chi² | p | Df | Chi² | p |
| Temperature | 1 | 10.96 | **0.001** | 1 | 8.08 | **0.004** | 1 | 7.25 | **0.007** | 1 | 4.31 | **0.038** |
| Daytime | 1 | 3.93 | **0.047** | 1 | 1.12 | 0.289 | 1 | 1.22 | 0.270 | 1 | 6.35 | **0.012** |
| Plant history | 1 | <0.01 | 0.949 | 1 | 0.60 | 0.439 | 1 | 2.96 | *0.085* | 1 | 0.29 | 0.589 |
| Soil history | 1 | 0.68 | 0.410 | 1 | 0.95 | 0.330 | 1 | 2.72 | *0.099* | 1 | 0.14 | 0.704 |
| Soil treatment | 2 | 2.46 | 0.293 | 2 | 0.54 | 0.763 | 2 | 0.95 | 0.622 | 2 | 1.49 | 0.474 |

**Root diameter**

| | Control | | | Drought | | | Nitrogen | | | D x N | | |
|---|---|---|---|---|---|---|---|---|---|---|---|---|
| | Df | Chi² | p | Df | Chi² | p | Df | Chi² | p | Df | Chi² | p |
| Plant history | 1 | 0.16 | 0.686 | 1 | 0.07 | 0.794 | 1 | 0.55 | 0.458 | 1 | 2.91 | *0.088* |
| Soil history | 1 | 3.06 | *0.080* | 1 | 0.31 | 0.579 | 1 | 0.95 | 0.329 | 1 | 0.06 | 0.800 |
| Soil treatment | 2 | 7.48 | **0.024** | 2 | 0.28 | 0.870 | 2 | 0.07 | 0.967 | 2 | 2.00 | 0.369 |

**SRL**

| | Control | | | Drought | | | Nitrogen | | | D x N | | |
|---|---|---|---|---|---|---|---|---|---|---|---|---|
| | Df | Chi² | p | Df | Chi² | p | Df | Chi² | p | Df | Chi² | p |
| Plant history | 1 | 2.10 | 0.147 | 1 | 0.94 | 0.332 | 1 | 1.82 | 0.178 | 1 | 1.04 | 0.308 |
| Soil history | 1 | 1.83 | 0.177 | 1 | 3.68 | *0.055* | 1 | 2.26 | 0.133 | 1 | 0.19 | 0.660 |
| Soil treatment | 2 | 5.73 | *0.057* | 2 | 0.56 | 0.755 | 2 | 1.97 | 0.374 | 2 | 1.83 | 0.401 |

**RLD**

| | Control | | | Drought | | | Nitrogen | | | D x N | | |
|---|---|---|---|---|---|---|---|---|---|---|---|---|
| | Df | Chi² | p | Df | Chi² | p | Df | Chi² | p | Df | Chi² | p |
| Plant history | 1 | 0.23 | 0.632 | 1 | 0.01 | 0.904 | 1 | 0.01 | 0.920 | 1 | 0.54 | 0.463 |
| Soil history | 1 | 3.38 | *0.066* | 1 | 0.07 | 0.792 | 1 | 0.01 | 0.926 | 1 | 0.16 | 0.685 |
| Soil treatment | 2 | 0.63 | 0.731 | 2 | 0.61 | 0.739 | 2 | 0.16 | 0.924 | 2 | 3.25 | 0.197 |

**Appendix 1—table 10.** Summary of mixed-effect model analyses testing the effects of legacy treatments (plant history, soil history, soil treatment) on mildew infestation of *D. glomerata* and *P. trivialis*, when non-treated (control) or treated with GC drivers (drought, nitrogen input, drought and nitrogen input (D x N)).

Shown are degrees of freedom (Df), Chi² and p-values (p). Significant effects (p < 0.05) are given in bold, marginally significant effects (p < 0.1) in italics.

**Mildew infestation** — *D. glomerata*

| | Control | | | Drought | | | Nitrogen | | | D x N | | |
|---|---|---|---|---|---|---|---|---|---|---|---|---|
| | Df | Chi² | p | Df | Chi² | p | Df | Chi² | p | Df | Chi² | p |
| Plant history | 1 | 0.58 | 0.447 | 1 | 1.18 | 0.277 | 1 | 0.88 | 0.348 | 1 | 0.26 | 0.613 |
| Soil history | 1 | 0.41 | 0.522 | 1 | 2.63 | 0.105 | 1 | <0.01 | 0.946 | 1 | 0.11 | 0.746 |
| Soil treatment | 2 | 6.01 | **0.049** | 2 | 7.65 | **0.022** | 2 | 0.93 | 0.628 | 2 | 0.09 | 0.958 |

*P. trivialis*

| | Control | | | Drought | | | Nitrogen | | | D x N | | |
|---|---|---|---|---|---|---|---|---|---|---|---|---|
| | Df | Chi² | p | Df | Chi² | p | Df | Chi² | p | Df | Chi² | p |
| Plant history | 1 | <0.01 | 0.996 | 1 | <0.01 | 0.973 | 1 | 0.03 | 0.860 | 1 | 0.21 | 0.647 |
| Soil history | 1 | 1.20 | 0.274 | 1 | 2.66 | 0.103 | 1 | 1.68 | 0.195 | 1 | 0.05 | 0.817 |
| Soil treatment | 2 | 3.94 | 0.139 | 2 | 1.78 | 0.412 | 2 | 0.16 | 0.921 | 2 | 2.10 | 0.350 |

**Appendix 1—table 11.** Summary of mixed-effect model analyses testing the effects of species identity (N = 4), legacy treatments (plant history, soil history, soil treatment) and their interactions on root-shoot ratio, when non-treated (control) or treated with global change drivers (drought, nitrogen input, drought and nitrogen input (D x N)).

Shown are degrees of freedom (Df), Chi² and p-values (p). Significant effects (p < 0.05) are given in bold, marginally significant effects (p < 0.1) in italics.

**Root-Shoot ratio**

| | Control | | | Drought | | | Nitrogen | | | D x N | | |
|---|---|---|---|---|---|---|---|---|---|---|---|---|
| | Df | Chi² | p | Df | Chi² | p | Df | Chi² | p | Df | Chi² | p |
| Species ID | 3 | 115.37 | **<0.001** | 3 | 116.36 | **<0.001** | 3 | 101.12 | **<0.001** | 3 | 108.37 | **<0.001** |
| Plant history (PH) | 1 | 0.02 | 0.880 | 1 | 1.48 | 0.225 | 1 | 1.64 | 0.200 | 1 | 0.46 | 0.496 |
| Soil history (SH) | 1 | 1.81 | 0.178 | 1 | 1.60 | 0.206 | 1 | 0.24 | 0.622 | 1 | <0.01 | 0.992 |
| Soil treatment (ST) | 2 | 0.46 | 0.793 | 2 | 1.96 | 0.376 | 2 | 1.19 | 0.551 | 2 | 3.54 | 0.170 |
| Species ID x PH | 3 | 3.88 | 0.275 | 3 | 1.47 | 0.690 | 3 | 0.86 | 0.836 | 3 | 2.77 | 0.428 |
| Species ID x SH | 3 | 5.98 | 0.113 | 3 | 3.99 | 0.263 | 3 | 2.53 | 0.471 | 3 | 3.71 | 0.295 |
| Species ID x ST | 6 | 10.54 | 0.104 | 6 | 6.76 | 0.344 | 6 | 1.85 | 0.933 | 6 | 14.79 | **0.022** |

## Appendix 2

### Hypothesis 2: global change drivers have a strong impact on biomass production and trait expression

Plant traits (separately for each species)

The grass *P. trivialis* was the only species which growth height decreased with drought, all other species showed no significant change under drought. Under nitrogen input, the species *P. trivialis*, *A. pratensis*, and *D. glomerata* (marginally significant) increased in growth height, while under the combined impact of both global change drivers, no species significantly changed in growth height (*D. glomerata* marginally significantly increased in height).

The species *A.elatius*, *D. glomerata,* and *A. pratensis* increased in shoot nitrogen concentration and leaf greenness under the impact of drought and/or nitrogen input (similar to analysis across all species). In *P. trivialis*, drought did not affect shoot nitrogen concentration or leaf greenness, and there was no additive impact of both global change drivers on leaf greenness (leaf greenness was as high as in fertilized plants).

Global change drivers had no significant influence on LDMC or SLA of *A. elatius* and *A. pratensis* except for LDMC decrease and SLA increase of *A. elatius* plants when treated with both global change drivers. Plants of *D. glomerata* decreased in LDMC and increased in SLA when treated with single global change drivers, while nitrogen input had a stronger impact than drought. When treated with both global change drivers, *D. glomerata* plants had still a significantly lower LDMC and higher SLA compared to control plants. In *P. trivialis*, drought had no significant influence on LDMC and SLA, while nitrogen input decreased LDMC and increased SLA. When treated with both global change drivers, LDMC and SLA were as high as in fertilized plants.

In *D. glomerata*, stomatal conductance was increased, when plants were treated with drought, and in *A. pratensis* decreased, when treated with both global change drivers. Stomatal conductance in *A. elatius* and *P. trivialis* did not change with global change treatments.

In *A. elatius*, SRL decreased when fertilized, irrespective of drought, while other root traits did not change significantly. In *A. pratensis*, drought, nitrogen input, and both global change drivers together had similar negative impacts on SRL and RLD (except for RLD under nitrogen input, which did not change). Root diameter of *A. pratensis* plants increased under single global change drivers with additive effects under the combined application. In *D. glomerata*, RLD increased and in *P. trivialis* RLD decreased and root diameter increased, when treated with both global change drivers.

**Appendix 2—table 1.** Summary of mixed-effect model analyses testing the effects of species identity (N = 4), legacy treatments (plant history, soil history, soil treatment), global change treatments (drought, nitrogen input) and their interactions on root-shoot ratio.

Shown are degrees of freedom (Df), Chi² and P-values (P). Significant effects (P < 0.05) are given in bold, marginally significant effects (P < 0.1) in italics.

| | Root-shoot ratio | | |
|---|---|---|---|
| | Df | Chi² | p |
| Species identity (ID) | 3 | 133.41 | **<0.001** |
| Plant history | 1 | 1.11 | 0.292 |
| Soil history | 1 | 1.08 | 0.300 |
| Soil treatment | 2 | 1.81 | 0.404 |
| Drought (D) | 1 | 60.01 | **<0.001** |
| Nitrogen input (N) | 1 | 89.83 | **<0.001** |
| Species ID x Plant history | 3 | 0.87 | 0.832 |
| Species ID x Soil history | 3 | 4.07 | 0.254 |
| Species ID x Soil treatment | 6 | 2.79 | 0.835 |
| Species ID x D | 3 | 95.53 | **<0.001** |

*Appendix 2—table 1 Continued on next page*

*Appendix 2—table 1 Continued*

| | Root-shoot ratio | | |
|---|---|---|---|
| | Df | Chi² | p |
| Species ID x N | 3 | 9.31 | **0.025** |
| D x N | 1 | 2.19 | 0.139 |
| Species ID x Plant history x D | 4 | 2.02 | 0.733 |
| Species ID x Soil history x D | 4 | 1.58 | 0.812 |
| Species ID x Soil treatment x D | 8 | 4.97 | 0.760 |
| Species ID x Plant history x N | 4 | 2.91 | 0.573 |
| Species ID x Soil history x N | 4 | 3.18 | 0.528 |
| Species ID x Soil treatment x N | 8 | 18.18 | **0.020** |
| Species ID x Plant history x D x N | 4 | 10.42 | **0.034** |
| Species ID x Soil history x D x N | 4 | 11.14 | **0.025** |
| Species ID x Soil treatment x D x N | 8 | 5.20 | 0.736 |

**Appendix 2—table 2.** Summary of mixed-effect model analyses testing the effects of species identity (N = 4), legacy treatments (plant history, soil history, soil treatment), global change treatments (drought, nitrogen input) and their interactions on plant performance (total biomass, shoot biomass, root biomass and root-shoot ratio).

Shown are degrees of freedom (Df), Chi² and p-values (p). Significant effects (p < 0.05) are given in bold, marginally significant effects (p < 0.1) in italics.

| | Total biomass | | | Shoot biomass | | | Root biomass | | | Root-shoot ratio | | |
|---|---|---|---|---|---|---|---|---|---|---|---|---|
| | Df | Chi | p | Df | Chi | p | Df | Chi | p | Df | Chi | p |
| Species ID | 3 | 73.25 | **<0.001** | 3 | 80.17 | **<0.001** | 3 | 121.30 | **<0.001** | 3 | 133.41 | **<0.001** |
| Plant history | 1 | 3.48 | *0.062* | 1 | 1.36 | 0.244 | 1 | 3.40 | *0.065* | 1 | 1.11 | 0.292 |
| Soil history | 1 | 0.01 | 0.915 | 1 | 0.04 | 0.851 | 1 | 0.49 | 0.484 | 1 | 1.08 | 0.300 |
| Soil treatment | 2 | 2.17 | 0.338 | 2 | 1.20 | 0.548 | 2 | 3.66 | 0.161 | 2 | 1.81 | 0.404 |
| Drought (D) | 1 | 83.05 | **<0.001** | 1 | 110.26 | **<0.001** | 1 | 2.81 | *0.094* | 1 | 60.01 | **<0.001** |
| Nitrogen input (N) | 1 | 257.26 | **<0.001** | 1 | 425.93 | **<0.001** | 1 | 15.89 | **<0.001** | 1 | 89.83 | **<0.001** |
| D x N | 1 | 29.23 | **<0.001** | 1 | 23.02 | **<0.001** | 1 | 8.50 | **0.004** | 1 | 1.75 | 0.185 |
| Plant history x D | 1 | 0.22 | 0.639 | 1 | 0.21 | 0.643 | 1 | 0.01 | 0.916 | 1 | <0.01 | 0.977 |
| Soil history x D | 1 | <0.01 | 0.944 | 1 | 0.07 | 0.786 | 1 | 0.10 | 0.746 | 1 | 0.23 | 0.635 |
| Soil treatment x D | 2 | 1.79 | 0.409 | 2 | 0.77 | 0.681 | 2 | 1.37 | 0.503 | 2 | 1.29 | 0.526 |
| Plant history x N | 1 | 1.48 | 0.224 | 1 | 1.59 | 0.207 | 1 | 0.60 | 0.437 | 1 | 0.35 | 0.553 |
| Soil history x N | 1 | 3.44 | *0.064* | 1 | 1.33 | 0.249 | 1 | 2.46 | 0.116 | 1 | 0.83 | 0.363 |
| Soil treatment x N | 2 | 1.43 | 0.489 | 2 | 1.40 | 0.496 | 2 | 0.43 | 0.806 | 2 | 0.49 | 0.782 |
| Plant history x D x N | 1 | 2.12 | 0.146 | 1 | 0.84 | 0.358 | 1 | 1.78 | 0.183 | 1 | 1.27 | 0.260 |
| Soil history x D x N | 1 | 0.95 | 0.330 | 1 | 2.78 | *0.095* | 1 | 0.08 | 0.780 | 1 | 0.03 | 0.864 |
| Soil treatment x D x N | 2 | 1.37 | 0.504 | 2 | 1.93 | 0.381 | 2 | 0.91 | 0.635 | 2 | 0.73 | 0.693 |

**Appendix 2—table 3.** Summary of mixed-effect model analyses testing the effects of species identity (N = 4), legacy treatments (plant history, soil history, soil treatment), global change treatments (drought, nitrogen input) and their interactions on plant trait expression.

Shown are degrees of freedom (Df), Chi² and p-values (p). Significant effects (p < 0.05) are given in bold, marginally significant effects (p < 0.1) in italics.

| | Growth height | | | Shoot nitrogen conc. | | | Leaf greenness | | |
|---|---|---|---|---|---|---|---|---|---|
| | Df | Chi² | p | Df | Chi² | p | Df | Chi² | p |
| Species ID | 3 | 71.45 | **<0.001** | 3 | 57.20 | **<0.001** | 3 | 79.55 | **<0.001** |
| Plant history | 1 | 0.15 | 0.694 | 1 | <0.01 | 0.960 | 1 | 0.05 | 0.830 |
| Soil history | 1 | 1.60 | 0.207 | 1 | 0.64 | 0.425 | 1 | 0.17 | 0.683 |
| Soil treatment | 2 | 3.98 | 0.137 | 2 | 2.27 | 0.321 | 2 | 0.60 | 0.742 |
| Drought (D) | 1 | 18.71 | **<0.001** | 1 | 65.46 | **<0.001** | 1 | 66.15 | **<0.001** |
| Nitrogen input (N) | 1 | 32.93 | **<0.001** | 1 | 772.20 | **<0.001** | 1 | 523.86 | **<0.001** |
| D x N | 1 | 1.10 | 0.294 | 1 | 48.85 | **<0.001** | 1 | <0.01 | 0.997 |
| Plant history x D | 1 | 2.99 | *0.084* | 1 | 0.06 | 0.806 | 1 | <0.01 | 0.950 |
| Soil history x D | 1 | 0.51 | 0.477 | 1 | 0.11 | 0.735 | 1 | 1.57 | 0.210 |
| Soil treatment x D | 2 | 3.54 | 0.171 | 2 | 0.02 | 0.990 | 2 | 0.69 | 0.707 |
| Plant history x N | 1 | 0.50 | 0.478 | 1 | 1.34 | 0.246 | 1 | 0.91 | 0.341 |
| Soil history x N | 1 | 1.41 | 0.235 | 1 | 0.19 | 0.666 | 1 | 1.54 | 0.215 |

*Continued on next page*

| | Growth height | | | Shoot nitrogen conc. | | | Leaf greenness | | |
|---|---|---|---|---|---|---|---|---|---|
| | Df | Chi² | p | Df | Chi² | p | Df | Chi² | p |
| Soil treatment x N | 2 | 1.87 | 0.392 | 2 | 3.30 | 0.192 | 2 | 2.42 | 0.299 |
| Plant history x D x N | 1 | 0.83 | 0.364 | 1 | 0.21 | 0.645 | 1 | 0.79 | 0.373 |
| Soil history x D x N | 1 | 0.69 | 0.407 | 1 | 3.06 | 0.080 | 1 | <0.01 | 0.977 |
| Soil treatment x D x N | 2 | 4.94 | 0.085 | 2 | 1.56 | 0.458 | 2 | 0.04 | 0.983 |

| | LDMC | | | SLA | | | Stomatal conductance | | |
|---|---|---|---|---|---|---|---|---|---|
| | Df | Chi² | p | Df | Chi² | p | Df | Chi² | p |
| Air temperature | - | - | - | - | - | - | 1 | 5.34 | 0.021 |
| Daytime | - | - | - | - | - | - | 1 | 38.25 | <0.001 |
| Species ID | 3 | 80.52 | <0.001 | 3 | 124.00 | <0.001 | 3 | 47.15 | <0.001 |
| Plant history | 1 | 0.80 | 0.373 | 1 | 0.06 | 0.805 | 1 | 1.25 | 0.264 |
| Soil history | 1 | 0.10 | 0.750 | 1 | 1.22 | 0.270 | 1 | 0.37 | 0.543 |
| Soil treatment | 2 | 1.13 | 0.570 | 2 | 1.64 | 0.441 | 2 | 3.38 | 0.185 |
| Drought (D) | 1 | 0.94 | 0.333 | 1 | 0.11 | 0.743 | 1 | 0.90 | 0.343 |
| Nitrogen input (N) | 1 | 62.84 | <0.001 | 1 | 61.63 | <0.001 | 1 | 8.16 | 0.004 |
| D x N | 1 | 6.69 | 0.010 | 1 | 0.01 | 0.904 | 1 | 9.33 | 0.002 |
| Plant history x D | 1 | 0.04 | 0.841 | 1 | 0.34 | 0.559 | 1 | 0.06 | 0.806 |
| Soil history x D | 1 | 0.49 | 0.484 | 1 | 0.02 | 0.883 | 1 | 0.65 | 0.420 |
| Soil treatment x D | 2 | 0.24 | 0.887 | 2 | 0.23 | 0.889 | 2 | 0.18 | 0.914 |
| Plant history x N | 1 | 0.65 | 0.421 | 1 | 0.16 | 0.688 | 1 | 0.69 | 0.406 |
| Soil history x N | 1 | 0.12 | 0.734 | 1 | 1.07 | 0.300 | 1 | 0.63 | 0.428 |
| Soil treatment x N | 2 | 0.66 | 0.719 | 2 | 2.92 | 0.232 | 2 | 0.08 | 0.960 |
| Plant history x D x N | 1 | 0.16 | 0.687 | 1 | 1.77 | 0.183 | 1 | 0.87 | 0.351 |
| Soil history x D x N | 1 | <0.01 | 0.962 | 1 | 0.95 | 0.331 | 1 | 0.02 | 0.887 |
| Soil treatment x D x N | 2 | 2.27 | 0.322 | 2 | 1.33 | 0.514 | 2 | 3.73 | 0.155 |

| | Root diameter | | | SRL | | | RLD | | |
|---|---|---|---|---|---|---|---|---|---|
| | Df | Chi² | p | Df | Chi² | p | Df | Chi² | p |
| Species ID | 3 | 165.58 | <0.001 | 3 | 174.84 | <0.001 | 3 | 125.84 | <0.001 |
| Plant history | 1 | 0.03 | 0.872 | 1 | 0.32 | 0.569 | 1 | 1.14 | 0.286 |
| Soil history | 1 | 0.37 | 0.544 | 1 | 0.36 | 0.546 | 1 | 0.25 | 0.617 |
| Soil treatment | 2 | 1.50 | 0.473 | 2 | 2.80 | 0.246 | 2 | 4.97 | 0.083 |
| Drought (D) | 1 | 11.19 | 0.001 | 1 | 7.67 | 0.006 | 1 | 16.09 | <0.001 |
| Nitrogen input (N) | 1 | 19.83 | <0.001 | 1 | 6.68 | 0.010 | 1 | 1.29 | 0.257 |
| D x N | 1 | 0.25 | 0.619 | 1 | 1.27 | 0.261 | 1 | 2.14 | 0.144 |
| Plant history x D | 1 | 0.37 | 0.544 | 1 | 0.34 | 0.559 | 1 | 0.67 | 0.414 |
| Soil history x D | 1 | 0.12 | 0.725 | 1 | 0.48 | 0.491 | 1 | 0.07 | 0.798 |
| Soil treatment x D | 2 | 1.67 | 0.434 | 2 | 0.65 | 0.723 | 2 | 0.44 | 0.802 |
| Plant history x N | 1 | 0.40 | 0.528 | 1 | 1.91 | 0.167 | 1 | <0.01 | 0.944 |
| Soil history x N | 1 | 0.42 | 0.515 | 1 | 0.15 | 0.703 | 1 | 0.86 | 0.353 |
| Soil treatment x N | 2 | 0.27 | 0.872 | 2 | 1.69 | 0.430 | 2 | 0.08 | 0.959 |
| Plant history x D x N | 1 | 0.20 | 0.652 | 1 | 1.22 | 0.270 | 1 | 0.12 | 0.734 |
| Soil history x D x N | 1 | 1.48 | 0.224 | 1 | 3.47 | 0.063 | 1 | 3.94 | 0.047 |
| Soil treatment x D x N | 2 | 0.75 | 0.686 | 2 | 0.84 | 0.659 | 2 | 1.02 | 0.600 |

**Appendix 2—table 4.** Summary of mixed-effect model analyses testing the effects legacy treatments (plant history, soil history, soil treatment), global change treatments (drought, nitrogen input) and their interactions on plant performance (total biomass, shoot biomass, root biomass and root-shoot ratio) of *A. elatius* and *A. pratensis*.

Shown are degrees of freedom (Df), Chi² and p-values (p). Significant effects (p < 0.05) are given in bold, marginally significant effects (p < 0.1) in italics.

| *A. elatius* | | | | | | | | | | | |
|---|---|---|---|---|---|---|---|---|---|---|---|
| | **Total biomass** | | | **Shoot biomass** | | | **Root biomass** | | | **Root-shoot ratio** | | |
| | **Df** | **Chi²** | **p** | **Df** | **Chi²** | **p** | **Df** | **Chi²** | **p** | **Df** | **Chi²** | **p** |
| Plant history | 1 | 0.26 | 0.609 | 1 | 0.05 | 0.827 | 1 | 0.10 | 0.747 | 1 | 0.11 | 0.738 |
| Soil history | 1 | 0.39 | 0.533 | 1 | 0.03 | 0.865 | 1 | 1.02 | 0.312 | 1 | 1.28 | 0.258 |
| Soil treatment | 2 | 2.06 | 0.357 | 2 | 1.59 | 0.452 | 2 | 1.31 | 0.520 | 2 | 0.30 | 0.861 |
| Drought (D) | 1 | 21.54 | **<0.001** | 1 | 50.79 | **<0.001** | 1 | 6.13 | **0.013** | 1 | 67.84 | **<0.001** |
| Nitrogen input (N) | 1 | 125.48 | **<0.001** | 1 | 128.72 | **<0.001** | 1 | 31.68 | **<0.001** | 1 | 13.70 | **<0.001** |
| D x N | 1 | 36.23 | **<0.001** | 1 | 45.06 | **<0.001** | 1 | 1.86 | 0.173 | 1 | 0.13 | 0.715 |
| Plant history x D | 1 | 1.01 | 0.315 | 1 | 2.37 | 0.123 | 1 | 0.05 | 0.823 | 1 | 1.28 | 0.258 |
| Soil history x D | 1 | 0.27 | 0.606 | 1 | 2.01 | 0.156 | 1 | 0.71 | 0.399 | 1 | 2.11 | 0.146 |
| Soil treatment x D | 2 | 1.21 | 0.545 | 2 | 3.22 | 0.200 | 2 | 0.13 | 0.939 | 2 | 1.21 | 0.545 |
| Plant history x N | 1 | 0.92 | 0.337 | 1 | 2.00 | 0.157 | 1 | 0.02 | 0.879 | 1 | 0.46 | 0.497 |
| Soil history x N | 1 | 0.87 | 0.352 | 1 | 0.05 | 0.832 | 1 | 1.37 | 0.242 | 1 | 2.29 | 0.130 |
| Soil treatment x N | 2 | 3.07 | 0.215 | 2 | 0.80 | 0.669 | 2 | 6.25 | **0.044** | 2 | 5.64 | *0.060* |
| Plant history x D x N | 1 | 0.07 | 0.792 | 1 | <0.01 | 0.980 | 1 | 0.15 | 0.696 | 1 | 0.02 | 0.884 |
| Soil history x D x N | 1 | 0.61 | 0.434 | 1 | 0.05 | 0.822 | 1 | 0.89 | 0.344 | 1 | 1.17 | 0.279 |
| Soil treatment x D x N | 2 | 3.61 | 0.165 | 2 | 2.25 | 0.326 | 2 | 1.33 | 0.515 | 2 | 0.56 | 0.757 |

| *A. pratensis* | | | | | | | | | | | |
|---|---|---|---|---|---|---|---|---|---|---|---|
| | **Total biomass** | | | **Shoot biomass** | | | **Root biomass** | | | **Root-shoot ratio** | | |
| | **Df** | **Chi²** | **p** | **Df** | **Chi²** | **p** | **Df** | **Chi²** | **p** | **Df** | **Chi²** | **p** |
| Plant history | 1 | 0.57 | 0.452 | 1 | 0.42 | 0.518 | 1 | 0.43 | 0.512 | 1 | <0.01 | 0.985 |
| Soil history | 1 | 0.68 | 0.408 | 1 | <0.01 | 0.945 | 1 | 1.47 | 0.225 | 1 | 0.80 | 0.371 |
| Soil treatment | 2 | 0.34 | 0.845 | 2 | 0.29 | 0.865 | 2 | 0.23 | 0.892 | 2 | 0.07 | 0.967 |
| Drought (D) | 1 | 71.43 | **<0.001** | 1 | 38.06 | **<0.001** | 1 | 60.92 | **<0.001** | 1 | 0.15 | 0.696 |
| Nitrogen input (N) | 1 | 74.74 | **<0.001** | 1 | 162.92 | **<0.001** | 1 | 9.71 | **0.002** | 1 | 55.50 | **<0.001** |
| D x N | 1 | 26.47 | **<0.001** | 1 | 3.98 | **0.046** | 1 | 24.94 | **<0.001** | 1 | 16.49 | **<0.001** |
| Plant history x D | 1 | 0.08 | 0.772 | 1 | 0.51 | 0.477 | 1 | 0.48 | 0.488 | 1 | 1.07 | 0.301 |
| Soil history x D | 1 | 0.43 | 0.512 | 1 | 0.37 | 0.546 | 1 | 0.20 | 0.653 | 1 | 0.01 | 0.912 |
| Soil treatment x D | 2 | 1.17 | 0.557 | 2 | 0.19 | 0.911 | 2 | 2.12 | 0.346 | 2 | 3.60 | 0.165 |
| Plant history x N | 1 | 0.40 | 0.529 | 1 | 1.26 | 0.261 | 1 | 0.02 | 0.875 | 1 | 0.14 | 0.709 |
| Soil history x N | 1 | 5.45 | **0.020** | 1 | 1.19 | 0.275 | 1 | 4.53 | **0.033** | 1 | 1.24 | 0.265 |
| Soil treatment x N | 2 | 2.78 | 0.249 | 2 | 2.50 | 0.287 | 2 | 1.21 | 0.547 | 2 | 0.13 | 0.938 |
| Plant history x D x N | 1 | 0.55 | 0.458 | 1 | 0.02 | 0.881 | 1 | 0.59 | 0.442 | 1 | 0.08 | 0.771 |
| Soil history x D x N | 1 | 0.28 | 0.595 | 1 | 0.30 | 0.585 | 1 | 0.78 | 0.376 | 1 | 1.44 | 0.230 |
| Soil treatment x D x N | 2 | 0.91 | 0.634 | 2 | 0.05 | 0.975 | 2 | 1.45 | 0.485 | 2 | 2.41 | 0.300 |

**Appendix 2—table 5.** Summary of mixed-effect model analyses testing the effects legacy treatments (plant history, soil history, soil treatment), global change treatments (drought, nitrogen input) and their interactions on plant performance (total biomass, shoot biomass, root biomass and

root-shoot ratio) of *D. glomerata* and *P. trivialis*.

Shown are degrees of freedom (Df), Chi² and p-values (p). Significant effects (p < 0.05) are given in bold, marginally significant effects (p < 0.1) in italics.

|  | *D. glomerata* | | | | | | | | | | | |
|---|---|---|---|---|---|---|---|---|---|---|---|---|
|  | **Total biomass** | | | **Shoot biomass** | | | **Root biomass** | | | **Root-shoot ratio** | | |
|  | Df | Chi² | p | Df | Chi² | p | Df | Chi² | p | Df | Chi² | p |
| Plant history | 1 | 1.51 | 0.219 | 1 | 1.32 | 0.251 | 1 | 1.12 | 0.289 | 1 | 0.19 | 0.662 |
| Soil history | 1 | 0.00 | 0.957 | 1 | 0.01 | 0.912 | 1 | 0.07 | 0.787 | 1 | 0.05 | 0.829 |
| Soil treatment | 2 | 0.79 | 0.673 | 2 | 0.11 | 0.948 | 2 | 2.65 | 0.266 | 2 | 2.94 | 0.230 |
| Drought (D) | 1 | 0.98 | 0.323 | 1 | 12.71 | **<0.001** | 1 | 20.48 | **<0.001** | 1 | 58.54 | **<0.001** |
| Nitrogen input (N) | 1 | 82.06 | **<0.001** | 1 | 124.42 | **<0.001** | 1 | 8.87 | **0.003** | 1 | 16.79 | **<0.001** |
| D x N | 1 | 0.07 | 0.790 | 1 | 0.04 | 0.843 | 1 | 0.61 | 0.434 | 1 | 0.53 | 0.467 |
| Plant history x D | 1 | 0.05 | 0.821 | 1 | 0.55 | 0.458 | 1 | 0.24 | 0.623 | 1 | 1.40 | 0.236 |
| Soil history x D | 1 | 0.56 | 0.453 | 1 | 2.20 | 0.138 | 1 | 0.14 | 0.706 | 1 | 0.27 | 0.601 |
| Soil treatment x D | 2 | 0.09 | 0.955 | 2 | 0.55 | 0.758 | 2 | 1.09 | 0.579 | 2 | 3.01 | 0.222 |
| Plant history x N | 1 | 1.55 | 0.213 | 1 | 1.85 | 0.174 | 1 | 0.62 | 0.432 | 1 | 0.29 | 0.592 |
| Soil history x N | 1 | 1.42 | 0.234 | 1 | 2.24 | 0.135 | 1 | 0.26 | 0.612 | 1 | 0.25 | 0.618 |
| Soil treatment x N | 2 | 0.05 | 0.976 | 2 | 0.72 | 0.699 | 2 | 1.94 | 0.378 | 2 | 3.83 | 0.147 |
| Plant history x D x N | 1 | 4.64 | **0.031** | 1 | 3.35 | *0.067* | 1 | 4.09 | **0.043** | 1 | 3.81 | *0.051* |
| Soil history x D x N | 1 | 4.21 | **0.040** | 1 | 3.68 | *0.055* | 1 | 2.87 | *0.090* | 1 | 1.64 | 0.200 |
| Soil treatment x D x N | 2 | 1.70 | 0.428 | 2 | 3.03 | 0.220 | 2 | 0.66 | 0.718 | 2 | 0.32 | 0.853 |

|  | *P. trivialis* | | | | | | | | | | | |
|---|---|---|---|---|---|---|---|---|---|---|---|---|
|  | **Total biomass** | | | **Shoot biomasp** | | | **Root biomass** | | | **Root-shoot ratio** | | |
|  | Df | Chi² | p | Df | Chi² | P | Df | Chi² | p | Df | Chi² | p |
| Plant history | 1 | 0.91 | 0.340 | 1 | 0.03 | 0.870 | 1 | 1.49 | 0.222 | 1 | 1.36 | 0.244 |
| Soil history | 1 | 0.26 | 0.611 | 1 | 0.08 | 0.781 | 1 | 2.43 | 0.119 | 1 | 4.29 | **0.038** |
| Soil treatment | 2 | 1.23 | 0.540 | 2 | 1.18 | 0.556 | 2 | 0.62 | 0.732 | 2 | 0.09 | 0.956 |
| Drought (D) | 1 | 23.05 | **<0.001** | 1 | 22.42 | **<0.001** | 1 | 8.93 | **0.003** | 1 | 0.00 | 0.988 |
| Nitrogen input (N) | 1 | 27.28 | **<0.001** | 1 | 87.31 | **<0.001** | 1 | 1.12 | 0.290 | 1 | 45.86 | **<0.001** |
| D x N | 1 | 3.81 | *0.051* | 1 | 2.16 | 0.141 | 1 | 2.81 | *0.094* | 1 | 2.10 | 0.147 |
| Plant history x D | 1 | 0.08 | 0.775 | 1 | 1.03 | 0.311 | 1 | 0.03 | 0.874 | 1 | 0.20 | 0.656 |
| Soil history x D | 1 | <0.01 | 0.969 | 1 | 0.21 | 0.649 | 1 | 0.15 | 0.696 | 1 | 0.21 | 0.646 |
| Soil treatment x D | 2 | 0.80 | 0.670 | 2 | 0.69 | 0.708 | 2 | 0.38 | 0.828 | 2 | 1.04 | 0.594 |
| Plant history x N | 1 | <0.01 | 0.972 | 1 | 0.87 | 0.350 | 1 | 0.32 | 0.569 | 1 | 0.73 | 0.391 |
| Soil history x N | 1 | <0.01 | 0.984 | 1 | 0.01 | 0.936 | 1 | 0.01 | 0.920 | 1 | 0.03 | 0.857 |
| Soil treatment x N | 2 | 4.20 | 0.123 | 2 | 1.87 | 0.392 | 2 | 6.33 | **0.042** | 2 | 7.28 | **0.026** |
| Plant history x D x N | 1 | 0.25 | 0.614 | 1 | <0.01 | 0.978 | 1 | 0.17 | 0.680 | 1 | 0.00 | 0.972 |
| Soil history x D x N | 1 | 0.02 | 0.890 | 1 | 1.11 | 0.292 | 1 | 1.09 | 0.296 | 1 | 2.88 | *0.089* |
| Soil treatment x D x N | 2 | 0.35 | 0.838 | 2 | 0.49 | 0.782 | 2 | 1.16 | 0.559 | 2 | 1.97 | 0.373 |

**Appendix 2—table 6.** Summary of mixed-effect model analyses testing the effects of legacy treatments (plant history, soil history, soil treatment), global change treatments (drought, nitrogen input) and their interactions on plant trait expressions of *A. elatius*.

Shown are degrees of freedom (Df), Chi² and p-values (p). Significant effects (p < 0.05) are given in bold, marginally significant effects (p < 0.1) in italics.

| A. elatius | Growth height | | | Shoot nitrogen conc. | | | Leaf greenness | | |
|---|---|---|---|---|---|---|---|---|---|
| | Df | Chi² | p | Df | Chi² | p | Df | Chi² | p |
| Plant history | 1 | 0.24 | 0.625 | 1 | 0.12 | 0.725 | 1 | 0.07 | 0.795 |
| Soil history | 1 | 0.61 | 0.436 | 1 | 0.36 | 0.547 | 1 | 0.67 | 0.413 |
| Soil treatment | 2 | 2.01 | 0.365 | 2 | 0.80 | 0.670 | 2 | 0.19 | 0.907 |
| Drought (D) | 1 | 2.11 | 0.146 | 1 | 36.64 | **<0.001** | 1 | 30.19 | **<0.001** |
| Nitrogen input (N) | 1 | 5.35 | **0.021** | 1 | 142.97 | **<0.001** | 1 | 153.54 | **<0.001** |
| D x N | 1 | 0.02 | 0.881 | 1 | 32.71 | **<0.001** | 1 | 0.27 | 0.604 |
| Plant history x D | 1 | 4.68 | **0.030** | 1 | 1.41 | 0.236 | 1 | 0.48 | 0.487 |
| Soil history x D | 1 | 0.01 | 0.904 | 1 | 0.26 | 0.612 | 1 | 0.06 | 0.813 |
| Soil treatment x D | 2 | 3.10 | 0.212 | 2 | 0.38 | 0.827 | 2 | 1.58 | 0.453 |

*Continued on next page*

| *A. elatius* | Growth height | | | Shoot nitrogen conc. | | | Leaf greenness | | |
|---|---|---|---|---|---|---|---|---|---|
| | Df | Chi² | p | Df | Chi² | p | Df | Chi² | p |
| Plant history x N | 1 | 1.15 | 0.284 | 1 | 1.08 | 0.300 | 1 | 3.76 | *0.053* |
| Soil history x N | 1 | 0.61 | 0.434 | 1 | 0.20 | 0.656 | 1 | 1.09 | 0.295 |
| Soil treatment x N | 2 | 3.03 | 0.220 | 2 | 0.27 | 0.874 | 2 | 2.37 | 0.305 |
| Plant history x D x N | 1 | 0.59 | 0.443 | 1 | 1.85 | 0.174 | 1 | 0.37 | 0.545 |
| Soil history x D x N | 1 | 0.93 | 0.334 | 1 | 0.03 | 0.854 | 1 | 0.06 | 0.813 |
| Soil treatment x D x N | 2 | 7.64 | **0.022** | 2 | 0.26 | 0.877 | 2 | 1.95 | 0.377 |

| | LDMC | | | SLA | | | Stomatal conductance | | |
|---|---|---|---|---|---|---|---|---|---|
| | Df | Chi² | P | Df | Chi² | P | Df | Chi² | P |
| Air temperature | - | - | - | - | - | - | 1 | <0.01 | 0.948 |
| Daytime | - | - | - | - | - | - | 1 | 8.05 | **0.005** |
| Plant history | 1 | 0.46 | 0.500 | 5 | 1.69 | 0.194 | 1 | 0.49 | 0.486 |
| Soil history | 1 | 0.19 | 0.666 | 6 | 1.83 | 0.176 | 1 | 0.05 | 0.823 |
| Soil treatment | 2 | 1.37 | 0.504 | 8 | 1.14 | 0.565 | 2 | 3.38 | 0.184 |
| Drought (D) | 1 | 7.57 | **0.006** | 9 | 12.37 | **<0.001** | 1 | 4.58 | **0.032** |
| Nitrogen input (N) | 1 | 1.05 | 0.307 | 10 | 0.05 | 0.832 | 1 | 2.00 | 0.158 |
| D x N | 1 | 0.02 | 0.889 | 11 | 1.87 | 0.171 | 1 | 0.17 | 0.681 |
| Plant history x D | 1 | 1.48 | 0.224 | 12 | 1.94 | 0.164 | 1 | 1.08 | 0.298 |
| Soil history x D | 1 | 0.36 | 0.549 | 13 | 0.79 | 0.373 | 1 | 0.05 | 0.830 |
| Soil treatment x D | 2 | <0.01 | 0.998 | 15 | 1.73 | 0.420 | 2 | 0.73 | 0.693 |
| Plant history x N | 1 | 0.01 | 0.904 | 16 | 0.08 | 0.782 | 1 | 0.04 | 0.836 |
| Soil history x N | 1 | 0.01 | 0.936 | 17 | 1.69 | 0.193 | 1 | 0.36 | 0.549 |
| Soil treatment x N | 2 | 2.16 | 0.339 | 19 | 2.01 | 0.367 | 2 | 0.24 | 0.886 |
| Plant history x D x N | 1 | <0.01 | 0.999 | 20 | 1.96 | 0.162 | 1 | 0.42 | 0.518 |
| Soil history x D x N | 1 | 0.10 | 0.752 | 21 | 0.15 | 0.696 | 1 | 1.48 | 0.224 |
| Soil treatment x D x N | 2 | 0.35 | 0.840 | 23 | 0.50 | 0.781 | 2 | 1.99 | 0.369 |

| | Root diameter | | | SRL | | | RLD | | |
|---|---|---|---|---|---|---|---|---|---|
| | Df | Chi² | P | Df | Chi² | P | Df | Chi² | P |
| Plant history | 1 | 0.08 | 0.783 | 1 | 0.31 | 0.576 | 1 | 0.09 | 0.767 |
| Soil history | 1 | 0.23 | 0.629 | 1 | 0.22 | 0.639 | 1 | 0.82 | 0.364 |
| Soil treatment | 2 | 2.89 | 0.236 | 2 | 5.30 | *0.071* | 2 | 3.35 | 0.187 |
| Drought (D) | 1 | 0.32 | 0.572 | 1 | 5.25 | **0.022** | 1 | 0.04 | 0.851 |
| Nitrogen input (N) | 1 | 3.46 | *0.063* | 1 | 13.72 | **<0.001** | 1 | 0.13 | 0.723 |
| D x N | 1 | 0.01 | 0.932 | 1 | 1.62 | 0.204 | 1 | <0.01 | 0.989 |
| Plant history x D | 1 | 0.39 | 0.531 | 1 | 0.11 | 0.740 | 1 | 0.77 | 0.380 |
| Soil history x D | 1 | 0.01 | 0.938 | 1 | 0.95 | 0.329 | 1 | 0.29 | 0.590 |
| Soil treatment x D | 2 | 2.11 | 0.349 | 2 | 0.51 | 0.775 | 2 | 0.45 | 0.797 |
| Plant history x N | 1 | 0.09 | 0.764 | 1 | 1.41 | 0.235 | 1 | 1.29 | 0.256 |
| Soil history x N | 1 | 1.35 | 0.246 | 1 | 0.32 | 0.573 | 1 | 3.53 | *0.060* |
| Soil treatment x N | 2 | 0.68 | 0.711 | 2 | 1.06 | 0.590 | 2 | 1.76 | 0.416 |
| Plant history x D x N | 1 | 1.68 | 0.194 | 1 | 2.73 | *0.099* | 1 | 3.70 | *0.054* |
| Soil history x D x N | 1 | 4.45 | **0.035** | 1 | 0.52 | 0.469 | 1 | 1.46 | 0.227 |
| Soil treatment x D x N | 2 | 2.00 | 0.369 | 2 | 2.75 | 0.253 | 2 | 2.26 | 0.324 |

**Appendix 2—table 7.** Summary of mixed-effect model analyses testing the effects of legacy treatments (plant history, soil history, soil treatment), global change treatments (drought, nitrogen input) and their interactions on plant trait expressions of *A. pratensis*.

Shown are degrees of freedom (Df), Chi$^2$ and p-values (p). Significant effects (p < 0.05) are given in bold, marginally significant effects (p < 0.1) in italics.

| *A. pratensis* | Growth height | | | Shoot nitrogen conc. | | | Leaf greenness | | |
|---|---|---|---|---|---|---|---|---|---|
| | Df | Chi$^2$ | p | Df | Chi$^2$ | p | Df | Chi$^2$ | p |
| Plant history | 1 | 1.35 | 0.246 | 1 | 0.16 | 0.687 | 1 | 0.49 | 0.485 |
| Soil history | 1 | 0.71 | 0.400 | 1 | <0.01 | 0.967 | 1 | 0.11 | 0.745 |
| Soil treatment | 2 | 8.50 | **0.014** | 2 | 1.38 | 0.501 | 2 | 0.20 | 0.903 |
| Drought (D) | 1 | 1.07 | 0.300 | 1 | 15.42 | **<0.001** | 1 | 16.09 | **<0.001** |
| Nitrogen input (N) | 1 | 10.63 | **0.001** | 1 | 246.65 | **<0.001** | 1 | 143.35 | **<0.001** |
| D x N | 1 | 1.40 | 0.236 | 1 | 17.58 | **<0.001** | 1 | 0.86 | 0.353 |
| Plant history x D | 1 | 0.16 | 0.692 | 1 | <0.01 | 0.979 | 1 | 0.58 | 0.446 |
| Soil history x D | 1 | 0.31 | 0.577 | 1 | 0.52 | 0.471 | 1 | 3.04 | *0.081* |
| Soil treatment x D | 2 | 1.11 | 0.575 | 2 | 0.50 | 0.778 | 2 | 3.39 | 0.183 |
| Plant history x N | 1 | 0.28 | 0.597 | 1 | 0.17 | 0.681 | 1 | <0.01 | 0.994 |
| Soil history x N | 1 | 0.01 | 0.919 | 1 | 0.10 | 0.747 | 1 | 1.10 | 0.293 |
| Soil treatment x N | 2 | 2.42 | 0.299 | 2 | 6.58 | **0.037** | 2 | 0.19 | 0.911 |
| Plant history x D x N | 1 | 0.18 | 0.672 | 1 | 0.87 | 0.352 | 1 | 1.06 | 0.304 |
| Soil history x D x N | 1 | 0.45 | 0.501 | 1 | 0.49 | 0.485 | 1 | 0.03 | 0.863 |
| Soil treatment x D x N | 2 | 0.85 | 0.654 | 2 | 2.08 | 0.353 | 2 | 0.32 | 0.854 |

| | LDMC | | | SLA | | | Stomatal conductance | | |
|---|---|---|---|---|---|---|---|---|---|
| | Df | Chi$^2$ | P | Df | Chi$^2$ | P | Df | Chi$^2$ | P |
| Air temperature | - | - | - | - | - | - | 1 | 0.16 | 0.685 |
| Daytime | - | - | - | - | - | - | 1 | 1.78 | 0.182 |
| Plant history | 1 | 2.82 | *0.093* | 1 | 0.19 | 0.665 | 1 | 0.43 | 0.513 |
| Soil history | 1 | 1.80 | 0.180 | 1 | 0.94 | 0.332 | 1 | 0.41 | 0.520 |
| Soil treatment | 2 | 3.57 | 0.168 | 2 | 5.69 | *0.058* | 2 | 3.67 | 0.159 |
| Drought (D) | 1 | 4.02 | **0.045** | 1 | 1.29 | 0.255 | 1 | 6.17 | **0.013** |
| Nitrogen input (N) | 1 | 0.75 | 0.388 | 1 | 2.93 | *0.087* | 1 | 3.64 | *0.056* |
| D x N | 1 | 0.33 | 0.566 | 1 | 0.41 | 0.524 | 1 | 3.45 | *0.063* |
| Plant history x D | 1 | 0.13 | 0.715 | 1 | 0.27 | 0.604 | 1 | 0.03 | 0.862 |
| Soil history x D | 1 | 0.16 | 0.685 | 1 | <0.01 | 0.980 | 1 | 0.64 | 0.423 |
| Soil treatment x D | 2 | 1.40 | 0.497 | 2 | 1.39 | 0.499 | 2 | 0.01 | 0.993 |
| Plant history x N | 1 | 1.03 | 0.311 | 1 | 1.02 | 0.313 | 1 | 0.58 | 0.447 |
| Soil history x N | 1 | <0.01 | 0.950 | 1 | 0.78 | 0.377 | 1 | 0.18 | 0.669 |
| Soil treatment x N | 2 | 0.64 | 0.726 | 2 | 2.56 | 0.278 | 2 | 0.27 | 0.874 |
| Plant history x D x N | 1 | 0.80 | 0.372 | 1 | 1.67 | 0.197 | 1 | 2.57 | 0.109 |
| Soil history x D x N | 1 | 4.17 | **0.041** | 1 | 1.01 | 0.315 | 1 | 0.23 | 0.634 |
| Soil treatment x D x N | 2 | 0.18 | 0.912 | 2 | 1.09 | 0.581 | 2 | 15.71 | **<0.001** |

| | Root diameter | | | SRL | | | RLD | | |
|---|---|---|---|---|---|---|---|---|---|
| | Df | Chi$^2$ | P | Df | Chi$^2$ | P | Df | Chi$^2$ | P |
| Plant history | 1 | 0.01 | 0.935 | 1 | 0.28 | 0.597 | 1 | 0.06 | 0.809 |
| Soil history | 1 | 0.18 | 0.676 | 1 | 0.01 | 0.934 | 1 | 0.92 | 0.337 |

*Continued on next page*

*Continued*

| | Root diameter | | | SRL | | | RLD | | |
|---|---|---|---|---|---|---|---|---|---|
| | Df | Chi² | P | Df | Chi² | P | Df | Chi² | P |
| Soil treatment | 2 | 0.54 | 0.763 | 2 | 0.97 | 0.615 | 2 | 0.12 | 0.940 |
| Drought (D) | 1 | 39.31 | **<0.001** | 1 | 5.25 | **0.022** | 1 | 82.01 | **<0.001** |
| Nitrogen input (N) | 1 | 51.80 | **<0.001** | 1 | 5.33 | **0.021** | 1 | 0.34 | 0.560 |
| D x N | 1 | 0.09 | 0.767 | 1 | 5.57 | **0.018** | 1 | 4.32 | **0.038** |
| Plant history x D | 1 | 0.01 | 0.906 | 1 | 0.30 | 0.587 | 1 | 0.26 | 0.611 |
| Soil history x D | 1 | 0.09 | 0.769 | 1 | 0.01 | 0.910 | 1 | 0.02 | 0.877 |
| Soil treatment x D | 2 | 2.58 | 0.276 | 2 | 4.88 | 0.087 | 2 | 0.11 | 0.948 |
| Plant history x N | 1 | 0.03 | 0.869 | 1 | 0.19 | 0.660 | 1 | 0.17 | 0.682 |
| Soil history x N | 1 | 6.39 | **0.011** | 1 | 8.14 | **0.004** | 1 | 0.63 | 0.426 |
| Soil treatment x N | 2 | 1.82 | 0.402 | 2 | 3.27 | 0.195 | 2 | 1.24 | 0.539 |
| Plant history x D x N | 1 | 0.54 | 0.461 | 1 | 0.15 | 0.700 | 1 | 0.28 | 0.594 |
| Soil history x D x N | 1 | 1.82 | 0.178 | 1 | 1.87 | 0.172 | 1 | 0.27 | 0.605 |
| Soil treatment x D x N | 2 | 3.23 | 0.199 | 2 | 1.63 | 0.443 | 2 | 0.70 | 0.703 |

**Appendix 2—table 8.** Summary of mixed-effect model analyses testing the effects of legacy treatments (plant history, soil history, soil treatment), global change treatments (drought, nitrogen input) and their interactions on plant trait expressions of *D. glomerata*.
Shown are degrees of freedom (Df), Chi² and p-values (p). Significant effects (p < 0.05) are given in bold, marginally significant effects (p < 0.1) in italics.

| *D. glomerata* | Growth height | | | Shoot nitrogen conc. | | | Leaf greenness | | |
|---|---|---|---|---|---|---|---|---|---|
| | Df | Chi² | p | Df | Chi² | p | Df | Chi² | p |
| Plant history | 1 | 0.05 | 0.831 | 1 | 0.58 | 0.444 | 1 | 0.22 | 0.640 |
| Soil history | 1 | 1.56 | 0.212 | 1 | 1.35 | 0.245 | 1 | 0.27 | 0.606 |
| Soil treatment | 2 | 5.25 | *0.073* | 2 | 0.75 | 0.687 | 2 | 0.55 | 0.760 |
| Drought (D) | 1 | <0.01 | 0.976 | 1 | 19.10 | **<0.001** | 1 | 29.41 | **<0.001** |
| Nitrogen input (N) | 1 | 11.51 | **0.001** | 1 | 183.85 | **<0.001** | 1 | 172.91 | **<0.001** |
| D x N | 1 | <0.01 | 0.949 | 1 | 3.72 | 0.054 | 1 | 0.08 | 0.781 |
| Plant history x D | 1 | 0.01 | 0.920 | 1 | 0.05 | 0.828 | 1 | 2.75 | *0.097* |
| Soil history x D | 1 | 0.82 | 0.366 | 1 | 0.08 | 0.774 | 1 | 0.22 | 0.639 |
| Soil treatment x D | 2 | 0.48 | 0.785 | 2 | 0.25 | 0.880 | 2 | 0.21 | 0.899 |
| Plant history x N | 1 | 0.91 | 0.341 | 1 | 2.96 | *0.086* | 1 | 0.61 | 0.437 |
| Soil history x N | 1 | 0.23 | 0.633 | 1 | 0.32 | 0.571 | 1 | 1.75 | 0.186 |
| Soil treatment x N | 2 | 0.35 | 0.840 | 2 | 0.29 | 0.866 | 2 | 4.92 | *0.085* |
| Plant history x D x N | 1 | 0.12 | 0.733 | 1 | 1.62 | 0.204 | 1 | 0.54 | 0.462 |
| Soil history x D x N | 1 | 0.71 | 0.400 | 1 | 5.07 | **0.024** | 1 | <0.01 | 0.998 |
| Soil treatment x D x N | 2 | 0.06 | 0.969 | 2 | 2.15 | 0.341 | 2 | 0.33 | 0.846 |

| | LDMC | | | SLA | | | Stomatal conductance | | |
|---|---|---|---|---|---|---|---|---|---|
| | Df | Chi² | p | Df | Chi² | p | Df | Chi² | p |
| Air temperature | - | - | - | - | - | - | 1 | 0.39 | 0.531 |
| Daytime | - | - | - | - | - | - | 1 | 20.31 | **<0.001** |
| Plant history | 1 | 0.12 | 0.727 | 1 | 0.80 | 0.371 | 1 | 5.08 | **0.024** |
| Soil history | 1 | 0.58 | 0.445 | 1 | 0.32 | 0.573 | 1 | <0.01 | 0.944 |
| Soil treatment | 2 | 0.58 | 0.749 | 2 | 1.20 | 0.548 | 2 | 0.54 | 0.765 |

*Continued on next page*

*Continued*

| | LDMC | | | SLA | | | Stomatal conductance | | |
|---|---|---|---|---|---|---|---|---|---|
| | Df | Chi² | p | Df | Chi² | p | Df | Chi² | p |
| Drought (D) | 1 | 0.07 | 0.798 | 1 | 0.54 | 0.461 | 1 | 9.01 | **0.003** |
| Nitrogen input (N) | 1 | 55.57 | **<0.001** | 1 | 57.43 | **<0.001** | 1 | 2.72 | *0.099* |
| D x N | 1 | 20.69 | **<0.001** | 1 | 6.61 | **0.010** | 1 | 6.34 | **0.012** |
| Plant history x D | 1 | 0.04 | 0.842 | 1 | 0.46 | 0.498 | 1 | 0.07 | 0.793 |
| Soil history x D | 1 | 0.01 | 0.926 | 1 | 0.09 | 0.762 | 1 | <0.01 | 0.991 |
| Soil treatment x D | 2 | 1.43 | 0.490 | 2 | 0.09 | 0.958 | 2 | 0.19 | 0.907 |
| Plant history x N | 1 | 0.99 | 0.320 | 1 | 0.02 | 0.893 | 1 | 0.32 | 0.571 |
| Soil history x N | 1 | 2.48 | 0.115 | 1 | 2.19 | 0.139 | 1 | 0.69 | 0.406 |
| Soil treatment x N | 2 | 0.13 | 0.938 | 2 | 1.56 | 0.459 | 2 | 0.09 | 0.958 |
| Plant history x D x N | 1 | 2.00 | 0.157 | 1 | 0.09 | 0.768 | 1 | 0.33 | 0.566 |
| Soil history x D x N | 1 | 1.30 | 0.254 | 1 | 4.99 | **0.026** | 1 | 5.98 | **0.014** |
| Soil treatment x D x N | 2 | 3.56 | 0.169 | 2 | 1.09 | 0.579 | 2 | 1.57 | 0.456 |

| | Root diameter | | | SRL | | | RLD | | |
|---|---|---|---|---|---|---|---|---|---|
| | Df | Chi² | p | Df | Chi² | p | Df | Chi² | p |
| Plant history | 1 | 0.60 | 0.438 | 1 | 0.96 | 0.326 | 1 | 2.61 | 0.107 |
| Soil history | 1 | 0.06 | 0.805 | 1 | 0.07 | 0.791 | 1 | 0.01 | 0.933 |
| Soil treatment | 2 | 0.07 | 0.967 | 2 | 0.58 | 0.749 | 2 | 2.44 | 0.296 |
| Drought (D) | 1 | 0.93 | 0.335 | 1 | 1.16 | 0.281 | 1 | 9.45 | **0.002** |
| Nitrogen input (N) | 1 | 1.22 | 0.270 | 1 | 0.37 | 0.545 | 1 | 7.05 | **0.008** |
| D x N | 1 | 0.80 | 0.370 | 1 | 1.73 | 0.189 | 1 | 0.08 | 0.773 |
| Plant history x D | 1 | 3.60 | *0.058* | 1 | 0.64 | 0.425 | 1 | 0.25 | 0.614 |
| Soil history x D | 1 | 1.41 | 0.235 | 1 | 0.62 | 0.430 | 1 | 0.23 | 0.632 |
| Soil treatment x D | 2 | 0.65 | 0.721 | 2 | 1.95 | 0.377 | 2 | 2.43 | 0.297 |
| Plant history x N | 1 | 0.19 | 0.667 | 1 | 2.05 | 0.152 | 1 | 0.03 | 0.854 |
| Soil history x N | 1 | 0.60 | 0.437 | 1 | <0.01 | 0.994 | 1 | 0.21 | 0.646 |
| Soil treatment x N | 2 | 0.85 | 0.653 | 2 | 0.97 | 0.616 | 2 | 1.76 | 0.414 |
| Plant history x D x N | 1 | 1.49 | 0.222 | 1 | 0.14 | 0.712 | 1 | 3.11 | *0.078* |
| Soil history x D x N | 1 | 0.49 | 0.483 | 1 | 3.54 | *0.060* | 1 | 1.07 | 0.301 |
| Soil treatment x D x N | 2 | 1.65 | 0.438 | 2 | 1.16 | 0.559 | 2 | 0.20 | 0.907 |

**Appendix 2—table 9.** Summary of mixed-effect model analyses testing the effects of legacy treatments (plant history, soil history, soil treatment), global change treatments (drought, nitrogen input) and their interactions on plant trait expressions of *P. trivialis*.

Shown are degrees of freedom (Df), Chi² and P-values (P). Significant effects (P < 0.05) are given in bold, marginally significant effects (P < 0.1) in italics.

| *P. trivialis* | Growth height | | | Shoot nitrogen conc. | | | Leaf greenness | | |
|---|---|---|---|---|---|---|---|---|---|
| | Df | Chi² | P | Df | Chi² | P | Df | Chi² | P |
| Plant history | 1 | 0.06 | 0.800 | 1 | 0.00 | 0.997 | 1 | 0.93 | 0.334 |
| Soil history | 1 | 2.29 | 0.131 | 1 | 0.05 | 0.824 | 1 | 1.10 | 0.294 |
| Soil treatment | 2 | 1.66 | 0.435 | 2 | 0.51 | 0.776 | 2 | 1.15 | 0.563 |
| Drought (D) | 1 | 30.17 | **<0.001** | 1 | 5.46 | **0.019** | 1 | 1.42 | 0.233 |
| Nitrogen input (N) | 1 | 12.16 | **<0.001** | 1 | 297.03 | **<0.001** | 1 | 108.82 | **<0.001** |

*Appendix 2—table 9 Continued on next page*

*Appendix 2—table 9 Continued*

| P. trivialis | Growth height | | | Shoot nitrogen conc. | | | Leaf greenness | | |
|---|---|---|---|---|---|---|---|---|---|
| | Df | Chi² | P | Df | Chi² | P | Df | Chi² | P |
| D x N | 1 | 1.72 | 0.190 | 1 | 17.06 | **<0.001** | 1 | 1.09 | 0.296 |
| Plant history x D | 1 | 0.22 | 0.637 | 1 | 0.11 | 0.736 | 1 | 3.08 | *0.079* |
| Soil history x D | 1 | 2.28 | 0.131 | 1 | 0.53 | 0.469 | 1 | 0.06 | 0.806 |
| Soil treatment x D | 2 | 3.11 | 0.211 | 2 | 1.03 | 0.598 | 2 | 0.18 | 0.916 |
| Plant history x N | 1 | 5.16 | **0.023** | 1 | 0.05 | 0.821 | 1 | 0.13 | 0.719 |
| Soil history x N | 1 | 3.49 | **0.062** | 1 | 0.04 | 0.842 | 1 | 0.36 | 0.549 |
| Soil treatment x N | 2 | 2.08 | 0.354 | 2 | 1.04 | 0.594 | 2 | 1.98 | 0.371 |
| Plant history x D x N | 1 | 0.92 | 0.336 | 1 | 0.03 | 0.865 | 1 | 0.11 | 0.738 |
| Soil history x D x N | 1 | 0.13 | 0.718 | 1 | 0.18 | 0.669 | 1 | 0.00 | 0.967 |
| Soil treatment x D x N | 2 | 2.11 | 0.348 | 2 | 5.57 | *0.062* | 2 | 1.74 | 0.418 |

| | LDMC | | | SLA | | | Stomatal conductance | | |
|---|---|---|---|---|---|---|---|---|---|
| | Df | Chi² | P | Df | Chi² | P | Df | Chi² | P |
| Air temperature | - | - | - | - | - | - | 1 | 38.70 | **<0.001** |
| Daytime | - | - | - | - | - | - | 1 | 18.64 | **<0.001** |
| Plant history | 1 | <0.01 | 0.965 | 1 | 0.62 | 0.431 | 1 | 0.18 | 0.675 |
| Soil history | 1 | 0.08 | 0.777 | 1 | 0.49 | 0.485 | 1 | 0.71 | 0.399 |
| Soil treatment | 2 | 1.64 | 0.441 | 2 | 2.12 | 0.346 | 2 | 3.25 | 0.197 |
| Drought (D) | 1 | 2.85 | *0.091* | 1 | 2.75 | *0.097* | 1 | 0.22 | 0.636 |
| Nitrogen input (N) | 1 | 57.72 | **<0.001** | 1 | 41.44 | **<0.001** | 1 | 0.06 | 0.800 |
| D x N | 1 | 0.39 | 0.534 | 1 | 0.62 | 0.431 | 1 | 2.87 | *0.090* |
| Plant history x D | 1 | 1.09 | 0.296 | 1 | 0.38 | 0.540 | 1 | 2.86 | *0.091* |
| Soil history x D | 1 | 2.26 | 0.133 | 1 | 0.45 | 0.502 | 1 | 0.01 | 0.908 |
| Soil treatment x D | 2 | 0.19 | 0.908 | 2 | 1.33 | 0.515 | 2 | 0.40 | 0.819 |
| Plant history x N | 1 | 3.37 | *0.066* | 1 | 1.56 | 0.212 | 1 | 0.35 | 0.554 |
| Soil history x N | 1 | 0.54 | 0.461 | 1 | 0.21 | 0.645 | 1 | 2.45 | 0.118 |
| Soil treatment x N | 2 | 1.89 | 0.388 | 2 | 3.10 | 0.213 | 2 | 1.36 | 0.508 |
| Plant history x D x N | 1 | 0.13 | 0.720 | 1 | 0.58 | 0.446 | 1 | 0.14 | 0.704 |
| Soil history x D x N | 1 | 1.15 | 0.283 | 1 | 1.01 | 0.315 | 1 | 7.44 | **0.006** |
| Soil treatment x D x N | 2 | 3.30 | 0.192 | 2 | 0.99 | 0.610 | 2 | 2.20 | 0.333 |

| | Root diameter | | | SRL | | | RLD | | |
|---|---|---|---|---|---|---|---|---|---|
| | Df | Chi² | P | Df | Chi² | P | Df | Chi² | P |
| Plant history | 1 | 2.10 | 0.147 | 1 | 2.38 | 0.123 | 1 | 0.04 | 0.840 |
| Soil history | 1 | 0.08 | 0.781 | 1 | 0.30 | 0.581 | 1 | 1.31 | 0.253 |
| Soil treatment | 2 | 0.13 | 0.938 | 2 | 0.31 | 0.856 | 2 | 1.13 | 0.568 |
| Drought (D) | 1 | 14.18 | **<0.001** | 1 | 0.89 | 0.347 | 1 | 18.25 | **<0.001** |
| Nitrogen input (N) | 1 | 0.17 | 0.677 | 1 | 3.49 | *0.062* | 1 | 0.03 | 0.872 |
| D x N | 1 | 0.88 | 0.349 | 1 | 0.25 | 0.618 | 1 | 1.16 | 0.282 |
| Plant history x D | 1 | 0.40 | 0.525 | 1 | 0.27 | 0.602 | 1 | 0.16 | 0.692 |
| Soil history x D | 1 | 0.48 | 0.487 | 1 | 0.20 | 0.655 | 1 | 1.36 | 0.244 |
| Soil treatment x D | 2 | 5.85 | *0.054* | 2 | 0.50 | 0.777 | 2 | 0.43 | 0.808 |
| Plant history x N | 1 | 1.28 | 0.258 | 1 | 0.07 | 0.795 | 1 | 0.28 | 0.594 |
| Soil history x N | 1 | 1.21 | 0.271 | 1 | 0.36 | 0.549 | 1 | 0.65 | 0.418 |

*Continued on next page*

*Continued*

| | Root diameter | | | SRL | | | RLD | | |
|---|---|---|---|---|---|---|---|---|---|
| | Df | Chi² | P | Df | Chi² | P | Df | Chi² | P |
| Soil treatment x N | 2 | 2.99 | 0.225 | 2 | 9.11 | **0.011** | 2 | 0.33 | 0.846 |
| Plant history x D x N | 1 | 0.33 | 0.566 | 1 | 0.05 | 0.821 | 1 | 0.02 | 0.878 |
| Soil history x D x N | 1 | 4.11 | **0.043** | 1 | 9.74 | **0.002** | 1 | 2.06 | 0.151 |
| Soil treatment x D x N | 2 | 0.52 | 0.772 | 2 | 1.40 | 0.495 | 2 | 1.43 | 0.488 |

**Appendix 2—table 10.** Summary of mixed-effect model analyses testing the effects of legacy treatments (plant history, soil history, soil treatment), global change treatments (drought, nitrogen input) and their interactions on mildew infestation of *D. glomerata* and *P. trivialis*.

Shown are degrees of freedom (Df), Chi² and p-values (p). Significant effects (p < 0.05) are given in bold, marginally significant effects (p < 0.1) in italics.

| Mildew infestation | *D. glomerata* | | | *P. trivialis* | | |
|---|---|---|---|---|---|---|
| | Df | Chi² | p | Df | Chi² | p |
| Plant history | 1 | 0.29 | 0.588 | 1 | 0.01 | 0.939 |
| Soil history | 1 | 0.24 | 0.622 | 1 | 4.16 | **0.041** |
| Soil treatment | 2 | 0.22 | 0.896 | 2 | 3.36 | 0.187 |
| Drought (D) | 1 | 2.44 | 0.119 | 1 | 10.69 | **0.001** |
| Nitrogen input (N) | 1 | 42.75 | **<0.001** | 1 | 38.76 | **<0.001** |
| D x N | 1 | 1.05 | 0.305 | 1 | 0.98 | 0.321 |
| Plant history x D | 1 | 0.03 | 0.855 | 1 | 0.02 | 0.889 |
| Soil history x D | 1 | 2.25 | 0.134 | 1 | 0.07 | 0.788 |
| Soil treatment x D | 2 | 5.79 | *0.055* | 2 | 0.25 | 0.884 |
| Plant history x N | 1 | <0.01 | 0.953 | 1 | 0.25 | 0.614 |
| Soil history x N | 1 | 0.21 | 0.643 | 1 | 0.50 | 0.477 |
| Soil treatment x N | 2 | 0.32 | 0.854 | 2 | 1.22 | 0.544 |
| Plant history x D x N | 1 | 3.00 | *0.083* | 1 | 0.09 | 0.770 |
| Soil history x D x N | 1 | 1.69 | 0.193 | 1 | 0.93 | 0.335 |
| Soil treatment x D x N | 2 | 7.15 | **0.028** | 2 | 0.62 | 0.734 |

## Appendix 3

## Calculation of irrigation water quantity per pot

1) After one week of growing, pots were watered until 100% saturation and then weighted ( = Weight $_{\text{wet soil}}$).

2) To determine the amount of water, which is needed to get 60% water saturation (control value), we used the following equations:

$$(I)\frac{22\% \left(\text{water holding capacity}\right) \times 60\% \text{ saturation}}{100\% \text{ saturation}} = 13.2\%$$

$$(II)\text{Weight}_{\text{wet soil}} - \frac{\text{Weight}_{\text{wet soil}} \times 100\% \text{ saturation}}{13.2\% + 100} = \text{Weight}_{\text{water control}}$$

First, we multiplied the water holding capacity of the Jena Experiment soil-sand mix (22%) times 60% saturation and then divided the result by 100% saturation. Second, Weight $_{\text{wet soil}}$ was multiplied with 100 and then divided by 113.2. Third, the calculated weight for a 60% saturation was subtracted from Weight $_{\text{wet soil}}$ per pot and averaged over all pots, which resulted in 380 ml water.

3) Drought was simulated by 50% lower water saturation (30% saturation), while the amount of water was calculated as followed:

$$(I)\frac{22\% \left(\text{water holding capacity}\right) \times 30\% \text{ saturation}}{100\% \text{ saturation}} = 6.6\%$$

$$(II)\text{Weight}_{\text{wet soil}} - \frac{\text{Weight}_{\text{wet soil}} \times 100\% \text{ saturation}}{6.6\% + 100} = \text{Weight}_{\text{water drought}}$$

