## [Editor Report]

Dietrich et al. aimed to test the hypothesis that a decline in species richness due to various global change drivers selects for traits that will make species more vulnerable to the further effects of these drivers, amplifying thus the initial diversity decline. This research is of prime importance to botanists, plant ecologists, and ecosystem ecologists wanting to model the effects of global climate change on plant diversity and productivity.

---

## [Decision Letter]

**Decision letter after peer review:**

Thank you for submitting your article "Diversity-induced plant history and soil history effects modulate plant responses to global change" for consideration by *eLife*. Your article has been reviewed by 2 peer reviewers, and the evaluation has been overseen by a Reviewing Editor and Detlef Weigel as the Senior Editor. The reviewers have opted to remain anonymous.

Essential revisions:

Both reviewers found inconsistencies between the aims, the framework set at the intro, and the results. Please review the concerns carefully, and, importantly, provide a clear reasoning of your aims in light of the fact that the statistical support for some of the most interesting results was limited.

*Reviewer #1 (Recommendations for the authors):*

Generally, I like the study, namely its concept. The research is cleverly-designed and well-performed, nicely utilizing the set-up of the Jena experiment. The manuscript is well written, with sound analyses as far as I can say. Despite a large number of response variables the authors measured, they succeeded to well-organize the results and the whole paper. However, what I am missing here is a clear take-home message: the reader gets an impression that all factors (selection history, soil origin, drought/N-addition) have some effect (and I indeed appreciate summary of the results in Figure 3) on plant performance. However, I think the paper would be much stronger, if you can conclude whether you detected (or not) an expected eco-evolutionary feedback loop between global change drivers and community diversity (p. 4, l. 80-84), i.e., that initial species loss generates further diversity decline through evolutionary changes in plants. It would be great if you can somehow project population dynamics of your grass species depending on selection history when they face the studied global change drivers, i.e. drought/fertilization (though I am not sure whether you have available data for doing so).

I would also suggest to add a variance decomposition to show how strong the effect of individual factors and namely their interaction on plant performance was.

Further comments:

– Please, provide reference (or data in Supplementary material) to a study that measured differences in selection regimes in low- and high-diversity plots. You hypothesized that the accumulation of soil-borne pathogens may cause persistent species to adapt to these changes by producing more defense compounds in low-diversity plots (p. 4, l. 64-67). But it would be great if you can provide evidence on that! I would expect that selection regime will change not only below-ground, but also above-ground (For example, due to higher stem density in high-diversity plots. I would then expect, for example, differences in light availability, soil moisture between low- and high-diversity plots). More info would be appreciated.

– It is not clear how you controlled for maternal environmental effects. If I got it correctly, you used seeds collected directly in the field (so you did not remove possible maternal effects by obtaining F1 seeds from plants grown in common environment). I do not mind, but you should comment on that. Related to that, could you provide more details about evolutionary processes in low- and high-diversity plots? I mean to provide reference to a study to a study from the JENA exp. that dealt with evolutionary processes in your plots (and that evaluated processes such as changes in (epi)genotypic frequencies, genetic drift etc.).

*Reviewer #2 (Recommendations for the authors):*

The second sentence of the abstract is very vague-'these changes' and how 'this affects their responses'- these should be specified.

line 46 "of particular concern"?

In line 98, you state the aim of this study is to test plant origin, soil origin, and soil treatment influence the response of plants to global change but make no reference to the impact of diversity. Is that included in plant origin?

I find Figure 2 to be quite difficult to follow given the large amount of information. Is there a way this could be distilled? Perhaps some of this information in a table and the rest in bar chart form?

Specific comments regarding manuscript framing

Is 6 plant species reflective of 'high diversity'? This would really depend on the system I suppose. How would the findings of this work extend to a system where 6 species is considered relatively low diversity (e.g. tropical or alpine systems). The authors should justify their use of 6 species as reflective of high diversity, especially when this was not the highest level in the experiment (9 species max).

Several of the hypothesized outcomes of this study are not fully described and supported in the introduction. This requires some additional information and literature review.

For example, in your introduction you set up the expectation that plants with history of growing in high diversity communities and in 'home' soil would have higher productivity. These correspond to hypotheses H1a and H1b however you don't set up the rationale for how high/low diversity soils would differentially influence plant responses to global change (H1c). Some justification/explanation for this is needed as higher diversity in soil microbial communities is not always correlated with higher plant growth. Or is this referring to soil from high diversity plant communities? Either way it is unclear.

Furthermore, in the introduction you discuss how plant diversity can influence plant productivity and community stability over time but there is no discussion of the different traits/resource use strategies that are presented in H1d. This should be expanded upon.

Broadly, I find hypothesis 2 to be unnecessary because we already know these patterns broadly from the literature. Unfortunately, this is where you found most of the results and focus the majority of your discussion.

Comments regarding results/conclusions

Your interpretation of the microbial mechanisms driving the biomass responses for A pratensis and P. trivalis (i.e. that they interact more with soil mutualists-line 361-364) are purely speculative because you did not test the actual roots or rhizosphere of these species for mycorrhizal abundance. There are other mechanisms that could be driving this response such as a lack of pathogens or a different saprotrophic community that creates higher nutrient availability.

Despite the title, you actually found very few examples across multiple species where soil history and plant history influence plant responses to global change. Looking at Figure 5, I only see one example (plus two marginally significant). Given this question whether this title is an accurate representation of the results as a whole and whether the take-home messages of this paper and over-reaching. Figure 2 shows a number of species specific responses that are interesting but perhaps not generalizable broadly.

[Editors' note: further revisions were suggested prior to acceptance, as described below.]

Thank you for resubmitting your work entitled "Eco-evolutionary feedbacks modulate plant responses to global change depending on plant diversity and species identity" for further consideration by *eLife*. Your revised article has been evaluated by Detlef Weigel (Senior Editor) and a Reviewing Editor.

The manuscript has been improved but there are some remaining issues that need to be addressed, as outlined below:

While most of these changes are for clarity, reviewers stressed, shall you agree, that evidence of a feedback loop (the connection back to community diversity) seemed weak. This lack of evidence does not 'break' the paper, but if you can use more precise language and tone down your references to feedback, the issue could be resolved.

*Reviewer #2 (Recommendations for the authors):*

As I indicated in my previous review, I like the study of Peter Dietrich and his collaborators aiming at testing the hypothesis that a species loss due to global change drivers will select for traits that will make species more vulnerable to these drivers. In my previous review, I also indicated that the research is cleverly-designed and well-performed, nicely utilizing the set-up of the JENA experiment.

After reading the revised version I can say that the authors considered most of my suggestions, and corrected the text accordingly. I appreciate that the authors added variance decomposition to Table 2. I think that it would be useful to comment on the importance of different factors (of legacy treatments, and of global change driver treatments) in the text (and not only in Table 2).

Related to the previous point, it is evident from Table 2 (from variance decomposition) that the importance of legacy treatments such as those of plant history is very small. The value of the present study thus lies in the outlined concept. I still remain less convinced regarding evidence on the expected eco-evolutionary feedback loop between global change drivers and community diversity.

---

## [Author Response]

Essential revisions:Both reviewers found inconsistencies between the aims, the framework set at the intro, and the results. Please review the concerns carefully, and, importantly, provide a clear reasoning of your aims in light of the fact that the statistical support for some of the most interesting results was limited.

We thank the editor and the reviewers from *eLife* for their detailed and constructive suggestions to improve the manuscript and the opportunity to submit a revised version. We have revised our manuscript taking carefully into account all the advice.

Reviewer #1 (Recommendations for the authors):Generally, I like the study, namely its concept. The research is cleverly-designed and well-performed, nicely utilizing the set-up of the Jena experiment. The manuscript is well written, with sound analyses as far as I can say. Despite a large number of response variables the authors measured, they succeeded to well-organize the results and the whole paper.

Thank you!

However, what I am missing here is a clear take-home message: the reader gets an impression that all factors (selection history, soil origin, drought/N-addition) have some effect (and I indeed appreciate summary of the results in Figure 3) on plant performance. However, I think the paper would be much stronger, if you can conclude whether you detected (or not) an expected eco-evolutionary feedback loop between global change drivers and community diversity (p. 4, l. 80-84), i.e., that initial species loss generates further diversity decline through evolutionary changes in plants.

We agree. Thus, we have revised the conclusion to make the take-home message clearer and more explicit: “In the present study, we showed for the first time that offspring of plants selected at low and high plant diversity differently respond to global change and that plant-soil interactions play a significant role in this process. Although we did not find clear evidence that plants selected at low diversity generally suffer more under global change than plants selected at high diversity, our results indicate that highly-productive dominant species differently respond to global change drivers than low-productive subordinate species due to differences in eco-evolutionary feedbacks. These results are in line with a recent field experiment using the same grass species, which experienced a natural summer drought (Dietrich et al. 2021b). Our findings suggest that global change drivers increase the probability that dominant species outcompete subordinate species over time leading to a further loss of species and thus an acceleration of global change effects. We conclude that not only direct effects of global change drivers, but related changes in eco-evolutionary feedbacks could promote the impact of global change and lead to a feedback loop. To confirm this assumption, future research should test the long-term influence of global change drivers on soil biota and plants selected at different diversity under more realistic conditions, such as plants growing in communities under field conditions.”

It would be great if you can somehow project population dynamics of your grass species depending on selection history when they face the studied global change drivers, i.e. drought/fertilization (though I am not sure whether you have available data for doing so).

Thank you for your suggestion. Unfortunately, we do not have data on population dynamics under the influence of global change drivers for the Dominance Experiment. However, we did a two-year field study, where we used the same seed material of the plant species to grow them as phytometers on the experimental plots with soil history after removing the vegetation. In this field experiment, we did not manipulate global change drivers, but the phytometer plants experienced a very harsh summer drought in 2018 (Dietrich et al. 2021b). Interestingly, in both experiments we found that dominant species showed different eco-evolutionary feedbacks than subordinate species, which can explain why we found different responses to global change drivers in the common garden experiment. We added this to the conclusion.

I would also suggest to add a variance decomposition to show how strong the effect of individual factors and namely their interaction on plant performance was.

We thank the reviewer for this helpful idea. We calculated variance decomposition for the full model (for biomass production) and added the results to Table 1.

Further comments:– Please, provide reference (or data in Supplementary material) to a study that measured differences in selection regimes in low- and high-diversity plots. You hypothesized that the accumulation of soil-borne pathogens may cause persistent species to adapt to these changes by producing more defense compounds in low-diversity plots (p. 4, l. 64-67). But it would be great if you can provide evidence on that! I would expect that selection regime will change not only below-ground, but also above-ground (For example, due to higher stem density in high-diversity plots. I would then expect, for example, differences in light availability, soil moisture between low- and high-diversity plots). More info would be appreciated.

Thank you for the comment. We have added more references dealing with altered selection environment for plants due to changes in the soil biota community (L 73):

– Dietrich et al. (2020) Nematode communities, plant nutrient economy and life-cycle characteristics jointly determine plant monoculture performance over 12 years. Oikos, 129, 466-479. - monocultures of the Jena Experiment show loss of biomass production over time and altered values of traits related to growth.

– Mraja et al. (2011) Plant community diversity influences allocation to direct chemical defence in *Plantago lanceolata*. PLoS One, 6, e28055. -> low plant diversity = more plant defense (in the Jena Experiment).

– Lau et al. (2011) Evolutionary ecology of plant–microbe interactions: soil microbial structure alters selection on plant traits. New Phytologist, 192, 215-224. -> soil biota influence the selection regime of plants.

We agree with Reviewer 1 that more factors, including aboveground factors (such as competition for light), play a role in plant microevolution. We have therefore added this to the introduction (L 59-63): “Next to these biotic drivers, also diversity-dependent differences in abiotic conditions can influence plant community productivity (Barry et al. 2019). Previous studies demonstrated that a decrease of species richness alters the vegetation structure and density (Lorentzen et al. 2008), which in turn can have strong impacts on the availability of light, water, and nutrients for plants (Bachmann et al. 2018; Fischer et al. 2019; Lange et al. 2019).”

– It is not clear how you controlled for maternal environmental effects. If I got it correctly, you used seeds collected directly in the field (so you did not remove possible maternal effects by obtaining F1 seeds from plants grown in common environment). I do not mind, but you should comment on that.

We agree. So, we added text to the Material and Method section: “It should be noted that we cannot exclude possible maternal effects in our common garden experiment, because we used seed material collected in different plots of the field experiment; however, differences in maternal effects can also be important drivers of eco-evolutionary feedbacks and can significantly influence plant responses to global change (Puy et al. 2022; Rottstock et al. 2017).”

Related to that, could you provide more details about evolutionary processes in low- and high-diversity plots? I mean to provide reference to a study to a study from the JENA exp. that dealt with evolutionary processes in your plots (and that evaluated processes such as changes in (epi)genotypic frequencies, genetic drift etc.).

Thank you for the helpful advice. We added text to the Introduction to address this in more detail (L 74-83): “Indeed, several studies in the Jena Experiment demonstrated that plants not only show phenotypic plasticity in response to altered growth conditions in low- and high-diversity communities, but provided first evidence for micro-evolutionary processes. For example, only four years after sowing, the grass species Lolium perenne L. showed genetic differentiation from the source population, which was probably due to genetic drift as well as genotype-specific interactions with other species in plant communities of different diversity (Nestmann et al. 2011). Moreover, a recent study by van Moorsel et al. (2019) in the Jena Experiment demonstrated genetic and epigenetic divergence among plants originated either from monocultures or mixtures in three out of five studied species, suggesting rapid emergence of low-diversity and high-diversity genotypes.”

Reviewer #2 (Recommendations for the authors):The second sentence of the abstract is very vague-'these changes' and how 'this affects their responses'- these should be specified.

We have changed it as follows: “However, little is known about how fast species can adapt to diversity loss and how this affects their responses to global change as reflected by their biomass production and trait expression.”

line 46 "of particular concern"?

We have changed the sentence part: “…whereby plant species are particularly affected due to their low mobility…”.

In line 98, you state the aim of this study is to test plant origin, soil origin, and soil treatment influence the response of plants to global change but make no reference to the impact of diversity. Is that included in plant origin?

Exactly, plants (seeds) were either originated from two- or six-species plant communities. The same applies to soil history: soil biota were either originated from two- or six-plant species communities. We added this information to the text (L 111-113).

I find Figure 2 to be quite difficult to follow given the large amount of information. Is there a way this could be distilled? Perhaps some of this information in a table and the rest in bar chart form?

We suspect that Reviewer 2 is referring to Figure 3 (which is now in fact figure 2 – we have changed the order of the figures). That figure indeed contains a lot of information (about legacy and global change effects on biomass production and trait expression). In an earlier version of this manuscript, we had bar charts and tables, but the amount was enormous, so either a lot of space would have been needed in the manuscript or everything would have been moved to the Appendix (the tables can still be found in the Appendix). Therefore, we decided to create an overview figure that bundles all the information, so that the reader gets a first impression of what was measured and how the plants change in terms of productivity and trait expression. Given the large amount of information contained in this manuscript, we think this is the best solution (which is also confirmed by Reviewer 1).

Specific comments regarding manuscript framingIs 6 plant species reflective of 'high diversity'? This would really depend on the system I suppose. How would the findings of this work extend to a system where 6 species is considered relatively low diversity (e.g. tropical or alpine systems). The authors should justify their use of 6 species as reflective of high diversity, especially when this was not the highest level in the experiment (9 species max).

We agree with the reviewer that a 6-species community can be a low diversity ecosystem. Nevertheless, we have shown in previous publications of the Dominance Experiment that aboveground community biomass production increased with increasing mixture species-richness even from two species to six species (Roscher et al., 2007). Moreover, a reduction from six to two species can be a strong change, especially for the soil community and thus plant-soil interaction effects (Dietrich et al. 2021). The reviewer should also note that we focus here mainly on local neighborhood interactions between plant species that are likely to differ substantially between 2 and 6 neighboring species and that species richness is dependent on the considered scale. These aspects are now better highlighted in the Materials and methods section: “It should be noted that from here on we refer to plots with two plant species as "low-diversity communities" and plots with six species as "high-diversity communities", although the species richness of high-diversity communities can be much greater in nature depending on the considered scale. We chose six-species plots as "high-diversity communities", because they were available with different community compositions and could better represent the effects of species richness as a measure of diversity than the replicates of the 9-species plots with the same species composition. Moreover, previous studies have shown that plant productivity and soil biota community already differ between 2- and 6-species communities (Dietrich et al. 2021a; Roscher et al. 2007).”

Several of the hypothesized outcomes of this study are not fully described and supported in the introduction. This requires some additional information and literature review.For example, in your introduction you set up the expectation that plants with history of growing in high diversity communities and in 'home' soil would have higher productivity. These correspond to hypotheses H1a and H1b however you don't set up the rationale for how high/low diversity soils would differentially influence plant responses to global change (H1c). Some justification/explanation for this is needed as higher diversity in soil microbial communities is not always correlated with higher plant growth. Or is this referring to soil from high diversity plant communities? Either way it is unclear.

In general, in H1, we expected that control plants (without global change driver treatment) differ in productivity and trait expression depending on plant history, soil history, and soil treatment. More precisely, in H1c, we expected that the soil of low-diversity plant communities would have a different effect on plant performance and trait expression than the soil of high-diversity plant communities, based on the fact that biodiversity experiments have shown that low plant diversity results in an accumulation of soil-borne pathogens (plants have to defend themselves and show lower growth and lower values of traits related to growth, e.g. specific leaf area or leaf greenness). We rephrased the hypotheses (especially H1) for a better understanding (L 117-133).

Furthermore, in the introduction you discuss how plant diversity can influence plant productivity and community stability over time but there is no discussion of the different traits/resource use strategies that are presented in H1d. This should be expanded upon.

As described above, we have rewritten H1 to explain this in more detail. In the Introduction, we also discussed changes in plant trait expression in more detail.

Broadly, I find hypothesis 2 to be unnecessary because we already know these patterns broadly from the literature. Unfortunately, this is where you found most of the results and focus the majority of your discussion.

We agree with Reviewer 2 that hypothesis 2 occupies a larger part of the discussion; however, we do not believe that this hypothesis provides no new insights or is redundant. There is a lack of knowledge on how drought and fertilization interact in their effects on plant growth and trait expression (especially for root traits), which is, however, important to understand, because the combined effect is becoming more common in nature (Rillig et al. 2019). In our study, we show that drought and fertilization did not have purely additive or neutralizing effects, but that plants often responded quite differently compared to a treatment with only one global change driver. Furthermore, the results are important to evaluate whether plants selected at high- and low-diversity communities differ from the “mean” response (related to H3).

Comments regarding results/conclusionsYour interpretation of the microbial mechanisms driving the biomass responses for A pratensis and P. trivalis (i.e. that they interact more with soil mutualists-line 361-364) are purely speculative because you did not test the actual roots or rhizosphere of these species for mycorrhizal abundance. There are other mechanisms that could be driving this response such as a lack of pathogens or a different saprotrophic community that creates higher nutrient availability.

Thank you for the advice. We agree that this was speculative. Thus, we rephrased this part: “We can only speculate about the underlying reasons. It is possible that soil of low-diversity communities containing A. elatius and/or D. glomerata had a higher number of (species-specific) pathogens than plots containing A. pratensis and/or P. trivialis due the higher productivity of A. elatius and D. glomerata in the field and thus more resources for pathogens. This accumulation of species-specific pathogens could lead to reduced productivity of A. elatius and D. glomerata offspring grown in low-diversity soil. However, it is also possible that A. elatius and D. glomerata benefit more, and A. pratensis and P. trivialis less, from soil mutualists, which can be more abundant in soil from high-diversity than in soil from low-diversity plant communities.”

Despite the title, you actually found very few examples across multiple species where soil history and plant history influence plant responses to global change. Looking at Figure 5, I only see one example (plus two marginally significant). Given this question whether this title is an accurate representation of the results as a whole and whether the take-home messages of this paper and over-reaching. Figure 2 shows a number of species specific responses that are interesting but perhaps not generalizable broadly.

We agree that we found little evidence for differences in productivity, when plants and soil biota were originated from low- or high-diversity plant communities (except when plants were fertilized); however, Figure 2 (before reordering Figure 3) shows that a wide range of other responses, i.e., plants differently change their phenotype and growth strategy (= trait expression) depend on plant origin (low- or high diversity community) and composition of soil biota (soil from low/high-diversity plant communities, home vs. away soil). We agree that these responses are species-specific, although, some responses (especially in terms of productivity) are similar in high-productivity vs. low-productivity species. We have rephrased the conclusion of our manuscript to better describe this. We also changed the title and made some additions to the abstract (L 31-32).

New title: “Eco-evolutionary feedbacks modulate plant responses to global change depending on plant diversity and species identity”.

References

Bachmann D., Roscher C., Buchmann N. (2018) How do leaf trait values change spatially and temporally with light availability in a grassland diversity experiment? Oikos 127:935-948.

Barry K.E., Mommer L., van Ruijven J., Wirth C., Wright A.J., Bai Y., Connolly J., De Deyn G.B., de Kroon H., Isbell F., Milcu A., Roscher C., Scherer-Lorenzen M., Schmid B., Weigelt A. (2019) The future of complementarity: disentangling causes from consequences. Trends in Ecology and Evolution 34:167-180.

Dietrich P., Cesarz S., Liu T., Roscher C., Eisenhauer N. (2021a) Effects of plant species diversity on nematode community composition and diversity in a long-term biodiversity experiment. Oecologia 197:297–311.

Dietrich P., Eisenhauer N., Otto P., Roscher C. (2021b) Plant history and soil history jointly influence the selection environment for plant species in a long‐term grassland biodiversity experiment. Ecology and Evolution 11:8156-8169.

Fischer C., Leimer S., Roscher C., Ravenek J., de Kroon H., Kreutziger Y., Baade J., Beßler H., Eisenhauer N., Weigelt A. (2019) Plant species richness and functional groups have different effects on soil water content in a decade‐long grassland experiment. Journal of Ecology 107:127-141.

Lange M., Koller-France E., Hildebrandt A., Oelmann Y., Wilcke W., Gleixner G. (2019) How plant diversity impacts the coupled water, nutrient and carbon cycles. Advances in Ecological Research 61:185-219.

Lorentzen S., Roscher C., Schumacher J., Schulze E.D., Schmid B. (2008) Species richness and identity affect the use of aboveground space in experimental grasslands. Perspectives in Plant Ecology, Evolution and Systematics 10:73-87.

Nestmann S., Sretenovic Rajicic T., Dehmer K., Fischer M., Schumacher J., Roscher C. (2011) Plant species diversity and composition of experimental grasslands affect genetic differentiation of Lolium perenne populations. Molecular Ecology 20:2188-2203.

Puy J., Carmona C.P., Hiiesalu I., Öpik M., de Bello F., Moora M. (2022) Mycorrhizal symbiosis alleviates plant water deficit within and across generations via phenotypic plasticity. Journal of Ecology 110:262-276.

Rillig M.C., Ryo M., Lehmann A., Aguilar-Trigueros C.A., Buchert S., Wulf A., Iwasaki A., Roy J., Yang G. (2019) The role of multiple global change factors in driving soil functions and microbial biodiversity. Science 366:886-890.

Roscher C., Schumacher J., Weisser W.W., Schmid B., Schulze E.D. (2007) Detecting the role of individual species for overyielding in experimental grassland communities composed of potentially dominant species. Oecologia 154:535-549.

Rottstock T., Kummer V., Fischer M., Joshi J. (2017) Rapid transgenerational effects in *Knautia arvensis* in response to plant community diversity. Journal of Ecology 105:714-725.

van Moorsel S.J., Schmid M.W., Wagemaker N., van Gurp T., Schmid B., Vergeer P. (2019) Evidence for rapid evolution in a grassland biodiversity experiment. Molecular Ecology 28:4097-4117.

[Editors' note: further revisions were suggested prior to acceptance, as described below.]

Reviewer #2 (Recommendations for the authors):As I indicated in my previous review, I like the study of Peter Dietrich and his collaborators aiming at testing the hypothesis that a species loss due to global change drivers will select for traits that will make species more vulnerable to these drivers. In my previous review, I also indicated that the research is cleverly-designed and well-performed, nicely utilizing the set-up of the JENA experiment.After reading the revised version I can say that the authors considered most of my suggestions, and corrected the text accordingly. I appreciate that the authors added variance decomposition to Table 2. I think that it would be useful to comment on the importance of different factors (of legacy treatments, and of global change driver treatments) in the text (and not only in Table 2).

We thank again Reviewer 2 for the nice assessment. We agree that the results of the variance decomposition should also be mentioned in the text. Therefore, we added this to the manuscript. Please see Lines 145-146 and 177-178.

Related to the previous point, it is evident from Table 2 (from variance decomposition) that the importance of legacy treatments such as those of plant history is very small. The value of the present study thus lies in the outlined concept. I still remain less convinced regarding evidence on the expected eco-evolutionary feedback loop between global change drivers and community diversity.

We agree that legacy effects were small, but we did find some significant effects (not only on biomass production, but also on trait expression), which suggest changes in eco-evolutionary feedbacks of plants due to loss of diversity. However, we fully agree that we have not found clear evidence for a feedback loop, i.e., that these changes in microevolution have a negative influence on the response of plants to global change. We have therefore revised the abstract and the conclusion to make this point more explicit.

Conclusion: “In the present study, we showed for the first time that offspring of plants selected at low and high plant diversity differently respond to global change and that plant-soil interactions play a significant role in this process. These differences were mainly related to changes in trait expression, while changes in biomass production were minor, and they were strongly dependent on plant species identity and their competitiveness in the field, as well as the type of global change driver (drought, nitrogen input, or both). Although we did not find clear evidence that plants selected at low diversity generally suffer more under global change than plants selected at high diversity, it is possible that the species-specific responses alter species interactions and accelerate global change effects in the long run. To better assess the risk of such a potential feedback loop, future research is urgently necessary, especially, studies that test the long-term influence of global change drivers on plants and soil biota selected at different diversity under more realistic conditions, e.g. as plants growing in communities under field conditions.**”**